# The Asymptotically Safe Standard Model:
# From quantum gravity to dynamical chiral symmetry breaking

Álvaro Pastor-Gutiérrez,[1,2] Jan M. Pawlowski,[2,3] and Manuel Reichert[4]

[1]*Max-Planck-Institut für Kernphysik P.O. Box 103980, D 69029, Heidelberg, Germany*
[2]*Institut für Theoretische Physik, Universität Heidelberg, Philosophenweg 16, 69120 Heidelberg, Germany*
[3]*ExtreMe Matter Institute EMMI, GSI Helmholtzzentrum für*
*Schwerionenforschung mbH, Planckstr. 1, 64291 Darmstadt, Germany*
[4]*Department of Physics and Astronomy, University of Sussex, Brighton, BN1 9QH, U.K.*

We present a comprehensive non-perturbative study of the phase structure of the asymptotically safe Standard Model. The physics scales included range from the asymptotically safe trans-Planckian regime in the ultraviolet, the intermediate high-energy regime with electroweak symmetry breaking to strongly correlated QCD in the infrared. All flows are computed with a self-consistent functional renormalisation group approach, using a vertex expansion in the fluctuation fields. In particular, this approach takes care of all physical threshold effects and the respective decoupling of ultraviolet degrees of freedom. Standard Model and gravity couplings and masses are fixed by their experimental low energy values. Importantly, we accommodate for the difference between the top pole mass and its Euclidean analogue. Both, the correct mass determination and the threshold effects have a significant impact on the qualitative properties, and in particular on the stability properties of the specific ultraviolet-infrared trajectory with experimental Standard Model physics in the infrared. We show that in the present rather advanced approximation the matter part of the asymptotically safe Standard Model has the same number of relevant parameters as the Standard Model, and is asymptotically free. This result is based on the novel UV fixed point found in the present work: the fixed point Higgs potential is flat but has two relevant directions. These results and their analysis are accompanied by a thorough discussion of the systematic error of the present truncation, also important for systematic improvements.

## I. INTRODUCTION

One of the most challenging open tasks in contemporary high energy physics is its ultraviolet (UV) closure including quantum gravity. In the past three decades, asymptotically safe gravity [1–3] has been established as a viable option for this endeavour, for reviews see [4–14].

Asymptotically safe matter-gravity systems potentially include the Asymptotically Safe Standard Model (ASSM), the UV-complete unification of the Standard Model (SM) with gravity. This minimal set-up for the fundamental theory of matter and gravity with an absence of new physics implies that our current understanding of particle physics may hold up to the Planck scale. Hence, all masses and couplings converge at an interacting asymptotically safe UV fixed point. The solidified existence of such a minimal set-up would also allow for a systematic and well-controlled extension toward UV-complete Beyond Standard Model scenarios including the potential exclusion of such UV-safe embeddings in specific models.

This highly interesting programme has been set-up and pursued in a number of works concerning the matter content compatible with asymptotic safety [15–23], as well as the relation to the IR physics [23–34], and beyond the SM physics [35–39]. The running of SM couplings and full Higgs potentials has also been studied without the inclusion of gravity in similar frameworks [40–47]. While the physical mechanisms in the UV and potential scenarios have been well understood, the control over the mechanisms is still lacking. For example, it is necessary to consider higher-order curvature invariants such as $R^2$

in order to reliably determine a bound on the field content compatible with asymptotic safety [23]. Also the gravity contributions to the matter couplings are only partially under control: while it has been well understood that gravity supports asymptotic freedom of the gauge coupling or is vanishing at leading order [23, 25, 48–52], the gravity contribution to the Yukawa coupling carries more intricacies [28, 53]. Moreover, so far, the full ASSM flows were investigated in perturbative threshold-free RG flows for the matter fields, in particular for sub-Planckian physics, as well as a (classical) $\phi^4$-approximation of the Higgs effective potential. For reliable predictions on the nature of the ASSM, a consistent RG flow including physical threshold effects is needed.

The ASSM covers physics at vastly different momentum scales ranging from trans-Planckian momenta with asymptotic safety in the UV to sub-Fermi momenta with strongly correlated quantum chromodynamics (QCD) in the infrared (IR). Its parameters or rather the specific UV-IR trajectory are fixed by their experimental values measured in the IR, which also fixes the respective UV fixed point. Importantly, the UV landscape includes several physically distinct fixed-point classes. They range from fixed points with stable asymptotically free matter parts (or shifted Gaußian fixed points), fixed points with fully interacting matter parts over unstable FPs to regimes without fixed points. These differences not only concern the existence, stability and interaction nature of the UV regime, but also the number of relevant directions. This can lead to predictive trajectories with fewer parameters in the matter sector of the ASSM than in the SM.

Trivial examples are boundary trajectories between the stable and unstable FP classes as well as the Gaußian FP class with a flat Higgs potential with only one relevant parameter, the Higgs mass.

While many of the SM parameters have little impact on the UV fixed point class, if varied largely about the experimental value, there are few whose precise value and correct IR-UV trajectory has a huge and qualitative impact on both, high energy physics below the Planck scale and the trans-Planckian UV physics. In particular, the values and correct scale dependence of the top quark and Higgs masses have a qualitative impact on the UV fixed point class of the ASSM, as well as the high energy stability of the Higgs potential.

In short, reliable access to even the *qualitative* physics properties of the ASSM require *quantitative* control of its physics at all scales. Accordingly, this task requires a systematic, self-consistent approach able to treat non-perturbative physics both in the UV and IR. In the present work, we add to this task by computing the UV-IR flow of the ASSM self-consistently within the functional renormalisation group (fRG) approach. Here, self-consistency refers to two important aspects: Firstly, all coupling parameters, whose flows are computed are fed back to the flow, which leads to full resummations. This property is required for the rapid convergence of physics results and is mandatory for a precision determination of the IR parameters of the ASSM. Secondly, all flows are computed within the renormalisation scheme inherent to the fRG approach. In particular, this scheme incorporates physical threshold effects naturally and self-consistently due to its Wilsonian nature. In terms of standard RG schemes this can be phrased as follows: the fRG scheme leads to independence of the physics results from the RG-point already in relatively simple approximations.

In the present work, we add substantially to this endeavour by considering for the first time a fully consistent fRG system of all couplings of the ASSM. In particular, the physical thresholds of the respective matter and gravity dynamics are included. This is required for reliable and quantitative access to the physics of electroweak spontaneous symmetry breaking and strong chiral dynamical symmetry breaking. Furthermore, we take into account that the Higgs mass parameter is related but not equivalent to the experimentally measured pole mass. We compute the pole mass of the top quark in the present non-perturbative setting and adjust the Higgs mass parameter such that the pole mass takes its experimental value. This is essential for a correct estimate of the metastability scale of the Higgs potential.

Finally, we compute the full effective Higgs potential in the trans-Planckian regime within a high order of the Taylor expansion in the Higgs field. This allows us to reveal the presence of a *novel* UV fixed point for the Higgs potential that is present for a large parameter range: while the full effective potential is *flat*, is features *two* relevant directions. One of them is aligned with the Higgs mass operator, but the operator of the most relevant

direction is non-polynomial in the Higgs field. In the current approximation, it is this novel non-trivial fixed point that is connected to the physical SM in the IR; where the parameters are fixed by experimental observables.

In contradistinction, the standard Gaußian fixed point potential features *one* relevant direction, the Higgs mass. Naturally, for a vanishing coupling of the most relevant non-polynomial operator this FP is embedded in the novel FP as a one-dimensional sub-manifold. Parameter values in this sub-manifold are not connected to the physical SM in the IR: the resulting Higgs and top mass are roughly $2.9\,\mathrm{GeV}$ from the central experimental values. Whether this distance is close or feasible is subject to the evaluation and interpretation of the systematic error of the present approximation. This is but one of the reasons for the rather detailed description of the systematic error estimates in the present work.

The physical thresholds and the dynamics of spontaneous symmetry breaking are specifically important for two interrelated reasons: Firstly, in the absence of any thresholds the infrared sector of the SM is not accessible, and the SM parameters have to be chosen for cutoff values at least above the electroweak scale. Such a procedure requires the identification of momentum and cutoff scales, which typically work qualitatively but not quantitatively. Secondly, the fRG renormalisation scheme differs significantly from the standard $\overline{\mathrm{MS}}$ and MOM schemes used in particle physics, and the use of the respective perturbative $\beta$-functions may only work qualitatively. In particular, the naive identifications of momentum and cutoff scales for physical thresholds such as the electroweak symmetry breaking scale may not be suitable. Indeed, we shall observe in the present computation, that it is nearly off by one order of magnitude. This has a significant impact on the parameter choices and hence on the selected UV-IR trajectory as well as potentially on the fixed point class.

To wrap up, the present work includes for the first time the sub-Planckian physics of spontaneous symmetry breaking, both in the electroweak sector and in QCD. Moreover, all physical threshold effects are included, and both properties are necessary (but not sufficient) for a precision determination of the matter parameters of the ASSM. Finally, we determine the pole mass of the top quark instead of estimating it by Euclidean running masses. This precision determination including that of the pole mass is of qualitative importance for the high energy stability range of the Higgs sector, as well as the UV fixed point physics. For the latter, we find in the present approximation *novel* fixed point properties: the matter fixed point is asymptotically free (Gaußian or shifted Gaußian), but with a flat Higgs potential with *two* relevant directions. Our analysis includes, also for the first time, a thorough error analysis of the present truncation, which suggests that the present findings have to be corroborated within systematic extensions of the truncation.

This work is organised as follows. In Section II we discuss the fRG approach to the ASSM and detail our approximation. In Section III, we discuss the existence,

stability, and physics properties of the UV fixed point and the ASSM. This also includes a part of the systematic error analysis, while most of it is distributed in the work. In Section IV we present our results on the full phase structure of the ASSM ranging from the asymptotically safe regime with the Reuter fixed point to the deep IR with QCD and chiral symmetry breaking, summarised in Figure 3. The novel property of a flat effective Higgs potential at the UV fixed point with two relevant directions is discussed in Section V. There we also discuss the fixed point landscape for general values of the gravity fixed point values. In Section VI we evaluate the predictivity of the UV scenario developed in the present work, including an error analysis of the main sources of the systematic error. In Section VII we conclude with a summary of our results, including a short discussion of some consequences. Most of the technical details have been deferred to Appendices.

## II. ASYMPTOTICALLY SAFE STANDARD MODEL

In the Asymptotically Safe Standard Model (ASSM), the coupled SM–quantum-gravity system is fully described as a quantum field theory. The underlying classical action consists of the classical gauge-fixed action of the SM and the gauge-fixed Einstein-Hilbert action of General Relativity,

$$S_{\text{ASSM}}[\phi] = S_{\text{SM}}[\phi] + S_{\text{gravity}}[\phi_{\text{grav}}],  \quad (1)$$

for the explicit form see Appendix A 1 (SM) and Appendix A 2 (gravity).

The field content $\phi$ is given by the gauge fields $\mathcal{A}_\mu$ of the SM gauge groups, $SU(3)_C \times SU(2)_L \times U(1)_Y$, as well as the matter fields, leptons and quarks, $l, \bar{l}, q, \bar{q}$ and the scalar doublet, $\Phi$. The metric field $g_{\mu\nu}$ is split into a flat Euclidean background metric $\bar{g}_{\mu\nu} = \delta_{\mu\nu}$ and a fluctuation field $h_{\mu\nu}$, which carries the dynamics of the metric,

$$g_{\mu\nu} = \delta_{\mu\nu} + \sqrt{16\pi G_N}\, h_{\mu\nu},  \quad (2)$$

We define the dynamical fluctuation superfield, which also includes the auxiliary ghost fields stemming from the gauge-fixing sector,

$$\phi = (\phi_{\text{grav}},\, \phi_{\text{SM}}),  \quad (3a)$$

with

$$\phi_{\text{grav}} = (h_{\mu\nu}, c_\mu, \bar{c}_\mu),$$
$$\phi_{\text{SM}} = (\mathcal{A}_\mu, \mathcal{C}, \bar{\mathcal{C}}, l, \bar{l}, q, \bar{q}, \Phi).  \quad (3b)$$

The matter superfield $\phi_{\text{SM}}$ comprises the gauge and ghost fields of the SM gauge group $\text{U}(1)_Y \times \text{SU}(2)_L \times \text{SU}(3)_C$,

$$\mathcal{A}_\mu = (B_\mu, A_\mu^a, G_\mu^b), \qquad \mathcal{C} = (c^a, c^b),  \quad (3c)$$

with the hypercharge gauge field $B_\mu$, the weak gauge fields $A_\mu^a$ with $a = 1, 2, 3$, and the gluons $G_\mu^b$ with $b = 1, ..., 8$. The field $\mathcal{C}$ contains the respective ghost fields of the weak and strong gauge groups. $\phi_{\text{SM}}$ also contains the three families of quarks $q$ and leptons $l$,

$$q = (d,\, u,\, s,\, c,\, b,\, t), \quad l = (e,\, \nu_e,\, \mu,\, \nu_\mu,\, \tau,\, \nu_\tau).  \quad (3d)$$

The full quantum effective action of the ASSM, $\Gamma[\phi]$ contains all interaction terms that are compatible with the symmetries of the ASSM.

### A. Functional RG for the ASSM

We are interested in the full UV-IR phase structure of the ASSM, ranging from the trans-Planckian asymptotically safe UV regime ruled by gravity to the electroweak (EW) scale with EW symmetry breaking, and finally to the deep IR to strongly correlated QCD with strong chiral symmetry breaking and confinement. For covering this vast range of scales and different physics we use the functional renormalisation group (fRG) approach, which already has proven its applicability in all these regimes. In this approach, an IR regulator is introduced in the path integral, which suppresses quantum fluctuations below a given IR cutoff scale $k$, leading to the respective scale-dependent effective action $\Gamma_k[\phi]$. Then, the full effective action $\Gamma[\phi] = \Gamma_{k=0}[\phi]$ is obtained by successively integrating out momentum fluctuations at the scale $k$. The respective flow equation for $\Gamma_k$, the Wetterich equation [54–56], reads

$$\partial_t \Gamma_k[\phi] = \frac{1}{2}\text{Tr}\left[\frac{1}{\Gamma_k^{(2)} + R_k}\partial_t R_k\right],  \quad (4a)$$

where

$$\Gamma_{\phi_{i_1}\cdots\phi_{i_n}}^{(n)}[\phi](p_1, ..., p_n) = \frac{\delta\,\Gamma[\phi]}{\delta\phi_{i_1}(p_1)\cdots\delta\phi_{i_n}(p_n)}.  \quad (4b)$$

Equation (4) provides us with a full non-perturbative setup, that enables us to access the full UV-IR phase structure of the ASSM.

### B. Approximations of the quantum effective action

In general, (4) cannot be solved for the full effective action of an interacting theory, leaving aside the ASSM. Hence, we have to approximate the full effective action and its flows. In the present section, we summarise the approximations and the reliability considerations behind them, more details can be found in Appendices A and B.

To begin with, we aim at an accurate description of the full dynamics of the gauge and scalar sectors, and specifically that of the EW sector. Hence, the scale dependence (average momentum dependence) of all primitively divergent vertices in this sector is taken into account.

Within this setup, we can, for the first time, reproduce the EW transition $\mathrm{U(1)_Y \times SU(2)_L \to U(1)_{EM}}$ including all threshold effects, starting from the asymptotically safe initial conditions.

### 1. Momentum symmetric point approximation

Below the Planck scale, quantum gravity effects quickly decouple and we are left with the quasi-perturbative quantum dynamics of the SM. Its effects are captured well by only considering the average momentum running (with $k$) of the primitively diverging vertices of the SM. To that end we consider general vertices $\Gamma_k^{(n)}(p_1, \ldots, p_n)$ defined at symmetric points with

$$p_i^2 = p^2 \,, \qquad \left| \frac{p_i^\mu p_j^\mu}{p^2} \right| = \cos\theta \,. \qquad (5)$$

It has been shown both for strongly correlated QCD [57–60], and in gravity [14, 61, 62], that the system of flow equations of symmetric point vertices is well-approximated by only feeding back these vertices in the diagrams: effectively these vertices represent a close system of flow equations, and define a good approximation $\Gamma_k^{(SP)}[\phi]$ of the full effective action if only being interested in symmetric point physics. Note however, that it can be shown in QCD, that this approximation is limited to vertices and regimes without resonant momentum channels in vertices such as the scalar and pseudo-scalar meson channels in QCD for momentum and cutoff scales $k, p \lesssim 1\,\mathrm{GeV}$.

We also emphasise that $\Gamma_k^{(SP)}[\phi]$ does not suffice to give access to important physics information such as S-matrix elements for generic scattering momenta related to the full vertices $\Gamma^{(n)}(p_1, \ldots, p_n)$. However, the latter can be obtained in a systematic way from $\Gamma_k^{(SP)}[\phi]$ by their flows. Importantly, feeding back the full scattering vertices to the flow of $\Gamma_k^{(SP)}[\phi]$ leads to subleading effects.

In a final step, we introduce a further approximation to the symmetric-point vertices: they only depend on the symmetric-point momentum, the mass gaps $\vec{m}^2$ of the ASSM, and the cutoff scale $k$. For cutoff scales above the mass gaps, the (dimensionless) dressings can only depend on the ratio $p^2/k^2$ and we can trade one for the other. This property is also at the root of mass-independent RG-schemes, where (in asymptotic regimes) one can read off the momentum dependence of couplings from the RG-scale dependence. In turn, for cutoff scales below the mass threshold of specific fields, their contribution to the flows and the physics is suppressed, while the remaining flow still only depends on $p^2/k^2$. In consequence, for symmetric-point vertices, the cutoff dependence (at $p = 0$) reflects well the momentum dependence with

$$k = \alpha\, p \,, \qquad (6)$$

and hence supports a low-order derivative expansion with vertices $\Gamma_k^{(n)}(p = 0)$ with an approximate effective action $\Gamma_k^{(av)}[\phi]$ of these average vertices. As for the symmetric-point vertices, this expansion holds in the absence of resonant vertex structures or more generally strong momentum and angular dependences of vertices. It is well-tested by now both in the asymptotically safe UV regime of gravity-matter systems [10, 16, 61–64] and also holds true in the SM including QCD [57, 59, 65, 66]. We remark that within such an expansion the mass parameter extracted at $p = 0$ corresponds to the Euclidean curvature mass and not to the pole mass. However, it has been shown that in the absence of strong momentum dependences of the wave-function renormalisation or anomalous dimensions the two agree well, see [67]. This intricate topic is discussed further in [12–14].

### 2. Couplings of the ASSM

In our approximation, all primitively divergent vertices, as well as all wave functions of the SM, are taken into account with scale-dependent dressings. Moreover, we consider an effective Higgs potential $V_{\Phi,\mathrm{eff}}$ in a high-order Taylor expansion about the flowing minimum. In short, we parametrise

$$\Gamma_k^{(n)}(p_1, ..., p_n) = \prod_{i=1}^n \sqrt{Z_{\phi_i,k}}\, S^{(n)}(p_1, ..., p_n; \vec{\lambda}_k)\,, \quad (7a)$$

where $S^{(n)}$ are the $n$th derivatives of the classical ASSM action in (1), augmented with higher-order Higgs-self interactions. In (7a) the classical couplings are substituted by their running quantum analogues,

$$\vec{\lambda}_k = (\vec{g}_{1,k}\,, \vec{g}_{2,k}\,, \vec{g}_{3,k}\,, \vec{y}_{q,k}\,, \vec{y}_{l,k}\,, \vec{\lambda}_{\Phi,k}\,, \vec{G}_k\,, \vec{\Lambda}_k)\,. \quad (7b)$$

Here the vector on the coupling indicates that they represent a set of avatars of this coupling originating from different vertex functions, which will be explained below. The $g_i$ with $i = 1, 2, 3$ are related to the hypercharge gauge coupling with $g_1 \equiv \sqrt{5/3}\, g_Y$, the weak gauge coupling $g_2$, and the strong gauge coupling $g_3$. The classical dispersions are augmented with wave functions,

$$Z_{\phi,k} = (Z_{\mathcal{A},k}\,, Z_{\mathcal{C},k}\,, Z_{l,k}\,, Z_{q,k}\,, Z_{\Phi,k}\,, Z_{h,k}\,, Z_{c,k})\,. \qquad (7c)$$

The wave-function factors in (7a) encode the running of the attached fields, and the coupling parameters (7b) are indeed running couplings (and masses), and not only vertex factors. From here on we drop the subscript $k$ on all couplings and wave functions and their scale dependence is implied.

In the Higgs sector, we go significantly beyond the approximation in (7a), and consider the full Higgs potential within a high-order of a Taylor expansion about the

flowing minimum. We parametrise the Higgs doublet $\Phi$ as

$$\Phi = \frac{1}{\sqrt{2}} \begin{pmatrix} \mathcal{G}_1 + i\mathcal{G}_2 \\ v + H + i\mathcal{G}_3 \end{pmatrix}, \qquad \rho = \operatorname{tr} \Phi^\dagger \Phi, \quad (8)$$

where $v$ is the flowing minimum, and $H$ is the (radial) fluctuation Higgs field with a vanishing expectation value $\langle H \rangle = 0$, as the latter is explicitly carried by $v$. The $\mathcal{G}_i$ with $i = 1, 2, 3$ are the Goldstone modes. Then, the Higgs potential is a function of $\rho - v^2/2$. In the symmetric regime with a vanishing minimum of the potential, $v = 0$, we use the parametrisation

$$V_{\Phi,\text{eff}}(\rho) = V_0 + \sum_{n=1}^{N_{\max}} \lambda_{\Phi,2n} \, Z_\Phi^n \rho^n, \quad (9)$$

where $V_0 = \Lambda/(8\pi G_N)$ is related to the cosmological constant, see (A16). The running mass parameter $\mu_\Phi > 0$ and the quartic Higgs self-coupling, already present in the classical Higgs potential, are

$$\mu_\Phi = \lambda_{\Phi,2}, \qquad\qquad \lambda_\Phi = \lambda_{\Phi,4}, \quad (10)$$

and the expansion starts at $n = 1$ with the mass term.

In turn, in the broken regime with $v > 0$, we parametrise

$$V_{\Phi,\text{eff}}(\rho) = \tilde{V}_0 + \sum_{n=2}^{N_{\max}} \lambda_{\Phi,2n} \, Z_\Phi^n \left( \rho - \frac{v^2}{2} \right)^n, \quad (11)$$

where $\tilde{V}_0$ is $V_0$ minus the series, evaluated at $\rho = 0$. The lowest term in the series with $n = 2$ encodes the classical Higgs potential. At the *spontaneous symmetry breaking* cutoff scale $k_{\text{SSB}}$, below which EW symmetry breaking kicks in, the flow is switched from (9) for $k > k_{\text{SSB}}$ with a flowing $\mu_\Phi$ to (11) for $k < k_{\text{SSB}}$ with a flowing $v_k$.

In the following we will refer to both $\mu_\Phi$, $v^2$ as $\lambda_{\Phi,2}$, and hence the set of coupling parameters of the Higgs potential is always given by $\vec{\lambda}_\Phi = \{\lambda_{\Phi,2n}\}$ with $n = 1, ..., N_{\max}$. It is understood that $\lambda_{\Phi,2} = v^2$ is taken in the broken regime while $\lambda_{\Phi,2} = \mu_\Phi$ is taken in the symmetric regime. More details on the EW symmetry breaking and the respective scales can be found in Section IV B.

In the UV-regime we will consider a high-order Taylor expansion with the maximal monomial power

$$N_{\max} = 17. \quad (12)$$

This rather high-order of the Taylor expansion allows us to discuss the convergence of the potential at the UV fixed point. For the sake of simplicity, we do not consider such a general potential but the classical one for scales below $k \leq 10^{17}$. Such non-trivial potentials at the EW and metastability scales will be discussed in detail elsewhere.

The flows of the couplings and wave functions in the approximation are extracted at vanishing momentum, which captures the average momentum dependence as discussed in detail at the beginning of this section. For

the details of the projection onto the flow equations for all couplings, see Appendix B.

We close this section with a discussion of the gauge and gravity couplings. The set of coupling parameters (7b) comprises different avatars of gauge and gravity couplings as required for the flow of different gauge and gravity vertices. For gravity, we consider vertex couplings of the $n$-graviton scattering vertices proportional to the classical tensor structure,

$$\Gamma_{h\cdots h}^{(n)} \simeq Z_h^{n/2} S_{h\cdots h}^{(n)}(G_n, \Lambda_n). \quad (13)$$

Note that $G_n$ and $\Lambda_n$ are the vertex couplings of the classical tensor structure. While they should not be confused with the Newton coupling $G_N$ and cosmological constant $\Lambda$, they have the scale and momentum running of couplings. The different $G_n$ and $\Lambda_n$ are related by modified Slavnov-Taylor identities, see [14]. In the classical limit or rather classical regimes, the effective action reduces to the Einstein-Hilbert action, and all these couplings agree,

$$G_n = G_N, \qquad\qquad \Lambda_n = \Lambda. \quad (14)$$

Within our approximation, we use (14) throughout.

We also emphasise that the vertex gauge couplings defined in (7b) are subject to two-loop universality (in mass-independent RG schemes). However, in the presence of threshold effects and/or non-perturbative physics they differ genuinely. Specifically, we then have to consider avatars of the gauge couplings for different vertices. In the present approximation to the ASSM this concerns the quark-gluon couplings as well as the matter-gauge couplings in the EW sector. For example, the quark-gluon couplings are

$$\vec{g}_3 = (g_{3,u}, g_{3,d}, g_{3,s}, g_{3,c}, g_{3,b}, g_{3,t}), \quad (15)$$

where the field subscripts $q_i = u, d, s, c, b, t$ indicate the quarks in the respective quark-gluon vertex. We consider the strong isospin symmetric approximation with

$$g_{3,u} = g_{3,d}, \quad (16)$$

for all scales. Moreover, for momentum scales beyond the top threshold, all avatars converge towards a unique strong vertex coupling, $g_{3,q_i} = g_3$. In turn, in the presence of non-perturbative physics and threshold effects the two-loop universality of gauge couplings (in mass-independent RG schemes) does not hold anymore, and the matter-gauge couplings do not necessarily agree anymore, and also differ from the pure gauge couplings.

## C. Strongly coupled UV & IR regimes

In the IR regime for $k, p \lesssim 10\,\text{GeV}$, the approximation (7) has to be improved both in the gluonic sector for including confinement, as well as in the matter sector for including dynamical spontaneous chiral symmetry

breaking, see [66]. Here, the epithet *dynamical* refers to the fact, that the quasi-goldstone modes of strong spontaneous chiral symmetry breaking in QCD are effective IR degrees of freedom, the pions, and not a fundamental field such as the Higgs field $\Phi$ in the ASSM. In turn, for more quantitative access to the asymptotically safe regime, momentum dependences for the pure gravity sector are required, see [62]. The two asymptotic regimes are discussed below in Sections II C 1 and II C 2.

### 1. Confinement and strong chiral symmetry breaking

At momentum scales $p, k \lesssim 10\,\text{GeV}$, strongly correlated IR-QCD starts getting relevant and even two-loop perturbative approximations successively lack reliability, for a detailed discussion see [68]. However, in this regime, we can resort to results in functional approaches for 2 and 2+1 flavour computations that meet lattice 2 and 2+1 flavour benchmarks in the IR regime of physical QCD, see in particular [59, 66, 68]. Hence, below an *interface scale* $k_{\text{inter}}$ with

$$5\,\text{GeV} \lesssim k_{\text{inter}} \lesssim 15\,\text{GeV}\,, \tag{17}$$

we use IR-QCD flows as an external input in our system of ASSM flow equations: it has been shown in [68] that below $k \approx 10\,\text{GeV}$ two-loop resummed approximations successively lose their reliability, a -weak- limit being given by $k \approx 5\,\text{GeV}$. In turn, for $k \gtrsim k_{\text{inter}}$, the approximation to the ASSM flows discussed here suffices. Consequently, we use the stability of the results under a variation of $k_{\text{inter}}$ as a consistency and reliability check of the interface flows, see Figure 12.

Specifically, we utilise results from [66] as the approximation of the respective computation resembles most that used here, and the results can be used directly. This concerns the running of the quark masses including dynamical chiral symmetry breaking, as well as the running of the dressing of the gluon propagator, whose mass gap is related to the physical mass gap of QCD.

In terms of the correlation functions considered here, this amounts to using the gluon dressing $Z_A$ or rather the anomalous dimension

$$\eta_A = -\frac{\partial_t Z_A}{Z_A}\,, \tag{18}$$

from [66]. This implements the physical gapping of glue interactions for small momentum scales and suffices to describe the respective decoupling of the glue sector from the matter sector semi-quantitatively. It has been shown in [66, 69] that the anomalous dimension is well-described as a function of $\alpha_s$ and the mass gaps of the theory. This allows us to write the full anomalous dimension as

$$\eta_A^{(\text{ASSM})} \approx \eta_A^{(2+1)} \left( \vec{\alpha}_s^{(\text{ASSM})}, \vec{m}^2 \right) + \eta_A^{(c,b,t)}\,, \tag{19}$$

where $\vec{\alpha}_s^{(\text{ASSM})}$ is the vector of all strong fine structure constants derived from different vertices, see (B21) in

Appendix B 5. The vector $\vec{m}^2$ is the vector of all mass gaps in the QCD sector, see (B19). The second term on the right-hand side, $\eta_A^{(c,b,t)}$, comprises the contributions from the heavier quarks $c, b, t$. The relation (19) works quantitatively for mapping the anomalous dimension from pure glue to two-flavour QCD, and from two flavour QCD to 2+1 flavour QCD, see [66]. The approximation becomes better for larger $N_f$ and heavier quarks.

While the use of the relation in (19) is required for a quantitative description of the full IR dynamics of QCD, for the applications in the present work it suffices to use a semi-quantitative approximation derived in Appendix B 5, see (B27),

$$\partial_t Z_A^{(\text{ASSM})} \approx \partial_t Z_A^{(2+1)} - \eta_A^{(c,b,t)} Z_A^{(\text{ASSM})}\,. \tag{20}$$

where $Z_A^{(\text{ASSM})}$ is determined by a UV-IR consistency condition at the interface scale $k_{\text{inter}}$ in the range (17), see (B30). In Appendix B 5 it is also discussed, that the results show a small dependence on $k_{\text{inter}}$ within this range, see Figure 12.

Spontaneous chiral symmetry breaking requires the inclusion of the resonant scalar–pseudo-scalar four-quark interaction channel as in [66]. Effectively this leads to an additional mass contribution to the quark masses,

$$m_q = \frac{y_q\,v}{\sqrt{2}} \qquad \rightarrow \qquad \frac{y_q\,v}{\sqrt{2}} + \Delta M_q^{\text{const}}\,. \tag{21}$$

The term $\Delta M_q^{\text{const}}$ is taken from the results for the $u, d$ quarks in [66], computed in a strong isospin symmetric approximation: $y_u = y_d$. Then, the QCD contribution is given by the full physical constituent quark mass $M_q^{\text{const}}$ without the current quark contribution, $\Delta M_q^{\text{const}} = M_u^{\text{const}} - y_u v$, where $y_u v$ is evaluated at $k = 0$. We also use that the resonant interaction channel does not directly enter the flows of the other parameters considered in our approximation.

At large cutoff scales $k \gtrsim 10\,\text{GeV}$, the approximation used in [66] matches that used here, so the QCD part of the present ASSM investigation flows into strongly correlated QCD. In combination this allows us to flow the ASSM down to $k = 0$, including confinement and strong chiral symmetry breaking, leading to a semi-quantitative IR closure of the ASSM.

### 2. Asymptotically safe UV regime

In the asymptotically safe regime for $k \gtrsim M_{\text{Pl}}$, we utilise the results of [61, 62] where pure gravity flows with momentum dependences of propagators and vertices at a symmetric point have been considered. We emphasise that the momentum dependences of the vertices imply the inclusion of higher-order curvature invariants and also the difference between background and fluctuation correlation functions is resolved.

Here, we consider the momentum-dependent couplings

$$g_{h,n}(p) = G_{h,n} k^2\,, \qquad \lambda_{h,n} = \Lambda_n/k^2\,, \tag{22}$$

of the curvature tensor structures $(\sqrt{g}\,\mathcal{R})^{(n)}$ and the volume-form tensor structures $(\sqrt{g})^{(n)}$ respectively. They are part of the $n$-point vertices $\Gamma_{h\cdots h}^{(n)}$ of the fluctuation graviton $h$, and the present approximation is built upon the assumption of their dominance over other tensor structures in the vertices. While reminiscent of Newton coupling and the cosmological constant, these coupling parameters are vertex couplings and should not be confused with the former physical observables, see [14].

Moreover, for the fixed-point analysis, we take into account the momentum-dependent wave functions of the fluctuation graviton, $Z_h(p)$, and the gravity ghost, $Z_c(p)$.

We identify the avatars of the vertex couplings, see (14), and compute the flow of

$$Z_h\,,\,Z_c\,,\qquad g_h = g_{h,3}\,,\qquad \mu_h = -2\lambda_{h,2}\,,\qquad (23)$$

where all the avatars of the Newton coupling are identified with that of the three-point vertex. In [62, 70, 71], it has been shown that they are of similar size. The flow of $g_h$ and the anomalous dimensions are evaluated bilocally between $p = 0$ and $p = k$. The couplings of the volume form tensor structures are of sub-leading importance and we use $\lambda_{h,n\geq 3} \approx 0$. This approximation has been chosen due to their small fixed-point values for $n = 3, 4, 5$.

In [21, 23, 70, 71], it has been shown that matter-gravity systems admit an *effective universality* and a *close perturbative* behaviour. This entails that we can (approximately) identify the different gravity-matter vertex couplings of the $n$ matter fields $\Phi_{i_1}, ..., \Phi_{i_n}$ with $m$ gravitons $h$ with the uniform one in pure gravity, $g_h$,

$$g_{\Phi_1\cdots\Phi_i\,h\cdots h} = g_h\,,\qquad (24)$$

also used for the back coupling of matter to gravity. We only consider gravity-matter couplings that arise from the dispersion of the matter fields (kinetic and mass term), that is two identical or conjugate matter fields and one to three fluctuating gravitons.

In summary, this approximation is informed by the results on the momentum dependence of vertices and propagators in [62] and the effective universality and close perturbative behaviour of matter-gravity couplings. It hence reflects the state-of-the art approximations for matter-gravity systems in the literature, except for gravity-induced higher order couplings, see, e.g., [21, 28, 53, 72–79]. The respective extension will be considered elsewhere.

A final important novel ingredient is the non-trivial dimensionless Higgs potential $u$ considered in the fixed-point analysis. It derives from (9) by rescaling all dimensionful quantities by respective powers of the cutoff scale $k$. Restricting ourselves to the parametrisation of the Higgs potential in the symmetric regime, (9), we are led to

$$u(\bar{\rho}) = \frac{V_{\Phi,\text{eff}}}{k^4} = u_0 + \sum_{n=1}^{N_{\max}} \bar{\lambda}_{\Phi,2n}\,\bar{\rho}^n\,,\qquad (25a)$$

where $u_0 = \lambda_{h,0}/(8\pi g_h)$ with $\lambda_{h,0}$ as defined in (22), and $N_{\max} = 17$, see (12). The dimensionless radial Higgs field $\bar{\rho}$ and couplings $\bar{\lambda}_{2n}$ are given by

$$\bar{\rho} = Z_\Phi \frac{\text{tr}\,\Phi^\dagger\Phi}{k^2}\,,\qquad \bar{\lambda}_{\Phi,2n} = \frac{\lambda_{\Phi,2n}}{k^{4-2n}}\,.\qquad (25b)$$

In (25b), we have also absorbed the wave-function factors $Z_\Phi^n$ into the definition of $\bar{\rho}$, for more details see Appendix B 3. This allows us to map out the non-trivial UV phase structure of the ASSM with unstable potentials, trivial Gaußian potentials with one relevant direction, non-trivial flat potentials with two relevant directions, and stable as well as semi-stable potentials in Section V.

### D. Intermediate high energy regime

For momentum scales above $k_{\text{inter}} \approx 10\,\text{GeV}$, the fermionic matter sector hosts all three families of quarks and leptons. We use the fact that for cutoff scales $k \gtrsim 10\,\text{GeV}$ only the $t$ quark experiences a threshold effect due to its mass, while the other quarks, $q_l = d, u, s, c, b$, and all leptons $l$ can be assumed to be quasi-massless. Hence, for the sake of simplicity, we do not consider the running of the gauge couplings separately as they agree for $k \gtrsim 10\,\text{GeV}$, but identify all couplings with

$$g_{1,\phi_i} = g_{1,e}\,,\qquad g_{2,\phi_i} = g_{2,e}\,,\qquad g_{3,q} = g_{3,u}\,,\qquad (26)$$

The approximation (26) ensures that all fields have the required universal running above the respective thresholds, and settle at their correct IR value at $k = 0$. Below the mass threshold $m_{\phi_i}$ of a given field $\phi_i$, the running of its couplings is approximated with that of the lightest fields of the same type. This guarantees that the light fields have the correct running above the threshold, and is of sub-leading consequence for the dynamics triggered by the heavier field $\phi_i$: all diagrams with internal lines of the field $\phi_i$ are suppressed with powers of the running scale $k$ over its mass, $k^2/m_{\phi_i}^2 \to 0$. Consequently, these contributions are suppressed.

A similar approximation could be applied to the set of Yukawa couplings. We have refrained from using it here, as quantitative access to the top Yukawa coupling is chiefly important for an accurate determination of the top pole mass to the per cent level. In conclusion, we include the running of all Yukawa couplings separately.

### E. Wrap-up of the approximation

In summary, the approximation of the full effective action of the ASSM, is given by the classical one with running couplings and wave functions and a full effective Higgs potential within a high-order Taylor expansion in the trans-Planckian regime. In the asymptotic IR and UV regimes, we also take into account momentum-dependences for the gravity part in the asymptotic safety regime and for QCD in the strongly correlated IR regime. Then, the flow of the dynamical fluctuation parameters

is solved self-consistently, the running parameters are fed back into the flow and importantly, we consider all physical threshold effects. The flow solved here is RG-consistent [80, 81], and can be systematically improved.

## III. THE ASSM: EXISTENCE & STABILITY

A highly non-trivial aspect of general gravity-matter systems and the ASSM, in particular, is the reliability of predictions in the asymptotically safe regime. A necessary but not sufficient reliability criterion is the requirement that results are stable under changes of the regularisation and improvements of the approximation and also pass benchmark tests in simpler subsystems. It has been argued in [23] that the Reuter fixed point is always present for minimally coupled matter systems. Therefore, any approximation to the full system of flow equations that do not accommodate this property lacks full predictive power. In [23], this was illustrated in the example of Yang-Mills gravity system. It was shown that the presence of the required Reuter fixed point including asymptotically free Yang-Mills theory seemingly depends on the regulator choice. Moreover, the sign of the graviton contributions of the gluon anomalous dimension $\eta_a(p)$ depends on the value of the graviton mass parameter as well as the momentum $p$.

In the current work, we extend this analysis to the full ASSM and provide a reliability analysis of the UV regime. We are interested in the scenario where the marginal matter couplings run into the asymptotically free UV fixed points and hence pure matter contributions are subleading in the scaling regime around the fixed point. Then, the system shows competing effects between graviton tadpole diagrams and diagrams with more vertices. These competing effects originate in the inherent momentum dependences of the matter-graviton vertices and they complicate the determination of the sign of the $\beta$-functions due to the non-trivial momentum dependence. The latter also implies that a derivative expansion about vanishing momentum has to be done with care. For works including momentum dependences see [21, 23, 61–64, 70, 71, 82], and the recent review [10]. An expansion in powers of curvature invariants as is done in most applications of heat-kernel methods is tantamount to a derivative expansion at vanishing momenta, see [10]. Consequently, the same reliability arguments apply and to access the full momentum dependence requires the inclusion of invariants with full covariant momentum dependence, see [83–87]. Most heat-kernel computations additionally use the background-field approximation which is lifted in the present fluctuation approach.

Most expansion schemes at work in asymptotically safe gravity explicitly or implicitly use expansions about specific backgrounds, in most cases the flat background. An optimal choice for such a background would be the (minimal) solution of the equations of motion, leading to an *on-shell* expansion. Notably, the flat background, or

equivalently an expansion in powers of curvature invariants in the background field approximation, is an *off-shell* expansion. Off-shell expansion schemes may show an unphysical behaviour of the system at least in low orders of the expansion scheme if the off-shell expansion point is too far away from the on-shell background. A further intricacy is the fact, that commonly used expansions about a given background have a finite radius of convergence and one may see convergence towards an unphysical behaviour. Finally, the theory may not exist for all possible metric backgrounds. A respective analysis requires the evaluation of the fixed point and the flow for all backgrounds. In the fluctuation approach the computation of background-dependent vertices was initiated in [22, 88], for respective works in the background field approximation see e.g. [89–92].

In the present case, we do not expect the approximation to provide stable results for general background fields but only in the vicinity of the equations of motion (on-shell). Even though the present approximation together with the accompanying stability checks is qualitatively more elaborate than those used in previous works we expect instabilities for far off-shell configurations.

### A. Beta functions of marginal matter couplings

In the following, we restrict our analysis to the flow of the three-point couplings $\lambda_i = (g_1, g_2, g_3, y)_i$ in the vicinity of their Gaußian fixed point. The UV properties of the other avatars of the gauge couplings as well as the other Yukawa couplings follow similarly. We drop the subleading pure matter contributions, and we arrive at

$$\partial_t \lambda_i(\mathbf{p}) = \beta_{\lambda_i}^{\mathrm{grav}}(\mathbf{p}) \,, \tag{27a}$$

where $\mathbf{p} = (p_1, p_2)$ are two of the three independent four-momenta of a three-point function. The gravity contributions to these $\beta$-functions have been previously computed. For example, the gravity contribution to the gauge coupling was computed within the fRG in [23, 25, 50, 75, 93–95] and within perturbation theory in [49, 51, 96]. Here we go beyond previous fRG works since we derive this contribution for the first time from the gauge-quark vertex. The gravity contribution to the Yukawa coupling was computed in [17, 28, 53, 97, 98] and we are in agreement with the results from, e.g., [53]. The gravity contribution to the quartic scalar coupling was studied in [30, 32, 97–102]. The right-hand side of (27a) reads

$$\beta_{\lambda_i}^{\mathrm{grav}}(\mathbf{p}) = \eta_{\lambda_i}^{\mathrm{grav}}(\mathbf{p}) \, \lambda_i(\mathbf{p}) + F_{\lambda_i}^{\mathrm{grav}}(\mathbf{p}) \,, \tag{27b}$$

with

$$\eta_{\lambda_i}(\mathbf{p}) = \frac{1}{2} \sum_{j_i} \eta_{\phi_{j_i}}(p_{j_i}) \,. \tag{27c}$$

Equation (27c) comprises the sum of (1/2 of) the momentum-dependent anomalous dimensions of the fields

$\phi_{j_i}(p_{j_i})$ whose scattering is described in the vertex at hand, and $\eta_{\lambda_i}^{\mathrm{grav}}(\mathbf{p})$ is the matter-gravity part.

The second term in (27b), $F_{\lambda_i}^{\mathrm{grav}}$, stands for the diagrams of the vertex flow, see also Appendix E. We can distinguish two classes of diagrams, the class of matter graviton tadpoles / polarisation diagrams (tp), and genuine three-point function triangle diagrams (tri). The vertices in these diagrams depend on more momenta than the original couplings, for example, the two-graviton–two-quark–gluon vertex depends on four independent four-momenta. In the present approximation, these momentum dependences cannot be fed back properly. In the tadpole diagrams, the two graviton legs have the momenta $q$ and $-q$ and therefore we identify $\lambda_i(\mathbf{p}, q, -q) = \lambda_i(\mathbf{p})$, and similarly for the polarisation diagrams where always two matter legs have the momenta $p_1$ and $p_2$. This leads us to

$$F_{\lambda_i}^{\mathrm{grav}}(\mathbf{p}) = F_{\lambda_i=1}^{\mathrm{tp}}(\mathbf{p})\,\lambda_i(\mathbf{p}) + F_{\lambda_i}^{\mathrm{tri}}(\mathbf{p})\,, \qquad (28)$$

where the tadpole and polarisation diagrams only include the dressings $\lambda_i(\mathbf{p})$ as a prefactor in the involved graviton-matter vertices. In turn, the triangle diagrams include the genuine matter three-point vertices, and the loop momentum runs through the coupling, and hence this coupling cannot be pulled out in front of the diagram. In these diagrams, we can for example choose to identify the momenta of the coupling with the internal momenta $q_1 = q, q_2 = -p_1 - q$, leading to

$$\lambda_i(q_1, q_2)|_{\mathrm{tri}} \xrightarrow{\mathbf{p}\to 0} \lambda_i(q, -q)\,. \qquad (29)$$

This also entails that the coupling at an exceptional momentum with $p_1 = 0$ is relevant for this investigation, and not that at the symmetric point.

The coupling $\lambda_i(\mathbf{p})$ in (27) is the renormalisation group invariant dressing of the vertex at hand. In the flow equation approach, respective couplings are typically evaluated at a symmetric point: feeding back only this dressing in an average momentum approximation can be shown to lead to quantitative results in the absence of resonant interaction channels, for more details see [10, 12, 13]. Further interesting couplings are defined at exceptional momenta with soft momenta for one of the fields.

In order to investigate the scaling regime in the vicinity of the UV fixed point for $k \to \infty$, we evaluate (27) at a fixed momentum $\mathbf{p}$. The dimensionless quantities in (27) are functions of $\mathbf{p}/k \to 0$ and we can safely use this limit for all fixed momenta, and hence the right-hand side of (27a) reduces to $\beta_{\lambda_i}^{\mathrm{grav}} = \beta_{\lambda_i}^{\mathrm{grav}}(\mathbf{p} = 0)$, with

$$\beta_{\lambda_i}^{\mathrm{grav}} = \left(\eta_{\lambda_i}^{\mathrm{grav}} + F_{\lambda_i=1}^{\mathrm{tp}} + \gamma^{\mathrm{tri}} F_{\lambda_i=1}^{\mathrm{tri}}\right)\lambda_i\,, \qquad (30a)$$

where

$$F_{\lambda_i=1}^{\mathrm{tp/tri}} = F_{\lambda_i}^{\mathrm{tp/tri}}\Big|_{\lambda_i(\mathbf{p})=1}(0)\,, \quad \gamma^{\mathrm{tri}} = \frac{F_{\lambda_i}^{\mathrm{tri}}(0)}{\lambda_i\, F_{\lambda_i=1}^{\mathrm{tri}}}\,. \qquad (30b)$$

The first two terms in the parenthesis on the right-hand side of (30a) only involve the anomalous dimension at

vanishing momentum $\eta_{\lambda_i}^{\mathrm{grav}} = \eta_{\lambda_i}^{\mathrm{grav}}(0)$, the tadpole and polarisation diagrams $F_{\lambda_i=1}^{\mathrm{tp}}$ and the coupling $\lambda_i = \lambda_i(0)$ at vanishing momentum. In contradistinction, the coupling $\lambda_i(q_1, q_2) \to \lambda_i(q, -q)$ in the flow term $F_{\lambda_i}^{\mathrm{tri}}(0)$ is integrated over loop momenta $q^2$. Hence, it is sensitive to all momenta smaller than the cutoff scale as $q^2 \lesssim k^2$. This is taken into account with the prefactor $\gamma^{\mathrm{tri}}$ which is cutoff-independent in the scaling regime. This relative prefactor changes the balance between the triangle diagrams and the tadpole and polarisation diagrams. It may increase or suppress the contribution of the triangle diagram, subject to the momentum dependence of the coupling $\lambda_i(q, -q)$ in the vicinity of the fixed point. The integrand of the triangle diagram involves a factor $q^3$ from the radial momentum integration and a factor $q^4$ from the vertices, leading to the factor $(q/k)^7$ in the integrand. This suppresses very efficiently the small loop momentum regime of the vertex dressing $\lambda_i(q, -q)$ in comparison to the regime with $q^2 \approx k^2$ in the triangle diagram.

Equation (30a) is readily solved which leads us to

$$\lambda_{i,k_1} = \lambda_{i,k_2}\,\exp\left(\int_{k_2}^{k_1} \frac{\mathrm{d}k}{k}\left[\eta_{\lambda_i}^{\mathrm{grav}} + F_{\lambda_i}^{\mathrm{tp}} + \gamma^{\mathrm{tri}} F_{\lambda_i}^{\mathrm{tri}}\right]\right)\,, \qquad (31)$$

where both cutoff scales $k_1$ and $k_2$ are in the scaling regime. The present Gaußian fixed point scenario for the matter couplings $\lambda_i$ is sustained for

$$\eta_{\lambda_i}^{\mathrm{grav}} + F_{\lambda_i}^{\mathrm{tp}} + \gamma^{\mathrm{tri}} F_{\lambda_i}^{\mathrm{tri}} < 0\,. \qquad (32)$$

This concludes the derivation of the UV flows of marginal matter couplings in the ASSM.

## B. Variations of the regulator and cutoff scales

We are now in the position to access the stability of the Gaußian fixed point as well as the reliability of the results in the present approximation. A necessary condition is the independence of UV fixed-point nature from the chosen regulator. A given approximation typically only works for a given class of regulators. In turn, extreme choices of regulators potentially invalidate any approximations. In this context, we have to take into account that in the present approximation minimally coupled systems do not show the Reuter fixed point for all cutoff choices [23]. Therefore we only expect to see a stable Gaußian fixed point for specific choices of regulators.

We analyse the fixed-point scenario under specific variations of the regulators. For the sake of simplicity, we only consider relative changes of the cutoff scales $k_{\mathrm{grav}}$ for pure gravity and $k_{\mathrm{mat}}$ for matter fields. To that end, we introduce

$$k = \min(k_{\mathrm{grav}}, k_{\mathrm{mat}})\,, \qquad k_{\mathrm{mat}} = \gamma_{\mathrm{mg}} k_{\mathrm{grav}}\,. \qquad (33)$$

The standard choice is $\gamma_{\mathrm{mg}} = 1$, but the qualitative features of the system should be independent of it. Its

choice can be constrained by optimisation arguments in a given approximation: in the present approximation, we have dropped momentum dependences of the effective action. Then, optimal regulators are those that minimise the momentum transfer in diagrams as such a transfer (strong momentum dependences) cannot be captured by the present approximation that relies on $p^2 \approx k^2$ for all momenta. While this implies $\gamma_{\mathrm{mg}} = 1$ for different fields of the same kind, this is by far not obvious within a system that either contains both fermions and bosons or shows vastly different anomalous dimensions. By considering a variation of the relative cutoff scale between the matter and gravity sector, we provide a first exploratory study of the stability of the system.

Technically, we proceed as follows: the fixed point analysis is done in dimensionless variables. These variables are defined by multiplying the dimensionful ones with appropriate powers of the gravity cutoff scale $k$. This implies that for $\gamma_{\mathrm{mg}} < 1$ the only change in our system of flow equation originates in the matter shape functions and vice versa for $\gamma_{\mathrm{mg}} > 1$. For $\gamma_{\mathrm{mg}} < 1$, the matter shape functions are given by

$$r_{\mathrm{mat}}(x) = r_{\mathrm{flat}}\left(\frac{1}{\gamma_{\mathrm{mg}}^2}x\right), \qquad x = \frac{p^2}{k^2}, \qquad (34)$$

and

$$r_{\mathrm{flat}}(x) = \left(\frac{1}{x} - 1\right)\Theta(1 - x). \qquad (35)$$

In a rough first estimate, this change introduces relative factors $\gamma_{\mathrm{mg}}^4$ or even higher powers in the diagrams with a scale-derivative $\partial_t R_{\mathrm{mat}}$, if evaluated at vanishing external momentum. The power four is obtained for momentum-independent couplings, the higher power is obtained for momentum-dependent couplings, triggered by

$$\int_0^{\gamma_{\mathrm{mg}}} \mathrm{d}q\, q^3(q^2)^n = \frac{1}{4 + 2n}\gamma_{\mathrm{mg}}^{4+2n}, \qquad (36)$$

with the highest power $n = 2$ for the triangle diagrams in (30a). For this estimate we have neglected the subleading effect that the graviton and matter propagators have a non-trivial momentum dependence for $q^2 > 1$ and $q^2 > \gamma_{\mathrm{mg}}^2$. In summary, the main effect of the choice (34) is the suppression of diagrams with $\partial_t R_{\mathrm{mat}}$ ($\gamma_{\mathrm{mg}} < 1$) or $\partial_t R_{\mathrm{grav}}$ ($\gamma_{\mathrm{mg}} > 1$). If diagrams are contributing with different sings, this may introduce a change in the sign of given $\beta$-functions, entailing a qualitative change of the nature of the fixed point.

We note that such an enhancement or suppression can also be obtained by varying the graviton mass parameter $\mu_h$: large positive masses $\mu_h \to \infty$ suppress diagrams with $\partial_t R_{\mathrm{grav}}$ as they contain one further graviton propagator. In turn, for $\mu_h \to -1$, these diagrams are enhanced. As discussed in [103], an on-shell analysis requires a non-trivial metric solution which compensates effects of the mass parameter but also changes the vertices. This implies that the physical consequences of a suppression or

enhancement that originates in the value of the mass parameter, should be taken with a grain of salt: they are based on an off-shell background that can easily destroy the fixed-point properties. On-shell backgrounds with constant curvature were calculated in [22, 88] within the vertex expansion and the on-shell background curvature turned out to be $\mathcal{O}(1)$ in units of the cutoff scale. This suggests that a natural estimate for the on-shell analysis is given by small values of the graviton mass parameter.

## C. Stability of the Gaußian matter fixed point

We proceed with the stability analysis of the Gaußian matter fixed point. The pivotal part of the full system of flow equations is the matter-gravity part of the $\beta$-functions: for graviton couplings this part may destabilise the Reuter fixed point, leaving us with no UV closure. In turn, for matter couplings, the matter-gravity part is required for the existence of respective UV fixed points. Therefore the following discussion is restricted to the matter-gravity part as this is already conclusive.

Now we use that in the UV regime the matter-gravity parts of all matter-gauge couplings are identical as are that of all Yukawa couplings

$$\beta_{g_i}^{\mathrm{grav}} \to \beta_g^{\mathrm{grav}}, \qquad \beta_{y_{q,l}}^{\mathrm{grav}} \to \beta_y^{\mathrm{grav}}, \qquad (37)$$

and we can restrict our analysis to the pair $(\beta_g^{\mathrm{grav}}, \beta_y^{\mathrm{grav}})$. For the non-Abelian gauge couplings, the pure matter part of the $\beta$-function is negative, $\beta_{g_{2,3}}^{\mathrm{mat}} \propto g_{2,3}^3$, which entails asymptotic freedom with $g_{2,3} \to 0$. Thus, the gravity part $\beta_{g_{2,3}}^{\mathrm{grav}} \propto g_{2,3}$ dominates the asymptotically safe UV regime and is required to be negative for supporting the Gaußian fixed point with asymptotic freedom.

The pure matter parts of the U(1) coupling is positive $\beta_{g_1}^{\mathrm{mat}} > 0$ and hence the matter-gravity parts has to be negative, $\beta_{g_1}^{\mathrm{grav}} < 0$. In the Yukawa beta function, the Yukawa interaction contributes positively $\beta_y^{\mathrm{mat}} \propto y^3 > 0$ while the Yukawa-gauge interaction contributes negatively $\beta_y^{\mathrm{mat}} \propto y\, g^2 < 0$. For the Gaußian matter fixed point, the matter-gravity parts have to be negative, $\beta_y^{\mathrm{grav}} < 0$. In summary, in all cases negative matter-gravity parts are required, $\beta_g^{\mathrm{grav}}, \beta_y^{\mathrm{grav}} < 0$.

The analysis is facilitated further as we focus on the leading order contribution and neglect the anomalous dimensions stemming from regulator insertions in the diagrams. Then the dimensionless Newton coupling, $g_h$, is simply an amplitude factor in $\beta_g, \beta_y$. In the limits $\gamma_{\mathrm{mg}} \to 0$ and $\gamma_{\mathrm{mg}} \to \infty$, also the graviton mass parameter only changes the amplitude of the flow as all diagrams left have the same number of graviton propagators. For all other values of $\gamma_{\mathrm{mg}}$, we have a competition of diagrams with one and two graviton propagators and the value of the graviton mass parameter can change the sign of the contribution. For the analysis, we use $\mu_h = 0$. The results for other values $\mu_h$ can be inferred by a simple rescaling of $\gamma_{\mathrm{mg}}$ via $\gamma_{\mathrm{mg},\mu_h=0} = (1 + \mu_h)\gamma_{\mathrm{mg}}$. Accordingly, the fixed

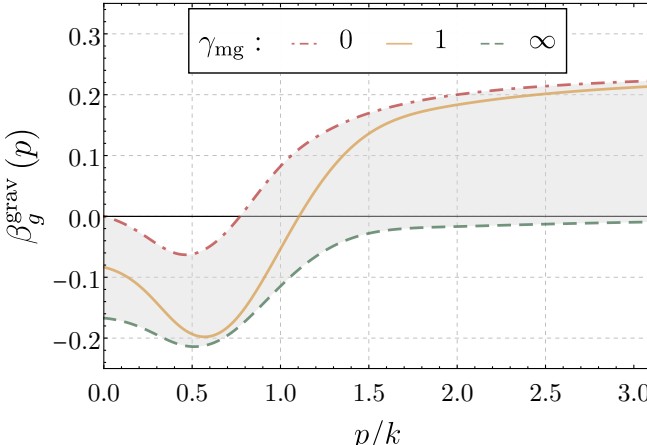

**Figure 1**. Matter-gravity contributions to the gauge beta functions $\beta_g^{\mathrm{grav}}$ derived from the gauge-fermion vertex for $g = 1$, $g_h = 1$, $\mu_h = 0$, and different values of $\gamma_{\mathrm{mg}}$. For $\gamma_{\mathrm{mg}} = 0$, $\beta_g^{\mathrm{grav}}$ vanishes due to an exact cancellation between the fermion anomalous dimension and the three-point diagrams.

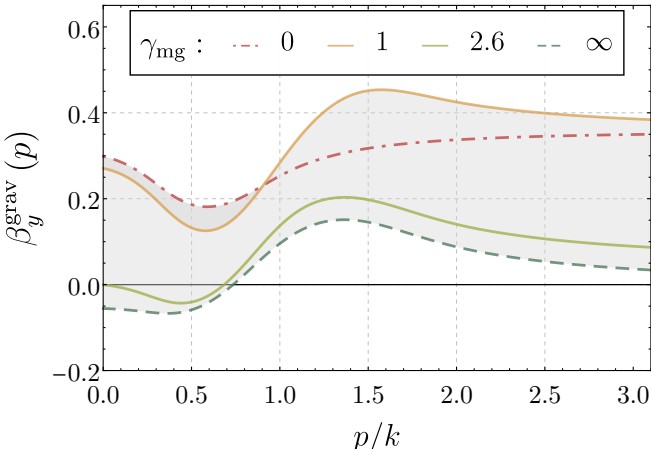

**Figure 2**. Matter-gravity contribution to the Yukawa beta function $\beta_y^{\mathrm{grav}}$ for $y = 1$, $g_h = 1$, $\mu_h = 0$, and different values of $\gamma_{\mathrm{mg}}$. For $\gamma_{\mathrm{mg}} \approx 1.6$, $\beta_y^{\mathrm{grav}}$ turns negative at $p/k \sim 0.5$, and for $\gamma_{\mathrm{mg}} \approx 2.55$ we observe a sign change at $p = 0$. Larger/smaller values of $g_h$ increase/decrease the amplitude of $\beta_y^{\mathrm{grav}}$.

point values of the pure gravity sector are irrelevant for the analysis.

We consider relative rescalings of matter and gravity cutoffs with $\gamma_{\mathrm{mg}}$ defined in (33) in the regime

$$\gamma_{\mathrm{mg}} \in [0, \infty), \tag{38}$$

which indirectly allows us to also access variations of the shape functions. As the present approximation does not allow for momentum transfers over large momentum scales, the limits $\gamma_{\mathrm{mg}} \to 0, \infty$ can only support the UV fixed point if matter fluctuations dominate the matter-gravity part ($\gamma_{\mathrm{mg}} \to 0$), or gravity fluctuations dominate the matter-gravity part ($\gamma_{\mathrm{mg}} \to \infty$). These are two physically different UV scenarios whose existence is accessed in these limits:

(i) For $\gamma_{\mathrm{mg}} \to 0$ discussed in Section III C 1, the matter sector is not IR regularised and quantum fluctuations of the matter fields are included at all scales. The matter propagators lack the IR mass introduced by the regulator and hence are enhanced relative to the graviton propagator. Therefore, this limit is a natural one for systems that are dominated by matter fluctuations (*matter matters* [15]).

(ii) For $\gamma_{\mathrm{mg}} \to \infty$ discussed in Section III C 2, the gravity sector is not IR regularised and quantum fluctuations of the graviton are included at all scales. The graviton propagator lacks the IR mass introduced by the regulator and hence is enhanced relative to the matter propagators. Therefore, this limit is a natural one for systems that are dominated by gravity fluctuations (*gravity rules* [16, 23]).

The results of [23] entail that minimally coupled gravity matter systems should show a gravity-dominated Reuter

fixed point apart from other fixed points. Evidently, an ASSM fixed point with a Gaußian or shifted Gaußian matter sector falls into the gravity-dominated class of fixed points, and we shall see in Section III C 2 that the ASSM fixed point with a Gaußian matter sector is supported in this limit, while the Gaußian matter fixed point is not present in the matter-dominated limit of the matter-gravity part as shown in Section III C 1. Note also, that this limit cannot work for the full system as it accesses the UV sector of the ASSM without UV fluctuations of gravity. This is simply the UV sector of the SM which is UV unstable.

This leaves us with the question in which regime for $\gamma_{\mathrm{mg}}$ we lose the ASSM fixed point. In Section III C 3 we show that this happens for $\gamma_{\mathrm{mg}} \approx 1$, which in our opinion is non-trivial support for the Gaußian fixed point scenario dominated by gravity fluctuations: stability only in one of the extreme limits $\gamma_{\mathrm{mg}} \to 0, \infty$, while it may show the correct physical behaviour also cast serious doubts on the approximation used. This is discussed in Section III D.

### 1. Matter matters: no matter cutoff with $\gamma_{mg} \to 0$

All diagrams with $\partial_t R_{\mathrm{mat}}$ drop out. This restates diffeomorphism and gauge consistency in the matter sector of the ASSM in terms of standard matter parts of the Slavnov-Taylor identities. Interestingly in this limit, the $\beta$-function of the minimal background gauge couplings vanishes as has been shown in [25]: the non-trivial cancellation of the graviton tadpole diagram and three-point function diagram in the graviton contribution to the background propagator is based on a kinematic identity rooted in diffeomorphism invariance of the gauge sector in this limit as well as the transversality of the kinetic term of the gauge field.

It is a highly non-trivial result of the present work, that this non-trivial kinematic identity at work in the flow for the gluon two-point function in [25] extends to the same result for the gauge-fermion coupling,

$$\beta_g^{\mathrm{grav}}(p = 0) = 0 \,, \qquad \text{for} \qquad \gamma_{\mathrm{mg}} = 0 \,, \qquad (39)$$

see also the *red dot-dashed* curve in Figure 1. As for the two-point functions of the gauge fields the identity (39) is rooted in diffeomorphism invariance of the gauge sector for $\gamma_{\mathrm{mg}} = 0$ and transversality of the gauge field.

We emphasise that it is the above combination of diffeomorphism and gauge symmetry leading to this non-trivial result. For the Yukawa couplings, this cancellation between tadpole, polarisation and triangle diagrams is not present and the graviton tadpole dominates. Its contribution to the $\beta$-function is positive and we are led to

$$\beta_y^{\mathrm{grav}} > 0 \,, \qquad \text{for} \qquad \gamma_{\mathrm{mg}} = 0 \,, \qquad (40)$$

see also the *red dot-dashed* curve in Figure 2.

In conclusion, the gauge-gravity part $\beta_g^{\mathrm{grav}}$, (39), supports the asymptotically free Gaußian fixed point of the non-Abelian gauge couplings $g_{2,3}$ and allows for a stable Gaußian as well as unstable interacting fixed point for the U(1) coupling $g_1$. Conversely, the matter-gravity part $\beta_y^{\mathrm{grav}}$, (40), is positive and no combined fixed point is present. One may speculate that the absence of the kinematic identity in the Yukawa coupling is a truncation artefact that originates in the lack of (modified) diffeomorphism invariance of the present approximation, also missing in other approximations used in the literature.

### 2. Gravity rules: no gravity cutoff with $\gamma_{mg} \to \infty$

All diagrams with $\partial_t R_{\mathrm{grav}}$ drop out. This reinstates diffeomorphism consistency in the pure gravity sector of the ASSM in terms of standard gravity parts of the Slavnov-Taylor identities. In this limit, the graviton tadpole is not present and the other diagrams lead to negative contributions to the $\beta$-function. In summary, we are led to

$$\beta_{g,y}^{\mathrm{grav}}(0) < 0 \,, \qquad \text{for} \qquad \gamma_{\mathrm{mg}} = \infty \,, \qquad (41)$$

see also the *green dashed* curves in Figures 1 and 2. This entails, that in the gravity-dominated UV scenario we encounter a Gaußian fixed point for all gauge and Yukawa couplings.

It is important to note that for $\gamma_{\mathrm{mg}} \neq 0$ the absence of the graviton tadpole can also be simulated by a sufficiently large positive graviton mass parameter $\mu_h \gg 0$ or a sufficiently negative cosmological constant $\Lambda \ll 0$ in the background field approximation. This limit suppresses the graviton propagator and leads to an *artificial* suppression of gravity: *on-shell* a large graviton mass parameter or large negative cosmological constant does not lead to a suppression of gravity fluctuations, and

already classically gravitons are massless in the presence of large curvatures. It is suggestive that this option is a mere truncation artefact, but this has to be investigated in better approximations.

As in the matter-dominated limit discussed in Section III C 1, a change of the graviton mass parameter only leads to a change of the amplitude of $\beta_{g,y}$, but does not change the sign. In short, the stability of the Gaußian fixed point in the gravity-dominated scenario persists off-shell.

### 3. Crossover regime at $\gamma_{mg} \approx 1$

So far we have analysed two limits $\gamma_{\mathrm{mg}} = 0, \infty$ and differences are expected between these extreme choices. In the gravity-dominated limit with $\gamma_{\mathrm{mg}} \to \infty$, the Reuter fixed point for the gravity couplings exists and *all* matter couplings have a stable Gaußian fixed point, while the Reuter fixed point is absent in the matter-dominated limit for $\gamma_{\mathrm{mg}} = 0$ and only *some* matter couplings have a stable Gaußian fixed point.

It is now decisive to investigate the crossover from $\beta_y > 0$ to $\beta_y < 0$. Remarkably, it takes place in the regime with $\gamma_{\mathrm{mg}} = \mathcal{O}(1)$: This hints at the fact, that the $\gamma_{\mathrm{mg}}$-dependence is caused by the low order of the approximation rather than being an artefact of an extreme choice of regulators.

For $\gamma_{\mathrm{mg}}$ larger than $\gamma_{\mathrm{mg}} \approx 1.6$ the $\beta$-function first dips below zero at momenta $p \approx k/2$, before it turns negative for $p = 0$ for $\gamma_{\mathrm{mg}} > \gamma_{\mathrm{mg}}^{\mathrm{stab}}$ with

$$\gamma_{\mathrm{mg}}^{\mathrm{stab}} \approx 2.55 \,, \qquad (42)$$

see Figure 2. The value $\gamma_{\mathrm{mg}}^{\mathrm{stab}}$ is very close to unity. This also entails that sign change may also be obtained by an appropriate change of the shape function of gravity and matter fields while keeping $\gamma_{\mathrm{mg}} = 1$. In our opinion, this gives a hint for the viability of the stable Gaußian fixed point scenario by the fact that the crossover between the two regimes takes place for $\gamma_{\mathrm{mg}} = \mathcal{O}(1)$.

### D. Existence of the ASSM

In short, the highly relevant question of the UV stability of the ASSM and its fixed-point properties cannot be answered conclusively in the present approximation. Future analyses should include a discussion of the on-shell or off-shell properties or instabilities. This entails, that we are in high demand for quantitative and qualitative upgrades, and a comprehensive fixed-point analysis should include all possible scenarios. In our opinion, despite the deficiencies of the present analysis, it still provides non-trivial hints for the existence of the ASSM with a stable Gaußian fixed point for all gauge and Yukawa couplings.

In the remainder of this work, we discuss the UV-IR and the fixed-point properties of the ASSM in the

present existence scenario. This entails a sub-leading behaviour of the graviton tadpole contributions to the flows of the matter couplings in the vicinity of the UV fixed point. Below the Planck scale, these contributions are also suppressed, and we can safely drop the respective diagrams. Then, the ASSM fixed point persists for all $\gamma_{\rm mg}$ and we choose $\gamma_{\rm mg} = 1$ for the sake of simplicity. This approximation can be summarised as

$$\gamma_{\rm mg} = 1 , \qquad \text{and} \qquad F_y^{\rm grav,tad} = F_g^{\rm grav,tad} \approx 0 , \quad (43)$$

i.e., we set the gravity-tadpole contributions to the running of the gauge and Yukawa couplings to zero.

## IV.  SUB-FERMI TO TRANS-PLANCKIAN ASSM

As a first application of the fRG set-up of the ASSM we compute its UV-IR phase structure. This includes the determination of the fundamental parameters by the respective experimental observables that give access to a comprehensive error analysis including both, experimental and systematic errors.

### A.   Fixing the coupling parameters of the ASSM

The physics trajectory of the ASSM is determined by matching its fundamental parameters to their experimental IR values. These observables have to be determined from $S$-matrix elements at $k = 0$ for the respective scattering momenta. We need to fix the following SM and gravity couplings

$$(g_1 , g_2 , g_3) , \quad (\vec{y}_q , \vec{y}_l) , \quad \vec{\lambda}_\Phi , \quad (g_h , \mu_h) , \quad (44)$$

where we have paired the parameters of the different sectors of the ASSM, the gauge couplings, Yukawa couplings, parameters of the Higgs potential, and the Newton coupling. All SM parameters except the strong coupling $g_3$ and the top Yukawa coupling $y_t$ are determined at $p = 0$ for the physical cutoff scale $k = 0$.

For the EW gauge couplings, we use the experimental values for $p \to 0$ to fix the electron avatars $g_1 = g_{1,e}$ and $g_2 = g_{2,e}$ at vanishing momentum

$$g_1 = 0.446 , \qquad g_2 = 0.618 . \quad (45a)$$

For the masses (except the top mass) we ignore the difference between the Euclidean curvature masses and the experimental pole masses, $M_{\phi_i}^{\rm (pole)}$. This leads us to

$$\lambda_{\Phi,4} = 0.129 , \qquad v = 246 \,{\rm GeV} , \quad (45b)$$

for the Higgs self-coupling and Higgs vacuum expectation

value, and

$$\begin{aligned} y_b &= 0.0240 , & y_c &= 0.00734 , \\ y_s &= 5.46 \cdot 10^{-4} , & y_u &= 1.32 \cdot 10^{-5} , \\ y_d &= 2.76 \cdot 10^{-5} , & y_\tau &= 0.0102 , \\ y_\mu &= 6.06 \cdot 10^{-4} , & y_e &= 2.93 \cdot 10^{-6} , \end{aligned} \quad (45c)$$

for the Yukawa couplings at $p = 0$ except for the top Yukawa coupling $y_t$. With the vacuum expectation value $v$ in (45b) and the Yukawa couplings in (45c), all curvature masses except the top mass match the experimental pole massed from [104].

The Newton coupling and cosmological constant are given by

$$G_{\rm N} = (1.22 \cdot 10^{19} \,{\rm GeV})^{-2} , \qquad \Lambda \approx 0 . \quad (45d)$$

It is left to determine the remaining two parameters, $y_t, g_3$. We first discuss the top Yukawa coupling $y_t$. While the identification of Euclidean curvature masses with the respective pole mass is a qualitatively reliable approximation for low-lying masses, see [67] for a case study, we cannot use it for the top quark mass or rather its Yukawa coupling $y_t$: the phase structure of the ASSM including its UV asymptotics is very sensitive to its precise value. The experimental pole mass from cross-section measurements is given by [104],

$$M_{t,\rm pole}^{\rm (exp)} = 172.5 \pm 0.7 \,{\rm GeV} , \quad (45e)$$

with a decay width of

$$\Gamma_{t,\rm pole}^{\rm (exp)} = 1.42^{+0.19}_{-0.15} \,{\rm GeV} . \quad (45f)$$

In Appendix F, we analytically compute the time-like momentum dependence of the top mass function $M_t(p)$ at vanishing cutoff scale, $k = 0$. The access to the analytic momentum dependence is rooted in the simple one-loop exact structure of the flow of the top quark propagator. The present determination includes full resummations of diagrams due to the iterative structure of the flow, and hence the result is very stable under further improvements of the approximation.

This allows us to determine the pole mass of the top as a function of the Euclidean mass parameter $m_t$, and for the experimental value of the pole mass, (45e), we arrive at the Euclidean mass parameter at vanishing cutoff scale, $k = 0$,

$$m_t = 165.4^{+0.9}_{-0.2} \,{\rm GeV} \quad \leftrightarrow \quad y_t = 0.950^{+0.005}_{-0.001} , \quad (45g)$$

see Appendix F. Having fixed the Euclidean mass parameter, the decay width is a prediction and we find

$$\Gamma_{t,\rm pole}^{\rm (theo)} = 1.72^{+0.09}_{-0.41} \,{\rm GeV} , \quad (45h)$$

in quantitative agreement with the experimental value (45f) from the PDG. Note, that the error in (45g) constitutes a systematic error estimate while the error in

(45h) only describes the relative weighting of QCD and non-QCD contributions, see Appendix F for details.

The error in $m_t$ in (45g) is dominated by the uncertainty of the value of the strong coupling $g_3$. To begin with, it cannot be determined at $k, p = 0$: In the deep IR, QCD is strongly correlated and the strong coupling cannot be defined by scattering experiments. For this reason experimental values are only present at and above the $\tau$-scale $p = 1.77\,\text{GeV}$, see [104]. Moreover, the definition of the strong coupling in the IR is subject to threshold effects and non-universality beyond two loops, see the discussion in Section II C 1. For this reason, we fix its value at the perturbative $Z$-scale. Here, its experimental value $(pp/p\bar{p})$ is given by [104]

$$
g_{3,k=M_Z}^{\overline{\text{MS}}} \approx 1.22 \quad \leftrightarrow \quad \bar{\alpha}_s := \frac{\left(g_{3,k=M_Z}^{\overline{\text{MS}}}\right)^2}{4\pi} \approx 0.118\,,
\tag{45i}
$$

matching the strong coupling in an $\overline{\text{MS}}$ or MOM-scheme. In the 2+1 flavour QCD computation in [66], it has been shown that the momentum-dependent (up-)quark-gluon coupling $g_3(p)$ at $k = 0$ agrees quantitatively with that at $g_{3,k}(p = 0)$ for $k = p$ for perturbative and semi-perturbative momenta, $k \gtrsim 4\,\text{GeV}$. This has been checked with lattice results as well as the momentum-dependent fRG and DSE results from [59, 68] and relates to the logarithmic running of the (up-)quark-gluon coupling in this regime far from the up-quark threshold.

It has been shown in [68], that the MOM$^2$ RG-scheme, used in the fRG, leads to an enhancement of the avatars of the running gauge coupling $\alpha_s(p)$. In pure Yang-Mills, the respective enhancement factor for $\alpha_s$ is approximately 4/3, [65], dropping to an enhancement factor of approximately 1.2 in the 2+1 flavour case (measured in the regime between 10-40 GeV). A linear extrapolation from these two values can be done for the ASSM quark content leading to a scaling factor of 1.07. This analysis suggests identifying

$$
g_{3,k=M_Z} = \sqrt{1.07\,(4\pi\bar{\alpha}_s)} = 1.26\,.
\tag{45j}
$$

However, being short of a full analysis, we show results within an estimate of the systematic error of the choice in (45j) with

$$
\alpha_{s,k=M_Z} \in \left[\bar{\alpha}_s,\, 1.10\,\bar{\alpha}_s\right]\,,
\tag{46}
$$

see also (E5) and the detailed discussion in Appendix E 3. There and in Appendix F further estimates and consequences are discussed.

In particular, the above scale setting of the strong coupling, and its uncertainty, has a sizeable impact on the top Yukawa coupling and therefore in the high energy development of the ASSM flows. For example, the seemingly small uncertainty of the strong coupling has a huge effect on the metastability scale $k_{\text{meta}}$ at which the quartic Higgs-self coupling turns negative. We discuss this point in detail and perform a systematic error analysis in Appendices E 3 and F.

## B. The ASSM trajectory

The scale-dependence of the ASSM parameters is depicted in Figure 3. We span momentum scales from $k \to 0$ and $p = 0$ in the deep IR with chiral symmetry breaking and confinement to $k \to \infty$ far beyond the Planck scale in the asymptotically safe UV fixed-point regime. While the cutoff-scale dependence is not directly a physical momentum dependence, it relates to the physical momentum dependence at the symmetric point. For a detailed discussion see Section VI and [12–14].

For cutoff scales below roughly 1 TeV, the ASSM enters the symmetry broken phase, as the effective potential of the Higgs field, $V_{\Phi,\text{eff}}$, develops a non-trivial minimum,

$$
\mu_\Phi(k < k_{\text{SSB}}) < 0\,, \quad \text{with} \quad k_{\text{SSB}} = 940\,\text{GeV}\,.
\tag{47}
$$

The non-trivial background of the Higgs renders all SM fields except photons and gluons massive. Consequently, this leads to a decoupling of the fluctuations of the respective fields below their mass thresholds, clearly visible in Figures 3 and 4.

The Euclidean masses of the SM fields read,

$$
m_{W^\pm} = \frac{g_2\,v}{2}\,, \quad m_Z = \frac{g_2\,v}{2\cos(\theta_W)}\,, \quad m_H = \sqrt{2\,\lambda_{\Phi,4}}\,v\,,
\tag{48}
$$

for the EW and Higgs bosons. The quark/lepton fields with/without the constituent mass contribution are given in (21). The scale dependence of all Euclidean particle masses is shown in Figure 4. From this Figure, the intricacy of the particle masses can be understood. These evolutions are important for the correct determination of observables at colliders, for example, the weak mixing angle is especially sensitive to those.

Finally, dynamical strong chiral symmetry breaking has a sizeable impact on the (constituent) quark mass function (21), in particular for the light flavours $u, d, s$. The respective $u$-quark-gluon exchange coupling, defined in (B31), is depicted in Figure 12. It decays for cutoff scales below $\sim 1\,\text{GeV}$ as a consequence of the QCD mass gap. Note that the avatars of the quark-gluon exchange couplings for the different quarks vary slightly due to the different quark wave functions, see (B21). Moreover, the strong exchange couplings defined in (B21) carry the physical threshold effect of the respective gluon exchange. This threshold is not present in the definition of the electroweak gauge couplings: the wave functions of $W^\pm, Z^0$ are the prefactors of the kinetic terms, while the masses originate from the interaction of the electroweak gauge bosons with the Higgs. These couplings are also called effective charges, e.g. [105, 106]. A related definition in QCD, that does not show the threshold of the mass gap in QCD, is given by the process-independent coupling or effective charge, see e.g. [107, 108]. In turn, the respective electroweak exchange couplings with the physical mass thresholds enter directly the ASSM flows, as do the quark-gluon exchange couplings, and directly show

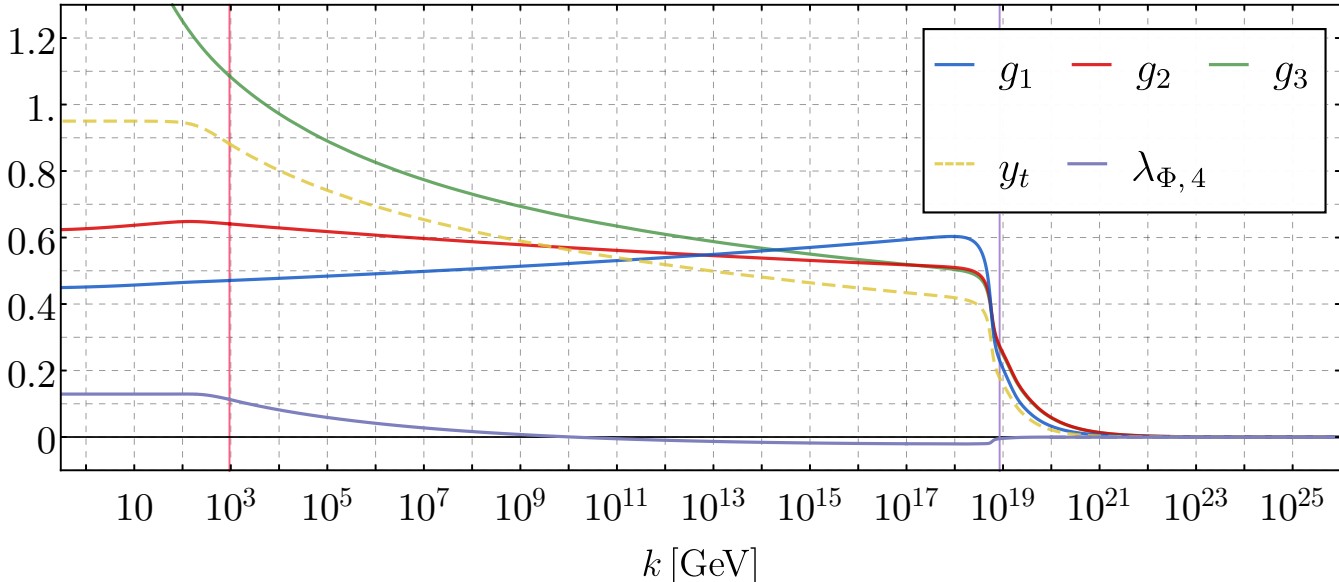

**Figure 3**. Scale dependence of the gauge couplings, the top Yukawa coupling and the Higgs self-coupling in the ASSM. We have omitted here the flow of the remaining Yukawas and the flow of the curvature mass and vacuum expectation value. In the UV regime beyond the Planck scale (indicated by a *violet* vertical line), all matter couplings approach the Gaußian fixed point, including a flat Higgs potential with two relevant directions. Below the $k_{\mathrm{SSB}}$ scale (indicated by a *red* vertical line), the top threshold is clearly visible in the $y_t$, $\lambda_{\Phi,4}$ and $g_2$ flows. Moreover, for the latter two the Higgs, $W^\pm$ and $Z^0$ bosons decoupling also play a quantitative role. For scales above $k = 10^{17}$ GeV the flow of $\lambda_{\Phi,4}$ is considered within the full Taylor expanded potential.

the decoupling of fluctuations below the mass thresholds. Their definition is provided in (B31) in Appendix B 5. For an illustration of the threshold effect and their similarity with the quark-gluon exchange couplings, they are also depicted there in Figure 12.

The exchange gauge-fermion couplings show the decoupling of fluctuations in the flows of the fermion mass terms due to the mass gaps of the gauge bosons. A second source for the decoupling of fluctuations is given by the mass thresholds of the fermions themselves. Finally, the electroweak fluctuations are suppressed due to the small electroweak couplings. All these threshold effects and the dominant nature of the strong fluctuations below the electroweak scale are visible in Figure 4, where we show the flows of the ASSM fermion masses below $k \leq k_{\mathrm{SSB}} = 940$ GeV, see (47). This regime covers all matter thresholds in the ASSM.

For $k \to k_{\mathrm{SSB}}$ from below, the Higgs expectation value $v$ drops sharply as a function of the cutoff scale, and all fields become massless. We note in passing, that while this is reminiscent of a second order phase transition, it should not be confused with it. It is a standard reflection of spontaneous symmetry breaking in the RG flow in the absence of an explicit one. In particular, it does not carry directly the order of the thermal electroweak phase transition.

Above the EW cutoff scale, $k > k_{\mathrm{SSB}}$, all fields are effectively massless and the $\beta$-functions and anomalous dimensions of all marginal (logarithmically running) pa-

rameters reduce to their universal form, and agree with that computed with perturbative methods, as do the respective trajectories, see e.g. [109]. In turn, the flow of the dimensionful Higgs mass parameter shows the expected quadratic running and hence is subject to quadratic fine-tuning in the UV. We emphasise that this is technically challenging but carries no conceptual intricacy.

At sufficiently large cutoff scales the quartic Higgs self-coupling changes sign and turns negative, see Figure 3. This indicates a metastable regime or the potential importance of higher-order couplings. In the present ASSM setup, this happens at

$$k_{\mathrm{meta}} = 1.2 \cdot 10^{10}\,\mathrm{GeV}\,. \tag{49}$$

This value compares well with the values obtained in perturbative computations within the $\overline{\mathrm{MS}}$-scheme: As discussed before, a direct translation of the RG scales is not straightforward, as is their relation to momentum scales. However, a conservative estimate of these uncertainties is well below an order of magnitude, and hence we identify RG scales. This entails that (49) is well compatible with

$$10^9\,\mathrm{GeV} \lesssim k_{\mathrm{meta}} \lesssim 10^{11}\,\mathrm{GeV}\,, \tag{50}$$

in the $\overline{\mathrm{MS}}$-scheme, see e.g. [109]. A conclusive analysis requires the inclusion of higher-order couplings in this regime [40–44, 46]. Importantly, the metastability scale $k_{\mathrm{meta}}$ is very sensitive to the size of the Yukawa coupling and hence the determination of the pole mass of the top

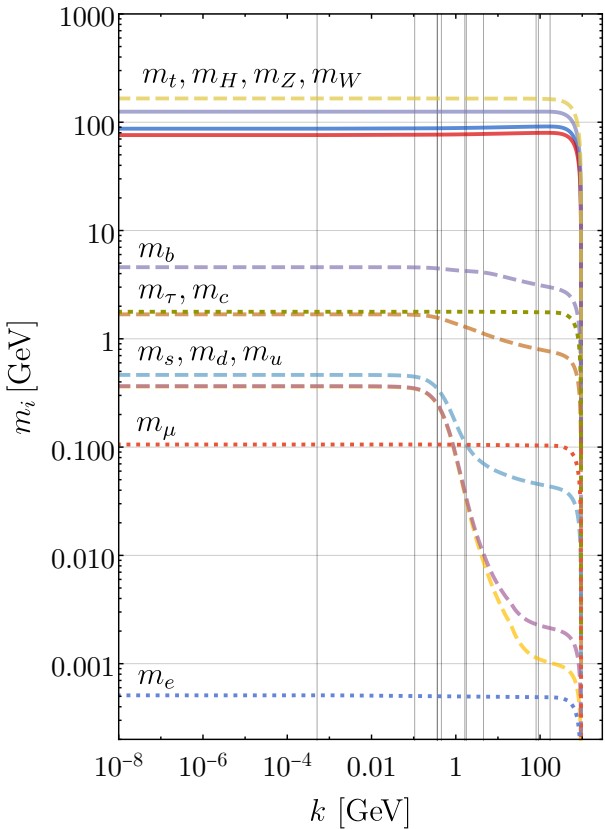

**Figure 4**. Flow of all SM Euclidean masses. Plain lines correspond to the EW and Higgs bosons, dashed to quarks and dotted to the leptons. The different particle decoupling scales are indicated by vertical lines. For quarks, the constituent masses as defined in (21) and obtained from [66] are also accounted. At scales around the $k_{\text{SSB}}$ the vacuum expectation value of the Higgs drops and all masses vanish.

quark, discussed in detail in Appendix F. There it is shown that the largest systematic error in the determination of the pole mass can be attributed to the uncertainty in the determination of the running QCD coupling $g_{3,k=M_Z}$, that stems from the requirement of a self-consistent RG scheme for all scales.

In the present work, we use the estimate (45j) with a relative factor 1.07 in comparison to the $\overline{\text{MS}}$-value. This is a linear extrapolation of the Yang-Mills and 2+1 flavour to the ASSM quark content with the error estimate (46), see also the discussion in Appendix E 3. Interestingly, for $\alpha_{s,k=M_Z}$ being roughly 15% larger than the $\overline{\text{MS}}$-value, the metastability scale approaches the Planck scale $M_{\text{pl}}^2 = G_{\text{N}}^{-1}$. For the strong coupling $g_{3,k=M_Z}$ this entails a 7% enhancement,

$$g_{3,k=M_Z} = 1.31, \qquad k_{\text{meta}} = M_{\text{pl}}. \qquad (51)$$

An analysis of the respective systematic error is deferred to Appendix E 3, where we also show $k_{\text{meta}}$ as a function of the strong gauge coupling $\alpha_{s,k=M_Z}$, see Figure 16.

Above the Planck scale, the ASSM quickly gets domi-

nated by gravity fluctuations, as is the ASSM UV fixed point. This has been discussed in particular in [16, 23]. Moreover, matter contributions of the fixed point equations of the pure gravity coupling and mass parameters $g_h$ and $\mu_h$ are simply closed matter loops, and the external gravitons couple via avatars of the Newton coupling. Consequently, the fixed point equations for these couplings are closed and only depend on the number of matter fields of a given kind. In the ASSM, this leads us to the gravitational fixed point

$$g_h^* = 0.147, \qquad \mu_h^* = -0.656, \qquad (52)$$

with the cosmological constant avatars $\vec{\lambda}_{h,n}^* \approx 0$ for $n \geq 3$. With the fixed point values of the pure gravity parameters at hand, we can discuss the qualitative properties of the trans-Planckian asymptotic safety regime. For the physical trajectory, quantum gravity effects grow large at about the Planck scale, and the flow is dominated by gravity fluctuations, see Figure 11 and [16, 23].

In the regime dominated by asymptotically safe quantum gravity, the gauge and Yukawa couplings are attracted towards the Gaußian fixed point. This is straightforward for the non-Abelian gauge couplings since they are asymptotically free without gravity and gravity is always supporting asymptotic freedom [23, 25]. For the Abelian hypercharge coupling, the gravity fluctuation must exceed the matter fluctuation that drives the hypercharge coupling into a Landau pole, which indeed happens here.

The UV Higgs sector requires special care, and it is discussed in detail in Section V. Depending on the pure gravity fixed point values (52) it has different stability and relevance properties. For the values computed in the present approximation, (52), we are left with a peculiar situation, which to our knowledge has not been discussed before: for increasing $N_{\text{max}}$ we see a convergence of the full potential to a flat one for fields within the validity regime of the Taylor expansion, see Figure 5. For each $N_{\text{max}}$ we have two relevant parameters whose Eigenvectors have maximal overlap with $\mu_\Phi$ and $\lambda_{\Phi,\text{max}}$. This is reminiscent of a standard Gaußian fixed point, but at the latter only $\mu_\Phi$ is relevant while all $\lambda_{\Phi,2n>2}$ are irrelevant. In Figure 3 we show $\lambda_{\Phi,4}$, which is obtained from the UV-IR flow of the potential (11) within the high (converged) order of the Taylor expansion with $N_{\text{max}} = 17$, see (12). This flow is initiated in the vicinity of the UV fixed point and the full system is flown down to cutoff scales below the Planck scale: $k \geq 10^{17}$ GeV. Below the Planck scale, gravity fluctuations and the higher-order ($n \geq 3$) Higgs self-couplings decouple rapidly. We have checked this also numerically, and $k = 10^{17}$ GeV is more than one order of magnitude below the decoupling regime. Hence the higher-order couplings can safely be dropped for $k < 10^{17}$ GeV, which is done here for reducing the numerical effort. Matter effects on the expanded potential at cutoff scales close to the EW scale have already been studied [42, 45], and their investigation in the current framework will be done elsewhere. In this work, for cutoff scales $k \leq 10^{17}$ GeV, we continue within the $\Phi^4$-approximation.

In summary, at the ASSM UV fixed point, we have a flat Higgs potential

$$\vec{\lambda}_\Phi^* = 0 \,, \tag{53}$$

and in particular the coupling parameters $\mu_\Phi^*$ and $\lambda_\Phi^*$ of the classical Higgs potential, (A8). Moreover, the ASSM has as many relevant parameters in the matter sector as we have in the SM, and an additional three relevant couplings in the gravity sector, that overlap with $G_N$, $\mu_h$, $\Lambda_3$. This concludes our discussion of the physical trajectory of the ASSM. For the first time, we connect the asymptotically safe UV regime of the full SM coupled to asymptotically safe gravity to the deep IR regime of the SM with IR-QCD with confinement and dynamical strong chiral symmetry breaking.

## V. FIXED-POINT HIGGS POTENTIAL AND THE PROPERTIES OF THE ASSM FIXED POINT

In this section, we discuss the emergence of the full fixed-point potential of the Higgs field within a high-order Taylor expansion, see (25). This enables us to discuss the fixed-point properties of the ASSM, and in particular the number of relevant parameters and their stability. Moreover, we discuss the location of the ASSM fixed point in the fixed-point landscape, including a discussion of the reliability and a systematic error of the respective result.

In particular, the analysis in this section reveals a novel non-trivial fixed point of the ASSM that has not been seen before: the Taylor expansion of the Higgs fixed point potential $u^*(\bar{\rho})$ converges to a flat potential $u^*(\rho) \equiv 0$ with *two* relevant directions in contradistinction to a standard Gaußian fixed point with *one* relevant direction, see Figures 5 and 6. Such a potential cannot be identified within a low order of the Taylor expansion, and certainly, it cannot be seen within the standard $\Phi^4$ approximation. Moreover, the two relevant directions cannot be revealed in a global analysis of such a fixed point without a full stability analysis including the computation of the relevant eigenperturbations, see Figure 7. In short, it is easily overlooked. Interestingly, a rather similar fixed-point pattern has been seen very recently for an effective potential $u((\partial_\mu \varphi)^2)$ in a shift-symmetric setting, see [110]. While there the novel fixed point has been attributed to shift symmetry, in the present case the flat non-trivial fixed point is revealed for the ASSM with its explicit breaking of shift symmetry.

Besides being a novel fixed point, its relevance comes from the fact that in the current approximation this fixed point is indeed connected with the physical SM (with the parameters fixed by experimental observables) in the IR. In turn, the standard Gaußian fixed point is obtained with a vanishing coupling for the most relevant (non-polynomial) operator. However, it is connected to the physical SM within the present approximation: the resulting Higgs and top mass are roughly $2.9\,\text{GeV}$ from the

central experimental values. Whether this distance is close or feasible is subject to the evaluation and interpretation of the systematic error of the present approximation. This is but one of the reasons for the rather detailed description of the systematic error estimates in the present work.

We emphasise in this context that the stability aspects of such an analysis are only fully conclusive if the following three aspects are taken into account:

First, within an Einstein-Hilbert truncation, the gravity-matter fixed point with minimally coupled matter can be mapped to the pure gravity fixed point, see [23]. Any approximation that shows instabilities in this approximation, lacks credibility also in improved ones. It is the $\mathcal{R}^2$, $\mathcal{R}^2_{\mu\nu}$ and $\mathcal{C}^2_{\mu\nu\rho\sigma}$ tensor structures, that can trigger instabilities or qualitative changes similar to the Bank-Zaks in QCD as also argued in [23].

Secondly, a full Higgs potential is required. In fact, the $\Phi^4$ truncation of the Higgs potential shows an unstable fixed point potential with $\mu_\Phi^* = 0.286$ and $\lambda_{\Phi,4}^* = -10.8$, which is qualitatively different from the solution it converges to, as we will show in this section.

Finally, one has to include gravity-induced matter interactions as discussed in [21, 28, 53, 75] that are relevant for the discussion of a potential weak gravity bound, see [75, 79, 111].

The current work takes the two first aspects partially into account: we consider a high-order Taylor expansion of the Higgs potential, and the gravity vertices used contain higher order momentum dependences that are generated by the higher order curvature invariants, for a respective discussion, see [14, 62], where it is argued that the $\mathcal{R}^2_{\mu\nu}$ is approximately absent at the fixed point.

### A. Higgs fixed-point potential

In this section, we discuss the Higgs fixed-point potential as obtained within a high-order Taylor expansion. We will see that the potential rapidly converges towards a flat potential with two relevant directions. This analysis is augmented with a resolution of the eigenperturbation, the respective linear differential equation being of the Sturm-Liouville type. Hence, both the eigenperturbation as well as the fixed-point potential take the form of Kummer functions, and the coefficient of the fixed-point potential vanishes.

#### 1. Taylor expansion of the fixed-point potential

We use a high-order Taylor expansion of the dimensionless Higgs potential $u(\bar{\rho})$ about vanishing field, see (25). At the UV fixed point the other SM couplings are vanishing, see Figure 3, and the gravity couplings take their fixed-point values, see (52). The fixed-point equation for the Higgs potential $u(\bar{\rho})$ reads

$$\text{FP}_u(u^*) = 0\,, \quad \text{with} \quad \text{FP}_u(u) = \mathcal{D}\, u - \text{Flow}_u\,, \tag{54a}$$

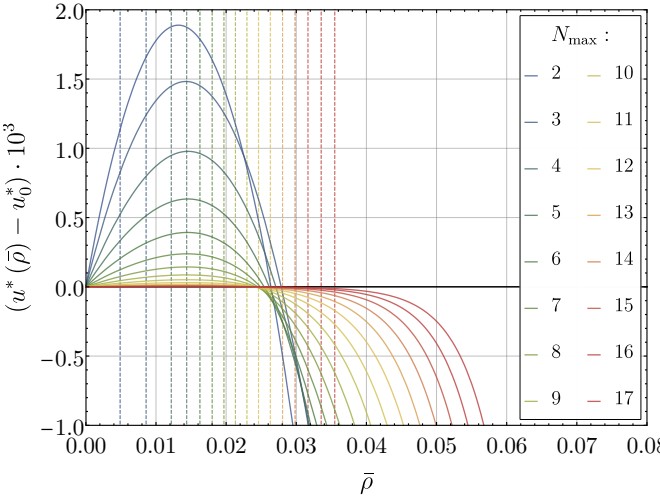
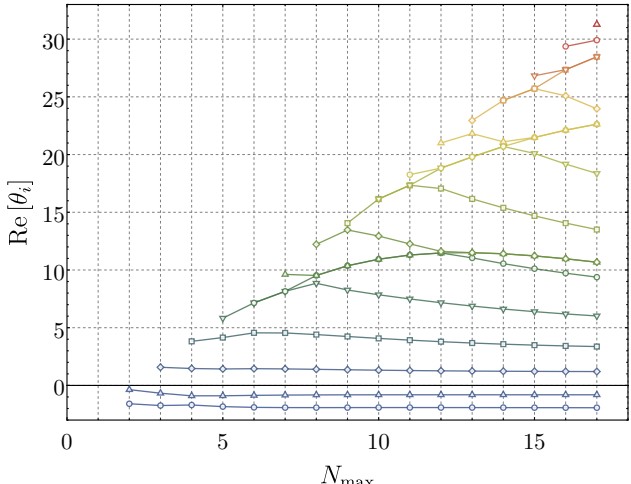

**Figure 5**. ASSM fixed point Higgs potential (left) and its critical exponents (right). The Higgs potential converges towards a flat potential within the reliability regime, which is indicated by the vertical lines as defined in (55). In contradistinction to a standard Gaußian Higgs potential with one relevant direction, we find two relevant directions related to the two negative eigenvalues (right) for all approximations, and the eigendirections have strong overlap with $\lambda_{\Phi,2}$ (*blue triangle*) and $\lambda_{\Phi,2N_{\max}}$ (*blue circle*), see Figure 6. The values rapidly converge with increasing $N_{\max}$, see (56). This rapid convergence is also seen for the lower lying positive eigenvalues, while the convergence of the large ones would require larger $N_{\max}$. The parameter space where such a non-trivial flat potential exist is indicated as the green regime in Figure 9, which includes the ASSM fixed point.

where $\mathrm{Flow}_u[u(\bar{\rho}),\bar{\rho}]$ comprises the flow diagrams for $V_{\Phi,\mathrm{eff}}$ divided by $k^4$, and subtracted at $\bar{\rho}=0$. This subtraction leads to $u(0)=0$ and eliminates the overlap of the Higgs potential with the cosmological constant. The scaling part is given by $\mathcal{D}u$ and reads

$$\mathcal{D}u = \left[4 - (2+\eta_\Phi)\,\bar{\rho}\,\partial_{\bar{\rho}}\right]u(\bar{\rho})\,. \qquad (54b)$$

The operator $\mathcal{D}$ generates full scalings on the potential $u$ including the anomalous part, and hence $\mathcal{D}u$ comprises the scaling parts of the fixed-point equation. Note that the part 4 entails the canonical dimension of the potential, and in general we have $4 \to d_\varphi$ for $\mathcal{D}\varphi$, where $d_\varphi$ is the full scaling dimension of the function $\varphi(\bar{\rho})$.

As the Yukawa and gauge couplings vanish at the fixed point, $\mathrm{Flow}_u$ only receives contributions from Higgs and graviton loops. In the present work, we use a high-order Taylor expansion of the potential. While this converges rather quickly for small $\bar{\rho}$, we have to carefully evaluate the reliability regime and the radius of convergence of the expansion. Accordingly we define the reliability regime by $\bar{\rho} \in [0,\bar{\rho}_{\max}]$ with $\Delta u = u - u_0$,

$$\bar{\rho}_{\max} = \max_{\bar{\rho}}\left\{\bar{\rho}\;\middle|\;\frac{|\mathrm{FP}_u(\Delta u^*)|}{\sqrt{(\mathcal{D}\,\Delta u^*)^2 + \mathrm{Flow}_u^2(\Delta u^*)}} \le 1\%\right\}, \qquad (55)$$

i.e., we aim at a relative accuracy of 1% in our fixed-point search. Furthermore, we only take into account potential fixed-point solutions for which the lower critical exponents display convergence, in particular the relevant (negative) ones. Naturally, the higher ones are less stable as they

are sensitive to the dropped even higher-order couplings $\Lambda_{\Phi,n}$ with $n > N_{\max}$. Fixed-point potentials, that do not satisfy these criteria are not considered.

At the ASSM fixed point, we find one non-trivial fixed-point solution, displayed in Figure 5 for $N_{\max} = 2,...,17$. We observe that the solution converges towards a flat potential within the reliability regime of the Taylor expansion and only outside of the regime we encounter an instability. Moreover, the reliability regime is growing with each order of the expansion. This is observed in the $N_{\max}$-dependence of the ratio of the highest-order coupling and the second-highest one, $r_u(N_{\max}) = |\lambda_{\Phi,2(N_{\max}-1)}/\lambda_{\Phi,2N_{\max}}|$. For $\bar{\rho} = r_u$, the highest-order term is equal to the second-highest one, and we expect convergence for $\bar{\rho} \ll r_u(N_{\max})$. We find that $r_u$ grows and reaches $r_u \approx 0.1$ at $N_{\max} = 17$. A simple extrapolation to $N_{\max} \to \infty$ leads us to $r_u \approx 0.23$. We expect the same tendency for the convergence radius of the Taylor expansion. Furthermore, we analysed the potential at a fixed field value $\bar{\rho} \subset [0,\bar{\rho}_{\max}]$ as a function of the expansion order $N_{\max}$. The functional behaviour is roughly given by $1/N_{\max}$ for any $\bar{\rho}$ chosen and thus the potential value in the $N_{\max} \to \infty$ limit is compatible with zero. This provides further evidence that the fixed-point potential is indeed converging towards a flat potential.

The eigenvalues are discussed in detail in Section V A 3 and are displayed in the right panel of Figure 5. The eigenvalues converge well, in particular, the most relevant ones. Importantly, the fixed point potential has two relevant parameters with eigenvalues given by

$$\theta_1 = -1.93 \pm 2 \cdot 10^{-3}\,, \quad \theta_2 = -0.811 \pm 7 \cdot 10^{-5}\,. \qquad (56)$$

Flat potentials with *two* relevant directions are *qualitatively* different from the standard Gaußian potential. The latter potential only has *one* relevant direction at the ASSM fixed point with the eigenvalue $\theta_{\Phi^2,\text{Gauß}} = -2 + \frac{3g_h^*}{(1+\mu_h^*)^2\pi} \approx -0.811$ equal to $\theta_2$ in (56). Naturally, this FP is embedded in the two-dimensional critical surface of the non-trivial flat fixed point if using a vanishing coupling for the operator related to $\theta_1$.

This begs the question of how the solution in Figure 5 can converge towards the flat potential but retain a stable second relevant direction. For that purpose, we investigate eigenvectors $v^{(i)}$ and eigenfunction $\varphi_i(\bar\rho)$, which are related with

$$\varphi_i(\bar\rho) = c_i(N_{\max}) \sum_{l=0}^{N_{\max}} v_l^{(i)} \bar\rho^l, \qquad (57)$$

with an $N_{\max}$-dependent normalisation $c_i$. For our evaluations within the Taylor expansion, the coefficient of the constant part, $v_0^{(i)}$, drops out and hence we do not consider it here. In Section V A 2, we will discuss the full (global) solution of the eigenfunctions including their constant parts. The constant part $u_0 = u(\bar\rho = 0)$ of the potential is related to the cosmological constant

$$u_0 = \frac{\lambda_{h,0}}{8\pi g_h}, \qquad u_0^* = \frac{1}{16\pi^2}\left(24 - \frac{3}{1+\mu_h^*}\right), \qquad (58)$$

with $\lambda_{h,n}$ and $g_{h,n}$ as defined in (22). Note also that in the present work we use the approximation, that all avatars of the Newton constants are identified with $g_h$. Note also that $\lambda_{h,0}$ or rather its dimensionful version $\Lambda_{h,0} = k^2\lambda_{h,0}$ is the observable cosmological constant at $k = 0$. It is an irrelevant coupling, as its dynamical rôle is taken by $\mu_h = -2\lambda_{h,2}$.

The normalisation with $c_i$ is required as the fixed-point potential vanishes in the limit $N_{\max} \to \infty$, and so does the norm of all eigenvectors $v^{(i)}$. Accordingly the normalisation $c_i$ has to be chosen such that the eigenvectors $\bar v^{(i)}$ converge with a finite norm, leading also to converging eigenfunctions $\varphi_i$. Accordingly, a convenient choice for $c_i$ is given by the inverse of a specific $v_{l_i}^{(i)}(N_{\max})$ with a fixed $l_i$. It is suggestive to choose $l_i = 1$ for all $i$. However, since potentially $v_1^{(i)} = 0$ for some eigenfunctions $\varphi_i$, this choice may not work globally. In the present case, we are only interested in $i = 1, 2$, where we indeed can choose

$$c_i(N_{\max}) = \frac{1}{v_1^{(i)}}, \qquad \text{for} \qquad i = 1, 2. \qquad (59)$$

We remark that the above construction only works if the respective eigenfunctions are well-defined in the limit $N_{\max} \to \infty$. We shall see that this is indeed the case, which provides further non-trivial evidence for the existence of the novel ASSM fixed point with a flat Higgs fixed point potential with two relevant directions.

### 2. Eigenperturbations and the full fixed-point potential

We briefly discuss the computation of the eigenfunctions from their respective fixed-point equations. Any potential can be expanded about the fixed-point solution in terms of the eigenfunctions,

$$u(\bar\rho) = u^*(\bar\rho) + \sum_i b_i\,\varphi_i(\bar\rho), \qquad (60)$$

where the eigenfunctions carry the scaling of the respective eigenperturbation with the eigenvalue $\theta_i$. Hence, the flow of the $\varphi_i(\bar\rho)$ in the vicinity of $u^*(\bar\rho)$ is given by

$$\partial_t\varphi_i(\bar\rho) = \theta_i\,\varphi_i(\bar\rho). \qquad (61)$$

This entails that the relevant perturbations die out with $\exp(\theta_i t)$ when approaching the UV fixed point as $\theta_i < 0$. In turn, the irrelevant ones grow with $\exp(\theta_i t)$ as $\theta_i > 0$.

We have shown above how the eigenfunctions $\varphi_i(\bar\rho)$ can be constructed from the $v^{(i)}$. They can also be derived from the respective fixed-point equation. For this derivation, we start with a potential that simply is the sum of the fixed-point potential $u^*(\bar\rho)$ and a perturbation in a given eigendirection $i$,

$$u(\bar\rho) = u^*(\bar\rho) + \varepsilon\,\varphi_i(\bar\rho), \qquad (62)$$

with an infinitesimal parameter $\varepsilon$. This is inserted in the flow and expanded in $\varepsilon$. The zeroth order in $\varepsilon$ is simply the fixed-point equation for the Higgs potential, leading to a potential converging to $u^*(\bar\rho) \equiv 0$. This implies that the fixed-point equation for the potential can also be expanded in powers of $u^*$ and hence we can relate $u^*$ to the eigenfunction $\varphi_0$ with vanishing eigenvalue in (61),

$$u^*(\bar\rho) = u_0^* + \varphi_0(\bar\rho), \qquad \text{with} \qquad \theta_0 = 0, \qquad (63)$$

with $u_0^*$ related to the cosmological constant, see (58). In (63), we have used that $u(\bar\rho) \to u_0^*$ when approaching the UV fixed point.

The linear order of the flow equation of $u(\bar\rho)$ gives the differential equation for the eigenperturbations $\varphi_i(\bar\rho)$ with

$$\mathcal{D}\,\varphi_i(\bar\rho) - \text{Flow}_u(\varepsilon\,\varphi_i(\bar\rho))\big|_{\mathcal{O}(\varepsilon)} = 0, \qquad (64)$$

where the flow term leads to further terms linear in $\varphi_i$, $\varphi_i'$, and $\varphi_i''$. The full scaling $\mathcal{D}\,\varphi_i$ of the eigenfunction $\varphi_i$ reads

$$\mathcal{D}\,\varphi_i(\bar\rho) = \Big[(4 + \theta_i) - (2 + \eta_\Phi)\,\bar\rho\,\partial_{\bar\rho}\Big]\varphi_i(\bar\rho). \qquad (65)$$

In the present case with a vanishing fixed-point potential $u^*(\bar\rho) \equiv 0$, the coefficients of the different $\bar\rho$-derivatives of $\varphi_i(\bar\rho)$ are at most linear in $\bar\rho$. After an appropriate redefinition of the eigenvalue, the scaling equation for the eigenperturbations reduce to that in a simple scalar theory,

$$(4 + \Theta_i)\,\varphi_i(\bar\rho) - (2 + \eta_\Phi)\,\bar\rho\,\varphi_i'(\bar\rho)$$
$$+ \frac{1}{16\pi^2}\left[2\varphi_i'(\bar\rho) + \bar\rho\,\varphi_i''(\bar\rho)\right] = 0, \qquad (66a)$$

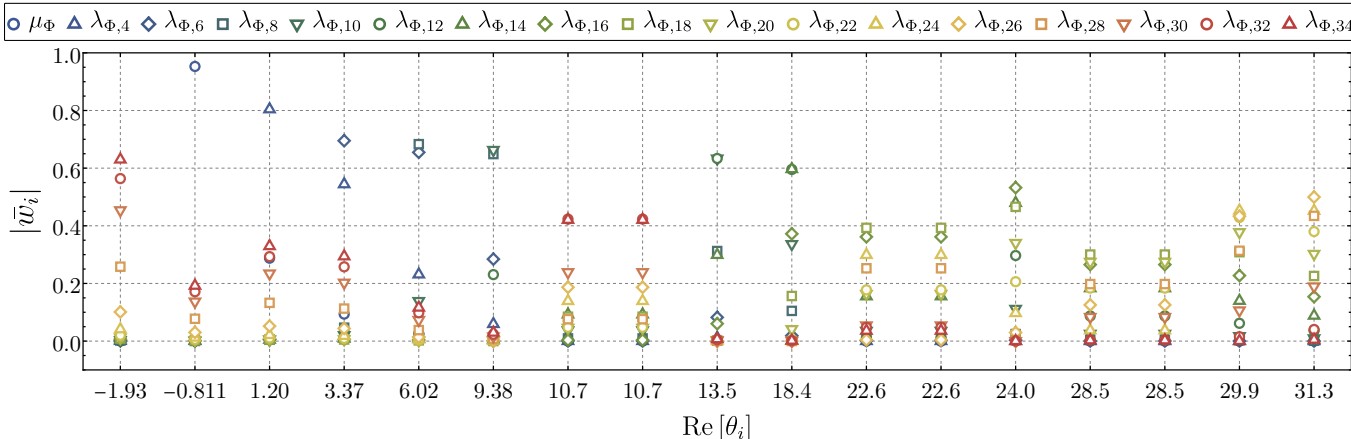

**Figure 6**. Weighted and normalised eigenvectors at order $N_{\max} = 17$. The most relevant eigenvalue $\theta_1 = -1.93$ has the largest overlap with the highest order coupling $\lambda_{\Phi,34}$. Otherwise we observe a clear correspondence between order of the coupling and relevance, i.e., $\theta_2 = -0.811$ has the largest overlap with $\mu_\Phi$, $\theta_3 = 1.20$ with $\lambda_{\Phi,4}$, etc.

with the shifted eigenvalues $\Theta_i$,

$$\Theta_i = \theta_i - \frac{3}{\pi} \frac{g_h^*}{(1 + \mu_h^*)^2} \, . \tag{66b}$$

In (66a), the full anomalous dimension of the Higgs field is proportional to $u_0' = \partial_{\bar\rho} u^*(0)$,

$$\eta_\Phi = \frac{3 g_h}{\pi} \frac{(2 + \mu_h)}{(1 + \mu_h)^2} \frac{u_0'}{(1 + u_0')^2} \, , \tag{66c}$$

and hence vanishes on the fixed point $u^*(\bar\rho) = 0$. This is a peculiarity of the gravity gauge fixing used here, the de-Donder gauge with $\alpha = 0$ and $\beta = 1$ [16]. For general $\alpha, \beta$ the Higgs anomalous dimension is present but this does not change the solution of (66) qualitatively.

In (66a) we have dropped the constraint $u^*(0) \equiv 0$ by simply omitting the subtraction of the flow at $\bar\rho = 0$. The latter would lead to $\varphi'(\bar\rho) \to \varphi'(\bar\rho) - \varphi'(0)$ in the square bracket. Apart from the slight simplification of the equation, dropping the subtraction also allows us to discuss the cosmological constant with the eigenvalue $\Theta_0 = -4$.

With (66b), the scaling equations (66a) are identical to those in a scalar SU(2) theory: the square bracket contains the contribution $3/2 \, \varphi_i'(\bar\rho)$ from three Goldstone modes, while the rest, $1/2 \, \varphi_i'(\bar\rho) + \bar\rho \, \varphi_i''(\bar\rho)$, stems from the radial Higgs field.

The scaling equations (66) for are of the Liouville-Sturm type for general $\Theta$, and the solutions $\varphi_i(\bar\rho)$ are Kummer's function of the first kind [112–114],

$$\varphi_i(\bar\rho) = c_i \, M\left( -\frac{4 + \Theta_i}{2}, \, 2, \, 32\pi^2 \bar\rho \right), \tag{67}$$

with free normalisations $c_i$. In (67), we have used $\eta_\Phi = 0$ for the present setup, the general solution is provided in Appendix C.

We are specifically interested in the $\bar\rho \to \infty$ behaviour of the eigenperturbations $\varphi_i(\bar\rho)$. For $\Theta_i = -4 + 2n$ with $n \in \mathbb{N}$, Kummer's function $M$ in (67) reduces to a polynomial of order $2 + \Theta_i/2$. In turn, for all other $\Theta$'s the solutions grow exponentially with $\exp(32\pi^2 \bar\rho)$ for $\bar\rho \to \infty$. Hence, the eigenfunctions and the potential are best expanded in Hermite polynomials, and the basis is square integrable with a unique solution for the potential. More details are provided in Appendix C.

The above analysis enables us to extend the fixed-point solutions $u^*(\bar\rho) = u_0^* + \varphi_0(\bar\rho)$ with $c_0 \to 0$ and the eigenperturbations $\varphi_{i>0}$ to the full $\bar\rho$-range. We concentrate on the fixed-point potential and fix the sign of the prefactor $c_0 \to 0$ with the first derivative $\varphi_0'(0) > 0$. This entails $\text{sign}(c_0) < 0$ and hence the fixed point potential diverges with $-\exp(32\pi^2 \bar\rho)$ for large $\bar\rho$. Evidently, this cannot be the global solution, while $u^*(\bar\rho) = 0$ is. It is suggestive that as in Dilaton gravity the coupling of the Higgs to the curvature scale as well as a $Z_\Phi(\bar\rho)$ is needed to arrive at a global solution. While this still suggests a potential with two relevant directions as the qualitative properties of the Taylor expansion remain unchanged, this global solution may also incorporate a non-trivial potential. While this is a highly relevant question, its resolution is beyond the scope of the present work and is left for future work.

### 3. Critical exponents and relevance

We introduce the weighted and normalised eigenvectors [115] of the critical exponents. This rescaling and normalisation are determinants to quantify the real overlap of each operator within each direction. The 'ad hoc' determined eigenvectors are known to be spuriously dominated by the highest orders of the expansion and therefore an equal weight of each operator must be imposed. The normalised eigenvector associated with the $i$-th critical

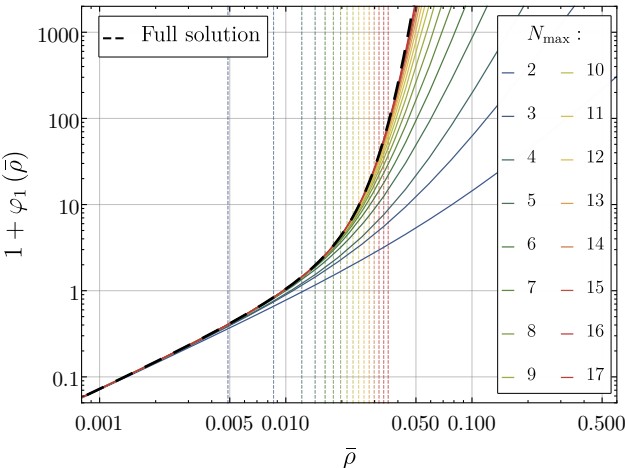

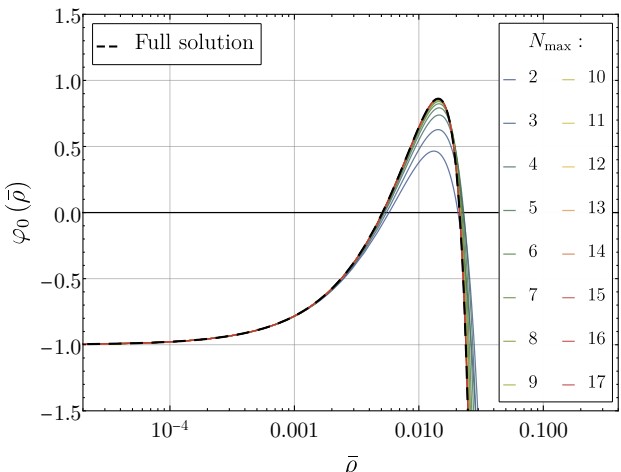

**Figure 7**. Eigenfunction $1 + \varphi_1(\bar{\rho})$ of the most relevant critical exponent $\theta_1$: full solution (dashed black), see (67), and different expansion orders $N_{\max} = 2, ..., 17$. The properties of $\varphi_1$ are best seen if plotted on a double logarithmic scale, hence we show $1 + \varphi_1(\bar{\rho}) \geq 0$ with the normalisation $c_1 = -1$ in (68). The rapid convergence of the Taylor expansion towards the full solution is clearly seen. We also depict the reliability regime of the Higgs fixed-point potential $u^*$ with the vertical lines as defined in (55).

**Figure 8**. The fixed-point potential $\varphi_0(\bar{\rho})$ defined with (63). We compare the full solution (dashed black), see (67), and with different orders in the Taylor expansion $N_{\max} = 2, ..., 17$. We choose a fixed normalisation with a negative $c_0 = -1$ which implies $\varphi'_0(0) = 221.9$. This captures the sign of the actual normalisation $c_0(N_{\max}) \to 0_-$: it goes to zero from below in agreement with the vanishing fixed-point potential $u^* \equiv 0$. The rapid convergence of the Taylor expansion towards the full solution is clearly seen.

exponent is defined as

$$\bar{w}_l^{(i)} = \frac{\zeta_l \, v_l^{(i)}}{\sqrt{\sum_{m=0}^{N-1} \left| \zeta_m v_m^{(i)} \right|^2}} \, , \tag{68}$$

where $\zeta_l$ are the rescaling vectors to the 'ad hoc' determined eigenvector $v_l^{(i)}$ with directions $l \subset (\mu_\Phi, \lambda_{\Phi,4}, \lambda_{\Phi,6}, \dots)$, see [115] for a detailed derivation.

The coefficients of the weighted eigenvectors are depicted in Figure 6. We observe that the eigenvector $\bar{\omega}^{(2)}$ of the eigenvalue $\theta_2 = -0.811$ has the largest overlap with $\mu_\Phi$. Indeed, solving the fixed-point equation for the respective eigenfunction $\varphi_2(\bar{\rho})$ in (67) for the exact eigenvalue $\Theta_2 = -2$ with $c_2 = -1/(16\pi^2)$, we find the exact solution

$$\varphi_2(\bar{\rho}) = \bar{\rho} - \frac{1}{16\pi^2} \, . \tag{69}$$

The constant comprises the overlap with the cosmological constant which is not present in $u(\bar{\rho})$ with its normalisation $u(0) \equiv 0$. Accordingly, it should be subtracted. Thus, the eigenvector $v^{(2)}$ of the full solution points exactly in the $\mu_\Phi$ direction, as does $\bar{\omega}^{(2)} = (1, 0, ....)$. In turn, the numerical solution of the fixed-point equation for $\varphi_2$ and the eigenvector $\bar{\omega}^{(2)}$ derived from the fixed-point potential $u^*$ and depicted in Figure 6 also includes higher-order terms. The latter originate in the subleading corrections in $\theta_2$ in comparison to the anomalous dimensions in the fixed-point equation and give an estimate for the numerical accuracy of the $\theta_i$. These terms grow large for large $\bar{\rho}$

and are an additional source for the finite validity range of the current Taylor expansion observed in the fixed-point potential, see the left plot in Figure 5.

The eigenvectors $\bar{\omega}^{(i)}$ of the higher eigenvalues $\theta_{i>2}$ are not exactly aligned with one coupling as it is the case for $\bar{\omega}^{(2)}$. Hence, the respective eigenfunctions $\varphi_i$ are not monomials. However, the normalised eigenvector $\bar{\omega}^{(3)}$ of the eigenvalue $\theta_3 = 1.20$ has the (by far) largest overlap with $\lambda_{\Phi,4}$, the eigenvector $\bar{\omega}^{(4)}$ of the eigenvalue $\theta_4 = 3.37$ overlaps dominantly with $\lambda_{\Phi,6}$, and in general the normalised eigenvectors $\bar{\omega}^{(i)}$ of $\theta_i$ have the largest overlap with $\lambda_{\Phi,2i-2}$ for $i \leq 2$. This matches the expectation from the mass dimension of the operator.

The eigenvector $\bar{\omega}^{(1)}$ of the most relevant direction with the eigenvalue $\theta_1 = -1.93$ has the largest overlap with the highest-order coupling, that is with $\lambda_{\Phi,N_{\max}}$. We emphasise that in the present case with a flat effective potential this is expected and almost required: in a Taylor expansion this is arranged (for small enough fields) by a cancellation between all monomials, leading to relatively large Taylor coefficients which trigger the observed relevance ordering. Indeed, this is confirmed within a comparison of the global solution $\varphi_0(\bar{\rho})$ in (67), which is not a polynomial, and hence features growing Taylor coefficients. Indeed, the high-order Taylor expansion approaches the global solution (67) as depicted in Figure 7. There we also depict the reliability regime of the Higgs fixed-point potential with the vertical lines as defined in (55). We could have defined a reliability regime analogously to (55) by simply substituting $\Delta u^*(\bar{\rho}) \to \varphi_i(\bar{\rho})$ and $\mathcal{D}(\Delta u^*(\bar{\rho}))$ with (65). A more conservative estimate

arises from using the reliability regime of the Taylor expansion of $\Delta u^*(\bar{\rho})$. Evidently, the latter leads to a smaller reliability regime since $\Delta u^*(\bar{\rho}) \to 0$ and the denominator in (55) is vanishing in the large $N_{\max}$ limit, apart from the absolute convergence of the fixed-point equation. Hence, we will resort to the reliability estimates from the potential throughout as a conservative estimate.

The same analysis can be done for the fixed-point potential $u^*(\bar{\rho})$ with the form (63): for $k \to \infty$ the potential converges towards its constant part $u^*(\bar{\rho}) = u_0^*$ which is related to the cosmological constant with (58). Accordingly, the $\bar{\rho}$-dependent part converges towards $\varphi_0(\bar{\rho})$ with a prefactor $c_0(N_{\max}) \to 0$. This expectation is confirmed by the explicit comparison, see Figure 8. There, we show the Taylor expansion of the potential for various $N_{\max}$ in comparison to the full solution $\varphi_0(\bar{\rho})$. In Figure 8, we fix the normalisation $c_0$ of $\varphi_0(\bar{\rho})$ defined in (67) as $c_0 = -1$, implying $\varphi_0'(0) = 221.9$. A fixed negative $c_0$ is taken as the Taylor expansion implies that $c_0(N_{\max})$ approaches zero from below. As is evident from Figure 8, the Taylor expansion is converging rapidly towards $\varphi_0(\bar{\rho})$ with $c_0(N_{\max}) \to 0_-$. This fully confirms the structure of the solution already seen in the Taylor expansion.

Finally, this global analysis entails that for finite $k$ the potential cannot approach the fixed-point potential in terms of the Kummer function. This is reminiscent to the situation in dilaton gravity where the effective potential had to be augmented with a field-dependent scalar wave functions and a field-dependent Newton coupling in order to access the global solution [116, 117]. The highly relevant question, whether the present solution can be extended to a global one, will be discussed elsewhere.

### B. Higgs potential landscape

We now proceed with an analysis of the landscape of possible Higgs fixed-point potentials in dependence of the gravity parameters, $g_h^*$ and $\mu_h^*$. This analysis explores the stability of the Higgs fixed-point solution under the variation of gravity parameters given that there is a big uncertainty on the gravity fixed-point values. We investigate the regime $g_h^* \in [0, 1]$ and $\mu_h^* \in [-1, 0]$. We will argue later that $g_h^*$ and $\mu_h^*$ can be combined into an effective gravity coupling and therefore the range of parameters gives us a complete view on the Higgs fixed-point solutions. The resulting phase structure of the UV fixed point is depicted in Figure 9.

Apart from the standard Gaussian fixed point potential, that exists for all values of $g_h^*$ and $\mu_h^*$, we found three further, non-trivial, types of fixed-point potentials. The first is the non-trivial flat potential discussed in the previous section, see Figure 5, whose two-dimensional critical manifold naturally includes the standard Gaußian FP with one relevant direction mentioned above. This non-trivial FP with a flat fixed-point potential exists in a rather large parameter space, depicted in *green* in Figure 9. At the upper and lower boundary of the green region, one of the

critical exponents approaches zero and it is suggestive that the fixed point vanishes due to a fixed-point collision. At the upper boundary, the second smallest eigenvalue $\theta_2$ approaches zero, while at the lower boundary, the third smallest eigenvalue $\theta_3$ approaches zero.

We also find a stable fixed-point potential with no relevant directions. This fixed-point potential exists only in the small *lime-green* band where it exists simultaneously with the non-trivial flat potential discussed before. We display the potential and the critical exponents of point (B) in Figure 15 in Appendix D. The upper boundary of this region coincides with the upper boundary of the green region: it is precisely the collision of these two fixed-point solutions that determine the end of these regions. At the lower boundary, the stable potential solution is lost, resembling a fixed point going to infinity.

The stable fixed-point potential has no relevant directions, and hence all parameters of the IR Higgs potential are fixed by the other couplings: It is a predictive fixed point, and specifically the ratio between Higgs and top mass as well as the ratio between EW and Planck scale are predictions. However, we find that in the current approximation the potential does not enter a symmetry-breaking regime in the IR and can therefore not be connected to SM physics.

The third fixed-point potential is unstable with one relevant direction, present in the *red* region of Figure 9. We display the potential and the critical exponents of point (A) in Figure 14 in Appendix D. Due to the instability of the potential, we consider this solution to be unphysical.

In summary, in the present approximation only the non-trivial solution in the green region is physical and compatible with the SM in the IR. We remark that the stability analysis here only takes into account the convergence regime of the Taylor expansion, no conclusion about the global stability properties of the potentials can be drawn.

The boundary between the red and lime-green region is determined by a fixed point collision and the eigenvalue corresponding to the $\Phi^2$ direction approaching zero. This allows us to analytically determine the boundary. For a fixed order in the Taylor expansion, both, the non-trivial flat potential and the stable potential, become flatter and we can evaluate the eigenvalue of the $\Phi^2$ direction for vanishing Higgs couplings. Then the eigenvalue is given by

$$\theta_{\Phi^2} = -2 + \frac{3 \, g_h^*}{(1 + \mu_h^*)^2 \, \pi} \,, \tag{70}$$

where $-2$ is the canonical value and the second term is the shift due to gravity. Solving for $\theta_{\Phi^2} = 0$, we obtain the boundary

$$\frac{3}{2\pi} \, g_h^* = (1 + \mu_h^*)^2 \,, \tag{71}$$

which is precisely the boundary line between the red and lime-green region in Figure 9.

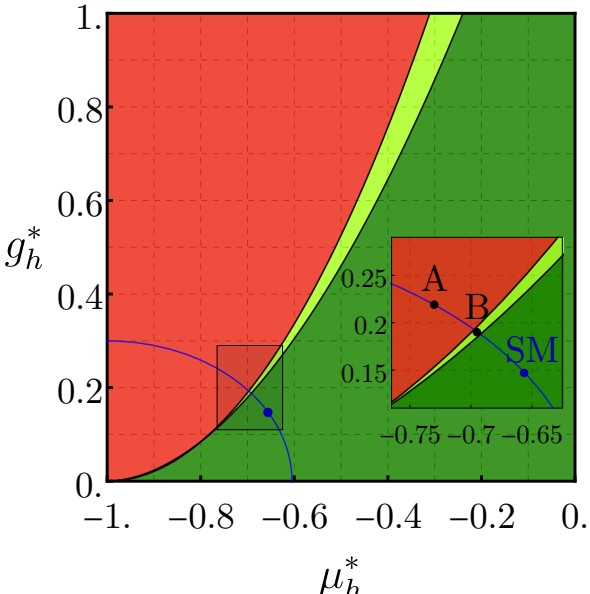

**Figure 9**. Fixed-point phase diagram of the scalar potential. In the green region, we observe the existence of a flat potential with two relevant directions, while in the lime region additionally a stable potential with zero relevant directions exists. In the red region, we find an unstable potential with one relevant direction. The boundary curves are described with (71) and (73). The effective gravity coupling is approximately given by $g_h^*/(1+\mu_h^*)^2$, see (75a), and it grows from the green to the red region. The *blue* circle segment avoids this redundancy as it is close orthogonal to the boundary lines. The Higgs potential corresponding to (SM) is shown in Figure 5, while the potentials for (A) and (B) are shown in Figures 14 and 15 in Appendix D.

The lower boundary of the green region is given by $\theta_{\Phi^4} = 0$, which we can again evaluate for vanishing Higgs couplings in the vicinity of the boundary

$$\theta_{\Phi^4} = \frac{3\,g_h^*}{(1+\mu_h^*)^2\,\pi}\,. \tag{72}$$

Therefore the boundary extends up to vanishing Newton coupling. Note also, that the solution does not exist on the boundary, $g_h^* = 0$.

We could not obtain the lower boundary of the lime-green region analytically. In fact, we found that the boundary is well described by a fractional power

$$0.6\,g_h^* = (1+\mu_h^*)^{1.85}\,, \tag{73}$$

which indicates a non-trivial competition between diagrams with $g_h^*/(1+\mu_h^*)^2$ and those with $g_h^*/(1+\mu_h^*)$.

The above analysis emphasises, that Figure 9 is effectively a one-dimensional plot since typical diagrams contain

$$(g_h^*)^n\,[G_h(\mu_h^*)]^m\,, \qquad n, m \in \mathbb{N}\,. \tag{74}$$

Typical combinations are $(n, m) = (1, 1)$ and $(n, m) = (1, 2)$, a detailed analysis can be found in [23]. Accordingly, a smaller Newton coupling $g_h^*$ is similar to a larger mass parameter $\mu_h^*$, as the latter decreases the fixed-point propagator $G_h(\mu_h^*)$. We define the effective fixed point (Newton) coupling

$$\hat{g}^* \approx \frac{g_h^*}{(1+\mu_h^*)^2}\,, \tag{75a}$$

which grows from the green to the red region

$$\hat{g}_{\text{green}}^* \lesssim \hat{g}_{\text{lime-green}}^* < \hat{g}_{\text{red}}^*\,, \tag{75b}$$

potentially implying a weak gravity bound $\hat{g}^* < \hat{g}_{\text{red}}^*$. Note that in an on-shell approximation the two-dimensional plot Figure 9 collapses to a line as this approximation lacks any $\mu_h$-dependence, and (75b) follows.

## VI. IR-PREDICTIVITY & SYSTEMATICS

In this section, we discuss the physics of the UV fixed point that has been presented in Section IV and also evaluate the systematic error estimate based on the current truncation. In the present approximation, the ASSM lies in a regime with a non-trivial but flat Higgs potential with two relevant directions. Consequently, the matter sector of the ASSM has as many relevant parameters as the SM. Additionally, the gravity sector shows in the present approximation another two relevant directions: the cosmological constant and the scalar curvature (Ricci scalar). The number of relevant directions in the gravity sector might increase to three, including the Ricci-scalar squared term, upon improving the truncation [62, 115, 118–120].

The non-trivial flat Higgs potential has two relevant parameters but nonetheless, we need to study which IR values can be reached from this fixed point. We observed that the fixed point is approached from below, see Figure 5. This translates into the requirement of a negative quartic Higgs-self coupling around the Planck scale. Lowering the top mass leads to a larger metastability scale and eventually to a positive quartic coupling at the Planck scale. In summary, the values of $M_H$ and $M_t$ that can be reached are displayed by the *green* regime in Figure 5. The boundary of this green region is given by the *plain white* line which contains all points that connect to the standard Gaußian Higgs potential in the UV. The *white dashed* lines in Figure 5 display our estimated error on this Gaußian line. The standard Gaußian Higgs potential has a higher predictivity since it has one relevant parameter less and therefore it relates the values of the Higgs and top masses, as displayed in Figure 10. The values of $M_H$ and $M_t$ in the *red* region connect neither to the non-trivial flat Higgs potential nor to the standard Gaußian Higgs potential. In fact, we did not find any viable UV completion for these points.

The *blue* ellipses in Figure 10 display the experimental uncertainties of $M_H$ and $M_t$ at $1\sigma$, $3\sigma$, and $5\sigma$. The

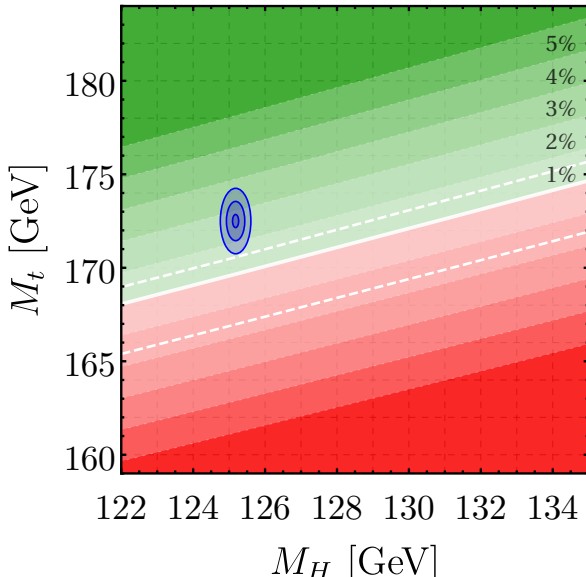

**Figure 10**. Values of $M_H$ and $M_t$ for which the standard Gaußian fixed-point Higgs potential is reached (white line). The values in the green region can be approached from the non-trivial flat Higgs potential presented in Figure 5 while the red region cannot. The white shadings show $1\%, \ldots, 5\%$ error in the overall computation, including the pole mass estimation. The blue ellipsis displays the SM experimental measurement at $1\sigma$, $3\sigma$, and $5\sigma$. The upper and lower dashed lines correspond to the Gaußian fixed-point trajectory for $\alpha_{s,k=M_Z} = 1.1\,\bar{\alpha}_s$ and $\alpha_{s,k=M_Z} = \bar{\alpha}_s$ respectively.

ellipses are in the green region and the central values have an approximate distance of $2.9\,\text{GeV}$ to the standard Gaußian fixed point marked by a white line. Due to this vicinity, it is highly important to discuss the uncertainties going into the present computation. The location of the white line depends predominantly on the top mass, while the gravity fixed-point values and the threshold effect around the Planck scale only play a strongly subleading role. In consequence, we focus now on the uncertainties going into the top mass determination. For illustrative purposes, we show green and red shaded regimes for $\pm 1,2,3,4,5\%$ variations of the top mass determination in Figure 10.

We have identified four systematic error sources within our computation, and below we discuss their influence on the top-mass determination:

*(i)* The first source is the general truncation error on the running of the couplings. Here we neglected, for example, higher-order contributions due to the anomalous dimension, see the discussion in Appendix E 1. We observed that the inclusion of the next order in the anomalous dimensions only had a subleading effect. Furthermore, our truncation includes all IR threshold effects, and therefore we believe that this source of error is subleading, significantly smaller than $1\,\text{GeV}$.

*(ii)* The second systematic error source concerns the

location of the UV gravity fixed point. We expect the errors on the values of the gravity fixed point to be rather large, and this influences the running of the SM couplings in particular through threshold effects around the Planck scale. We tested this error source by varying the UV fixed-point values and found a very subleading effect on the location of the standard Gaußian line in Figure 10. The reason is the threshold effect at the Planck scale is washed out through the long RG running to the IR. We estimate that this effect is smaller than $0.1\,\text{GeV}$. Importantly, the gravity fixed-point values determine the existence of the non-trivial but flat Higgs fixed-point potential, see Figure 9 and thus the existence of a green regime in Figure 10.

*(iii)* The third error source relates to the conversion from Euclidean curvature mass to pole mass discussed in detail in Appendix F: the wave function $Z_t(p)$ of the top quark has a negligible cutoff dependence for $k \leq k_{\text{SSB}} = 930\,\text{GeV}$, see Figure 18. This also entails a negligible momentum dependence for $p \lesssim 10^3\,\text{GeV}$, see Section II B 1 and Appendix F. Hence, we have approximated the wave function with a constant one in this regime also for timelike momenta of the same size, $Z_t(p) \approx 1$ for at least $|p^2| \leq (210^2\,\text{GeV})^2$. From the minimal variation of the cutoff dependence in Figure 18 in this regime, we estimate the respective systematic error with $0.5\%$, which translates into an error of roughly $0.8\,\text{GeV}$ for the top pole mass determination, see also Section II B 1. An additional uncertainty stems from the strong coupling, as discussed in the next point. For the translation of Euclidean curvature mass to pole mass, this leads to an error of $^{+0.9}_{-0.2}\,\text{GeV}$, see (45g).

*(iv)* The largest systematic error stems from the uncertainty on the strong coupling, due to the translation of the $\overline{\text{MS}}$-value at $p = M_Z$, $\bar{\alpha}_s$, to the MOM$^2$ value used here, $\alpha_s$, see Appendix E 3. From results in pure Yang-Mills and $2 + 1$ flavour QCD [65, 68], we extrapolated that the conversion factor is given by $\alpha_{s,k=M_Z} = 1.07\,\bar{\alpha}_s$ in the ASSM, see (45j). This choice corresponds to the central white line in Figure 10. We use the interval $\alpha_{s,k=M_Z} \in [\bar{\alpha}_s, 1.10\,\bar{\alpha}_s]$, see (46), to estimate our error in this identification. Using $\alpha_{s,k=M_Z} = \bar{\alpha}_s$ shifts the standard Gaußian line by $-2.6\,\text{GeV}$ in the top mass, see the lower white dashed line in Figure 10, while using $\alpha_{s,k=M_Z} = 1.1\,\bar{\alpha}_s$ shifts the standard Gaußian line by $0.9\,\text{GeV}$ in the top mass, see the upper white dashed line in Figure 10, which touches the $5\sigma$ band of the experimental error. For $\alpha_{s,k=M_Z} \approx 1.15\,\bar{\alpha}_s$, the standard Gaußian line goes through the central experimental values since the metastability scale reaches the Planck scale, see (51), and the non-trivial flat potential needs to be approached from below, see Figure 5. A better control over this error would require a fully self-consistent computation of the strongly coupled IR-QCD sector with all SM flavours.

In summary, the experimental values of the Higgs and top mass can be reached from the non-trivial flat Higgs

potential with two relevant directions (56). The standard Gaußian fixed point has a distance of roughly 2.9 GeV from the central experimental values, which is similar to the sum of the errors listed above. This gives us confidence that the ASSM is not in the red regime, which we consider as non-trivial evidence for the *existence* of the ASSM. With the systematic errors listed above, we also consider it viable, while not most likely, to reach the SM values of $M_H$ and $M_t$ from the standard Gaußian Higgs potential.

In the present work, we have solely discussed the minimal ASSM. Hence, physics properties such as the stability considerations and the location and properties of the UV fixed points may change even qualitatively with the introduction of beyond Standard Model (BSM) degrees of freedom. These are relevant for a comprehensive discussion of open cosmological questions such as dark matter. A survey of asymptotically safe BSM theories, and in particular the impact on the stability of the Higgs potential and the physics properties of the UV fixed point is a natural and highly relevant extension of the present work. We hope to report on this in the near future.

## VII. CONCLUSIONS

In the present work, we have presented a comprehensive analysis of the phase structure of the Asymptotically Safe Standard Model (ASSM). The present analysis includes several qualitative improvements:

Firstly, RG-consistent flows have been derived and all flows carry the physical threshold effects required for a quantitative determination of the physical point in the parameter space of the ASSM. Secondly, in the deep IR the present approximation to the ASSM contains QCD with confinement and chiral symmetry allowing for a determination of most parameters at the physical cutoff scale $k = 0$ at vanishing momentum, $p = 0$. Thirdly, we have directly computed the pole mass of the top quark, which eliminates all ambiguities that go with mapping Euclidean curvature masses to the physical pole masses. Finally, our approximation includes a non-trivial Higgs potential for the transplanckian regime which is required for a predictive stability analysis of the Higgs fixed point potential.

A detailed discussion of our results on the phase structure of the ASSM and the nature of the UV fixed points has been presented in Sections III to V and Appendix D, cumulating in an evaluation of the systematics in Section VI. In short, in the present approximation the ASSM lies in a regime of a novel UV fixed point with a non-trivial but flat Higgs potential with two relevant directions with the eigenvalues (56), see Section V. Consequently, the matter sector of the ASSM has as many relevant parameters as the SM, and the gravity sector shows another three relevant directions with overlap to the cosmological constant, the scalar curvature (Ricci scalar) term and the Ricci-scalar squared term. The distance of the solution

to the standard Gaußian fixed point regime with one relevant parameter less (also called 'predictive regime') is depicted in Figure 10. For sufficiently small top pole mass we first cross the standard Gaußian regime and the non-trivial stable regime, before entering the unstable regime. We remark, that while not excluded, we consider the distance of roughly 10 GeV as rather large. While the current approximation is not fully quantitative, significant changes would be required for entering the 'predictive' regime. Note also, that this regime is but one step away from the unstable one, and we consider the relatively large distance from the latter as a positive outcome of the present analysis.

The current minimal ASSM does not provide an answer to fundamental open problems as for example the nature of dark matter or the fermion mass hierarchies. Nevertheless, the framework here presented shows great potential for BSM studies in the need of non-perturbative methods or mass dependence schemes. It provides an easily adaptable and RG-consistent playground in which the viability of models phenomenology can be tested. Moreover, this setup can be easily modified towards the study of effective field theories, different matter contents or other types of UV completions thanks to its versatility. Investigations of non-trivial Higgs potentials [42, 45] which follow this line have already been performed.

In conclusion, the present comprehensive analysis of the ASSM constitutes a further important step towards the quantitative construction of the ASSM. Moreover, the present RG-consistent set-up can be systematically improved, important next steps in the SM sector of the ASSM are the inclusion of the CKM matrix, the inclusion of the full Higgs potential at all scales, the resolution of further pole masses, as well as general momentum dependences. On the gravity side, the momentum dependences of the fixed point solutions have to be used in the flow, gravity-induced matter vertices have to be added and a more complete tensor basis for the gravity vertices has to be used. All the above steps either have already been investigated in parts of the ASSM, or are work in progress. The present set-up also allows for a comprehensive study of scattering amplitudes giving access to the chiefly important question of unitarity in the ASSM. We hope to report on the respective progress soon.

## ACKNOWLEDGMENTS

We thank T. Denz, G. de Brito, H. Gies, F. Goertz, B. Knorr, A. Pereira, M. Schiffer, G. P. Vacca, C. Wetterich, N. Wink, M. Yamada, and L. Zambelli for discussions. Á. Pastor-Gutiérrez thanks T. Denz for technical support on VERTEXPAND. M. Reichert acknowledges support by the Science and Technology Research Council (STFC) under the Consolidated Grant ST/T00102X/1. This work is supported by EMMI, and is funded by Germany's Excellence Strategy EXC 2181/1 - 390900948 (the Heidelberg STRUCTURES Excellence Cluster) and the

DFG Collaborative Research Centre "SFB 1225 (ISO-QUANT)".

## Appendix A: Effective Action of the ASSM

We provide the gauge-fixed classical action (A1) of the ASSM in Appendix A 1 (gravity-matter sector) and Appendix A 2 (pure gravity part). The classical gauge fixed-action of the ASSM can be split into a gravity-matter part and a pure gravity part

$$S_{\text{ASSM}} = S_{\text{SM}} + S_{\text{gravity}} \,. \qquad (A1)$$

The gravity-matter part is the SM action in a given geometry defined by the metric $g_{\mu\nu}$. For the gauge fixing in the presence of a metric one usually introduces a linear split of the full metric into a background $\bar{g}_{\mu\nu}$ and a dynamical fluctuation $h_{\mu\nu}$ as shown in (2) which is then also taken for the expansion of the pure gravity action in terms of quantum fluctuations in a given background. The prefactor of the fluctuation term is chosen such that the fluctuation field $h_{\mu\nu}$ has momentum dimension one and its kinetic term has the canonical form of a spin-two field, a more detailed discussion of general splits can be found in [14].

### 1. Gauge-fixed Standard Model

The Euclidean SM action reads

$$
\begin{aligned}
S_{\text{SM}} = {}& \frac{1}{4} \int_x F_{\mu\nu}^a F_{\mu\nu}^a + \frac{1}{4} \int_x B_{\mu\nu} B_{\mu\nu} + S_{\text{gf}}^{\text{EW}} + S_{\text{ghosts}}^{\text{EW}} \\
& + \frac{1}{4} \int_x G_{\mu\nu}^b G_{\mu\nu}^b + S_{\text{gf}}^{SU(3)} + S_{\text{ghosts}}^{SU(3)} \\
& + \int_x (D_\mu \Phi_i)^\dagger (D_\mu \Phi_i) + V\left(\Phi_i^\dagger \Phi_i\right) \\
& + \sum_{j=1,2,3} \int_x \bar{\psi}_{i,j}^{L/R} \, \slashed{D} \, \psi_{i,j}^{L/R} + S_{\text{Yukawa}} \,, \qquad (A2)
\end{aligned}
$$

where

$$\int_x = \int \mathrm{d}^4 x \, \sqrt{|g|} \,. \qquad (A3)$$

The covariant derivative is defined in a concise form as

$$D_\mu = \partial_\mu - i \, \mathbf{g} \, \mathcal{A}_\mu \,, \qquad \mathcal{A}_\mu = \mathcal{A}_\mu^i t^i \,, \qquad (A4)$$

with the Lie algebra generators $t^i$,

$$[t^i \,, t^j] = i f^{ijk} t^k \,, \quad t^i \in \text{u}(1)_{\text{Y}} \times \text{su}(2)_{\text{L}} \times \text{su}(3)_{\text{C}} \,, \quad (A5)$$

and the structure constants $f^{ijk}$ of the SM gauge group $\text{U}(1)_{\text{Y}} \times \text{SU}(2)_{\text{L}} \times \text{SU}(3)_{\text{C}}$. The coupling matrix $\mathbf{g}$ in (A4) is diagonal in the Lie algebra with entries $g_{\text{Y}}, g_2, g_3$ in the respective subspaces. Here, $g_y$ is the hypercharge

coupling, $g_2$ is the weak coupling, and $g_3$ is the strong coupling. For the weak hypercharge gauge coupling we employ the normalisation

$$g_1 = \sqrt{\frac{5}{3}} \, g_Y \,. \qquad (A6)$$

We compute $\vec{g}_1, \vec{g}_2, \vec{g}_3$ from the flow of gauge-matter vertices, where the vectors $\vec{g}_i$ comprise the couplings derived from the different gauge-matter vertices.

The field strength tensor $\mathcal{F}_{\mu\nu}$ reads

$$\mathcal{F}_{\mu\nu} = \partial_\mu \mathcal{A}_\nu - \partial_\nu \mathcal{A}_\mu - i[\mathcal{A}_\mu \,, \mathcal{A}_\nu] \,. \qquad (A7)$$

The $\psi_{i,j}$ in (A2) stands for a general fermion field representing quark $q_{i,j}$ and lepton $l_{i,j}$ doublets with their respective chiralities. The index $j$ is the generation index, and $i$ is the isospin eigenvalue. We have included all three generations of leptons and quarks. We only consider interactions within fermions of the same generations, hence, a diagonal CKM matrix. This approximation is also used in the effective action discussed below.

The classical Higgs potential $V_\Phi$ in (A2) is given by

$$V_\Phi(\rho) = \mu_\Phi \rho + \lambda_\Phi \rho^2 \,, \qquad (A8)$$

with the Higgs doublet $\Phi$ and its radial field $\rho$,

$$\Phi = \frac{1}{\sqrt{2}} \begin{pmatrix} \mathcal{G}_1 + i\mathcal{G}_2 \\ v + H + i\mathcal{G}_3 \end{pmatrix}, \qquad \rho = \text{tr} \, \Phi^\dagger \Phi \,. \qquad (A9)$$

Here, $H$ is the Higgs field that acquires an expectation value and the $\mathcal{G}_i$ are the Goldstone modes.

The Yukawa terms of the SM read

$$
\begin{aligned}
S_{\text{Yukawa}} = \int_x {}& y_{\text{u}} \, \bar{q}_{i,1}^L \, \tilde{\Phi}_i \, q_{1,1}^R + y_{\text{d}} \, \bar{q}_{i,1}^L \Phi_i q_{2,1}^R \\
& + y_{\nu_e} \, \bar{l}_{i,1}^L \, \tilde{\Phi}_i \, l_{1,1}^R + y_e \, \bar{l}_{i,1}^L \, \Phi_i \, l_{2,1}^R + \text{h.c.} + \dots
\end{aligned}
$$
$$(A10)$$

where we have only shown explicitly one single generation of fermions and where

$$\tilde{\Phi}_i \equiv i \, \sigma_2 \Phi_i^* \,. \qquad (A11)$$

The interaction terms between Goldstones and fermions will be used to read off the background field dependence of the Yukawa couplings. This subject is discussed within our systematics overview in Appendix E 4. Moreover, we have checked that the Yukawa interaction and respective mass terms of the Neutrinos lead to negligible effects in the flow. Thus, we have dropped them for the results obtained here.

Finally, we have to specify the gauge fixings in the EW and QCD sectors. In the EW sector, we employ an $R_\xi$-gauge fixing which is adapted to the occurrence of massive modes in the broken phase

$$
\begin{aligned}
S_{\text{gf}}^{\text{EW}} = \int_{\bar{x}} {}& \frac{1}{2 \, \xi} \left( \partial_\mu A_\mu^a - i \, g_2 \, \xi \, t_{ij}^a \left( \Phi_i^\dagger \, v_j - v_i^\dagger \, \Phi_j \right) \right)^2 \\
& + \frac{1}{2 \, \xi_Y} \left( \partial_\mu B_\mu - \frac{i}{2} \, g_Y \, \xi_Y \left( \Phi_i^\dagger \, v_i - v_i^\dagger \, \Phi_i \right) \right)^2 \,.
\end{aligned}
$$
$$(A12)$$

Here,

$$\int_{\bar{x}} = \int \mathrm{d}^4 x \, \sqrt{|\bar{g}|} \,, \qquad (A13)$$

refers to the volume factor of the background metric, which is taken to be the flat metric here. The associated Faddeev-Popov ghost action reads

$$S_{\mathrm{ghosts}}^{\mathrm{EW}} = \int_{\bar{x}} \Big[ -\partial_\mu \bar{c}_Y \left( \partial_\mu c_Y \right) - \xi_Y g_Y^2 \left( \bar{c}_Y c_Y \right) \phi_i^\dagger v_i \Big]$$

$$+ \int_{\bar{x}} \Big[ - \partial_\mu \bar{c}^a \partial_\mu c^a - \xi g_2^2 t_{ij}^a t_{jl}^b \left( \bar{c}^a c^b \right) \Phi_i^\dagger v_l$$

$$+ g_2 A_\mu^c f^{abc} \partial_\mu \bar{c}^a c^b \Big]. \qquad (A14)$$

Here, $v_i$ are the components of the Higgs background field, see (A9).

For the high-energy regime of QCD, we use the gauge-fixing and ghost action

$$S_{\mathrm{gf}}^{SU(3)} + S_{\mathrm{ghosts}}^{SU(3)} = \frac{1}{2\,\xi_3} \int_{\bar{x}} \left( \partial_\mu G_\mu^{c_a} \right)^2 + \int_{\bar{x}} \bar{c}^{c_a} \partial_\mu D_\mu^{c_a c_b} c^{c_b} \,, \qquad (A15)$$

in the Landau limit, $\xi_3 \to 0$.

## 2. Gauge-fixed gravity

The classical Einstein-Hilbert action is given by

$$S_{\mathrm{EH}} = \frac{1}{16\pi G_{\mathrm{N}}} \int_x (2\Lambda - R) + \frac{1}{2\alpha} \int_{\bar{x}} \bar{g}_{\mu\nu} F_\mu F_\nu$$

$$+ \int_{\bar{x}} \bar{c}_h^\mu \, \mathcal{M}_{\mu\nu} \, c_h^\nu \,, \qquad (A16)$$

with the linear de-Donder gauge fixing

$$F_\mu = \frac{1}{16\pi} \left( \bar{D}^\nu h_{\mu\nu} - \frac{1}{2} \bar{D}_\mu h_\nu^\nu \right) . \qquad (A17)$$

Here, $\bar{D}$ is the covariant derivative only dependent on the background metric. The Faddeev-Popov operator for the gauge fixing (A17) reads,

$$\mathcal{M}_{\mu\nu} = \bar{D}^\sigma \left( g_{\mu\nu} D_\sigma + g_{\rho\nu} D_\mu \right) - \bar{D}_\mu D_\nu \,. \qquad (A18)$$

## Appendix B: Flow equations in the ASSM

In this Appendix, we describe the projection procedure used to derive the flows for the couplings and anomalous dimensions from the functional flow (4) for the effective action. The anomalous dimensions are discussed in Appendix B 1, the flow of matter couplings except the Higgs self couplings is discussed in Appendix B 2. The flow of the Higgs potential is discussed in Appendix B 3, and that of the pure gravity parameters is discussed in Appendix B 4. Finally, in Appendix B 5 we discuss the flows in the strongly correlated IR sector of QCD.

In short, we perform field derivatives of the functional flow (4) to obtain flows for correlation functions $\partial_t \Gamma^{(n)}(p_1, \ldots, p_n)$, and all flowing parameters are obtained by evaluating the flow of the respective correlation functions (or its momentum derivative in the case of anomalous dimensions) at vanishing momentum, $p_1 = \ldots = p_n = 0$, in the spirit of the derivative expansion, with the exception of the gravity couplings. The symbolic expressions of the flows on $n$-point functions were derived using the Mathematica package DoFun [121].

In the present work, we use RG-adapted [80] regulators

$$R_\phi(p) = Z_\phi R_\phi^{(0)}(p) \,, \quad R_\phi^{(0)}(p) = P_\phi(p) r(p^2/k^2) \,, \quad (B1)$$

with the classical dispersions $P_\phi(p)$, and the shape function of the Litim-type regulator [122, 123],

$$r(x) = (1 - x) \, \Theta(1 - x) \,, \qquad x = \frac{p^2}{k^2} \,, \qquad (B2)$$

with the Heaviside step function $\Theta$. The Litim-type regulator leads to analytic flows. It is also optimised for the 0th order derivative expansion [80, 122, 123], however, it is not the optimal regulator beyond this order. Optimal regulators can be constructed by functional optimisation [80]. Moreover, in the light of highly interesting further investigations of the resonance structure of the ASSM, scattering processes and the quantitative determination of (pole) masses there are further constraints on the analytic structure of the regulator also for timelike momenta, for related discussions in scalar theories, QCD and pure gravity see [103, 124–130].

## 1. Anomalous dimensions

The anomalous dimensions

$$\eta_{\phi_i} = -\frac{\partial_t Z_{\phi_i}}{Z_{\phi_i}} \,, \qquad (B3)$$

are obtained from the flow of the respective two-point functions $\partial_t \Gamma^{(2)}$. For leptons and quarks, we define

$$\eta_\psi = \frac{\mathrm{i} \, \mathrm{tr} \Big[ \slashed{p} \, \partial_t \Gamma_{\psi\bar{\psi}}^{(2)}(p) \Big]}{Z_\psi \, p^2 \, \mathrm{tr} \, \mathbb{1}} \Bigg|_{p=0} \,, \qquad (B4)$$

where the trace sums over all internal and group indices. Note also that the evaluation at $p = 0$ is tantamount to a $p^2$-derivative at $p = 0$ of the numerator.

For the Higgs field we use

$$\eta_\Phi = \frac{\mathrm{tr} \Big[ \partial_t \Gamma_{\Phi\Phi}^{(2)}(p) - \partial_t \Gamma_{\Phi\Phi}^{(2)}(0) \Big]}{Z_\Phi \, p^2 \, \mathrm{tr} \, \mathbb{1}} \Bigg|_{p=0} \,, \qquad (B5)$$

where the trace sums over all internal and group indices.

For the gauge fields, we use

$$\eta_{\mathcal{A}} = \left. \frac{\text{tr } \Pi^{\perp}(p)\left[\partial_t \Gamma^{(2)}_{\mathcal{A}\mathcal{A}}(p) - \partial_t \Gamma^{(2)}_{\mathcal{A}\mathcal{A}}(0)\right]}{Z_{\mathcal{A}}\, p^2 \,\text{tr } \Pi^{\perp}(p)} \right|_{p=0}, \quad \text{(B6)}$$

where the trace sums over all internal and group indices. In (B6) we have used the transversal projection operator

$$\Pi^{\perp}_{\mu\nu}(p) = \delta_{\mu\nu} - \frac{p_\mu p_\nu}{p^2}. \quad \text{(B7)}$$

The anomalous dimensions of the auxiliary ghost fields are extracted with the flows

$$\eta_{\mathcal{C}} = \left. \frac{\text{tr}\left[\partial_t \Gamma^{(2)}_{\mathcal{C}\mathcal{C}}(p)\right]}{Z_{\mathcal{C}}\, p^2} \right|_{p=0}, \quad \text{(B8)}$$

Finally, we remark that the wave functions completely drop out of the diagrams as they are multiplicative factors in all vertices and (inverse) propagators. The global wave function factor $Z_{\phi_i}$ due to the external legs each carrying a $Z_{\phi_i}^{1/2}$ is cancelled by the $1/Z_{\phi_i}$ in the definition of $\eta_{\phi_i}$. The only remnant dependence on $Z_\phi$ arises from the $t$-derivatives of the cutoff line $G_\phi\, \partial_t R_\phi\, G_\phi$ with propagators $G_\phi$ and regulators $R_\phi$. We are led to

$$\frac{1}{Z_\phi}\partial_t R_\phi(p) = \left[\partial_t - \eta_\phi\right]R_\phi^{(0)}, \quad \text{(B9)}$$

more details on the structure of the equations for the anomalous dimension and possible systematic extensions have been deferred to Appendix E.

### 2. Flows of matter couplings

All couplings considered here are computed from three-point functions. The only exceptions are the Higgs mass $\mu_\Phi$ and self-couplings $\lambda_{\Phi,\,2n}$ in (10). The flow of the four-point function avatars of the gauge couplings are identified with that computed from the three-point function of the gauge fields coupled to fermions. In the present approximation, this identification is a consequence of gauge consistency and can be derived from the respective Slavnov-Taylor identities which collapse into Ward identities in the present approximation. However, the present projections also overlap with non-classical tensor structures hence the gauge couplings from different correlation functions differ, and the Slavnov-Taylor identities are more complicated in improved approximations.

The flows of the couplings of three-point functions can be obtained from (7a) with

$$\partial_t \frac{\Gamma^{(n)}_k(p_1,...,p_n; \vec{\lambda}_k)}{\prod_{i=1}^n \sqrt{Z_{\phi_i,k}}} = S^{(n)}(p_1,...,p_n; \partial_t \vec{\lambda}_k). \quad \text{(B10)}$$

The left-hand side is simply given by

$$\partial_t \frac{\Gamma^{(n)}_k(p_1,...,p_n; \vec{\lambda}_k)}{\prod_{i=1}^n \sqrt{Z_{\phi_i,k}}} = \frac{1}{2}\sum_{i=1}^n \eta_{\phi_i,k} S^{(n)}(p_1,...,p_n; \vec{\lambda}_k)$$
$$+ \text{Flow}^{(n)}(p_1,...,p_n; \vec{\lambda}_k), \quad \text{(B11)}$$

where the second line is simply the flow diagrams of the given $n$-point function divided by the wave functions $\sqrt{Z_{\phi_i}}$ for each of its external legs,

$$\text{Flow}^{(n)}(p_1,...,p_n; \vec{\lambda}_k) = \frac{\partial_t \Gamma^{(n)}_k(p_1,...,p_n; \vec{\lambda}_k)}{\prod_{i=1}^n \sqrt{Z_{\phi_i,k}}}. \quad \text{(B12)}$$

As in the case of the anomalous dimensions this division, together with the use of RG-adapted regulators, eliminates all $Z_\phi$-dependence from the diagrams at the expense of an $\eta_\phi$-dependence.

Given the flow of a three-point function of the fields $\phi_{i_1}, \phi_{i_2}, \phi_{i_3}$ with the coupling $\lambda_{123} = \lambda_{\phi_{i_1}\phi_{i_2}\phi_{i_3}}$ we arrive at

$$\left(\partial_t - \frac{1}{2}\sum_{j=1}^3 \eta_{\phi_{i_j},k}\right)\lambda_{123}$$
$$= \text{tr}\left[P_{123}(\boldsymbol{p})\text{Flow}^{(n)}_{\phi_{i_1}\phi_{i_2}\phi_{i_3}}\right](\boldsymbol{p}=0), \quad \text{(B13)}$$

where $P_{123}$ is a projection operator or rather procedure such as used for the anomalous dimensions. We performed consistency checks to confirm the veracity of our setup. We found the expected universal results up to one loop in agreement with the literature [109].

In the solutions to the flows presented in Figures 3 and 4, we made different gauge fixing choices for each of the phases. In the symmetric phase, we chose the Landau gauge $\xi_Y = \xi = 0$. Note that in this regime the gauge dependency appears beyond one loop. In the broken phase, to facilitate a correct cancellation between the unphysical modes of the Goldstone bosons and EW ghosts and longitudinal polarisations we chose the Feynman gauge $\xi_Y = \xi = 1$.

### 3. Flow of the Higgs potential

In this section we derive the flow of the scalar field parameters: couplings, curvature mass and vacuum expectation value. Starting from the renormalised potential (9) and its dimensionless version (25a), the minimum $\bar{\rho}_0 = \frac{Z_H v^2}{2\,k^2}$ is defined by

$$\partial_{\bar\rho}\, u\,(\bar\rho)|_{\bar\rho_0} = 0. \quad \text{(B14)}$$

Taking a $t$-derivative, we obtain

$$\partial_t \bar\rho_0 = -\left.\frac{\partial_{\bar\rho}\left(\partial_t|_\rho\, u\,(\bar\rho) - \partial_t|_\rho\, \bar\rho\, \partial_{\bar\rho} u\,(\bar\rho)\right)}{\partial_{\bar\rho} u\,(\bar\rho)}\right|_{\rho_0}. \quad \text{(B15)}$$

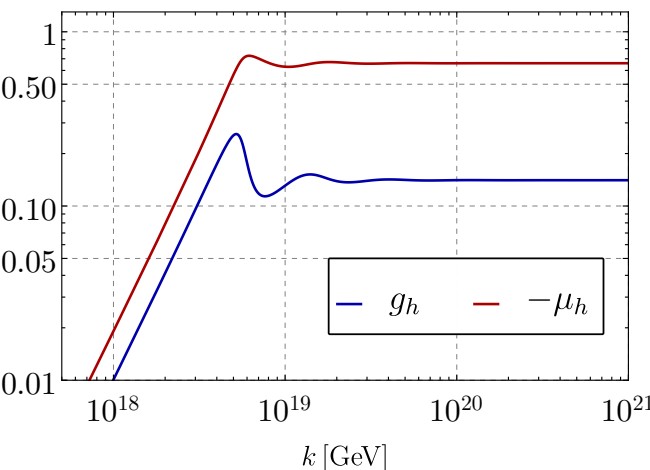

**Figure 11**. Dimensionless Newton coupling $g_h$ and graviton mass parameter $\mu_h$ as a function of the RG scale $k$. Above the Planck scale, the couplings approach an interacting fixed point while in the IR both coupling decrease quadratically towards the Gaußian fixed point.

Introducing the definition of the diagrammatic flows

$$\partial_t|_\rho V_{\Phi,\text{eff}}(\rho) = \text{Flow}\,[V_{\Phi,\text{eff}}]$$
$$= k^4 \left( \partial_t|_\rho \, u\,(\bar\rho) - \partial_t|_\rho \, \bar\rho\, \partial_{\bar\rho} u\,(\bar\rho) \right), \quad \text{(B16)}$$

and

$$\partial_t|_\rho \, \bar\rho = - \left( 2 + \eta_\Phi \right) \bar\rho. \qquad \text{(B17)}$$

we reach the final expression for the flow of the vacuum expectation value of the Higgs field in the broken ($\mu_\Phi < 0$) regime.

The flow of the scalar coupling $\lambda_{\Phi,4}$ and the curvature mass $\mu_\Phi$ have been obtained from the flow of scalar two- and four-point functions. For the higher $n$-point functions, this becomes tedious due to the large number of diagrams and external legs. Instead, we derive the flow of the scalar couplings $\lambda_{\Phi,2n}$ directly from (B16). Considering a vanishing background for simplicity, the flow of any arbitrary $\lambda_{\Phi,2n}$ can be derived by performing field derivatives of the flow of the effective potential,

$$\left( \partial_t|_{\bar\rho}\, \lambda_{2n} \right)\Big|_{\bar\rho=0} = \frac{1}{n!} \partial_{\bar\rho}^n \left[ \frac{1}{k^4} \text{Flow}\,[V_{\Phi,\text{eff}}] \right]$$
$$+ \lambda_{2n} \left( n\, \eta_\Phi + (2\,n - 4) \right). \quad \text{(B18)}$$

This procedure gives identical results to the derivation from $n$-point functions. On the technical side, it is important to take properly into account the mixing between the Higgs field and the scalar modes of the graviton.

### 4. Flow of gravity parameters

We extract the flow of the Newton coupling $g_h$ from the transverse-traceless graviton three-point function and the flow of the graviton mass parameter $\mu_h$ from the transverse-traceless graviton two-point function. All gravity flows and anomalous dimensions take into account momentum dependences and are evaluated bilocally between $p = 0$ and $p = k$. The flows are identical to [71] with the approximation $\lambda_{h,n\geq 3} = 0$, and the identification of all avatars of the Newton coupling with the avatar from the three-graviton vertex, see (24). This entails that the matter fields only contribute in their minimally coupled versions to gravity flows and in the end only enter via their field multiplicities. In the SM, these multiplicities are given by $N_s = 4$, $N_f = 45/2$, and $N_v = 12$. In the IR, we tune the Newton coupling and the graviton mass parameter such that we reach the physical value of the Newton coupling and a vanishing cosmological constant, see (45d). In Figure 11, we display the trajectories of $g_h$ and $\mu_h$.

### 5. Infrared regime of QCD

The IR regime of QCD with confinement and dynamical chiral symmetry breaking has been evaluated both qualitatively and quantitatively with functional approaches, for recent reviews see [13, 131]. We are solely interested in the running of the quark-gluon coupling, which is dominated by the gluon anomalous dimension $\eta_{A,k}$ and its IR flow below an *interface* scale $k_{\text{inter}}$ defined in (17).

It has been argued and checked in [66, 69], that the gluon anomalous dimension $\eta_{A,k}$ is well described as a function of the running coupling $\alpha_s$ and the mass gaps of quarks and gluons,

$$\eta_{A,k} \approx \eta_A(\vec\alpha_{s,k}, \vec{\bar{m}}_k^2), \qquad \bar m^2 = \frac{m^2}{k^2}, \qquad \text{(B19)}$$

with

$$\vec{\bar{m}}_k^2 = (m_{A,\text{gap}}^2, \vec{\bar{m}}_{q,k}^2) \qquad \text{(B20)}$$

The gluon mass gap in (B20) should not be confused with a gluon mass. The vector $\vec\alpha_{s,k}$ again comprises all avatars of the strong coupling; related to the primitively divergent part of the ghost-gluon, three-gluon, four-gluon and quark-gluon vertices. In the matter sector, we have avatars for each quark flavour, that differ below the mass thresholds of the quarks,

$$\alpha_{Aq_i\bar q_i} = \frac{1}{4\pi} \frac{\lambda_{Aq_i\bar q_i}}{Z_A^{1/2} Z_{q_i}}, \quad \text{with} \quad q = (d, u, s, c, b, t),$$
$$\text{(B21a)}$$

and in the pure glue sector we have three- and four-gluon couplings as well as the ghost-gluon coupling,

$$\alpha_{A^n} = \frac{1}{4\pi} \frac{\lambda_{A^n}}{Z_A^{n/2}}, \qquad \alpha_{A\bar cc} = \frac{1}{4\pi} \frac{\lambda_{A\bar cc}}{Z_A^{1/2} Z_c}, \qquad \text{(B21b)}$$

where the $\lambda$'s are the dressings of the classical tensor structures of the respective vertices [59, 66]. All the

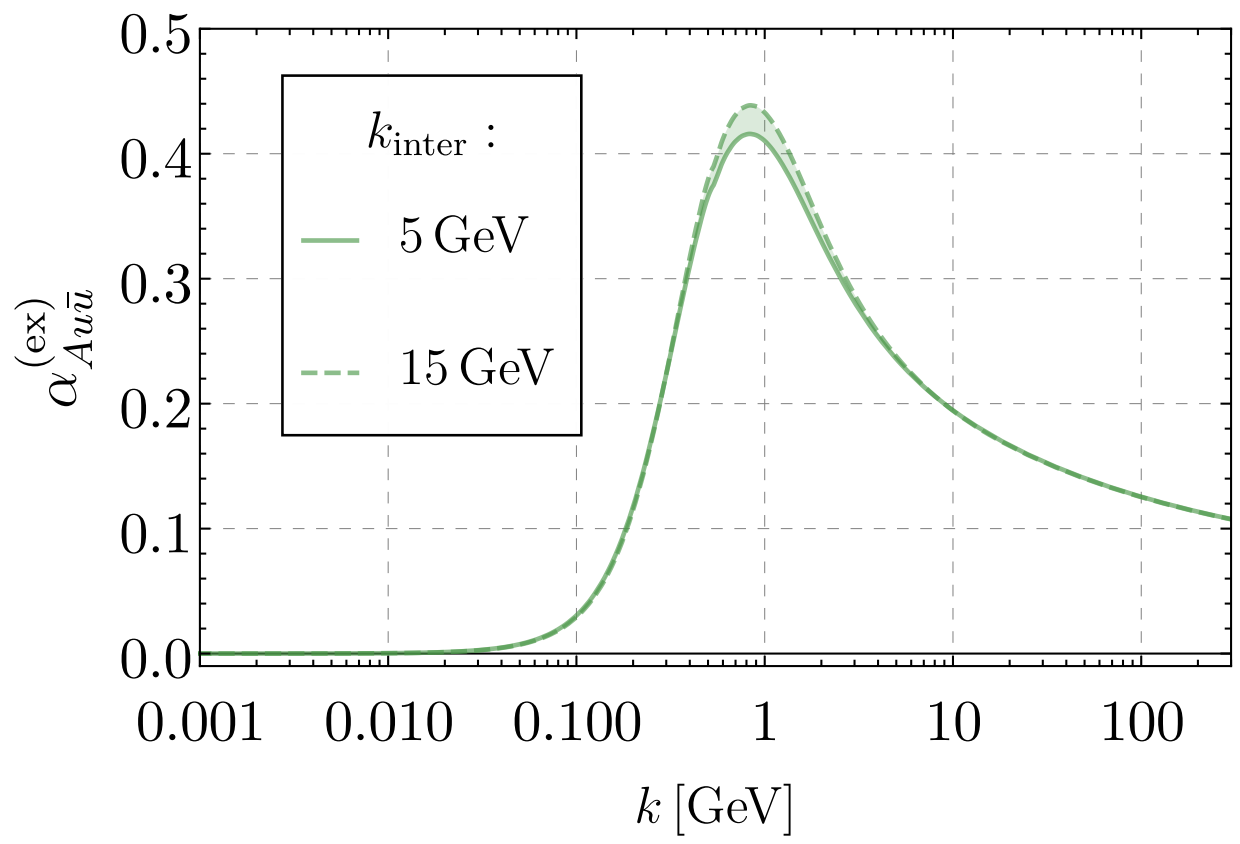

**Figure 12**. The quark-gluon exchange coupling $\alpha^{(\mathrm{ex})}_{Au\bar{u}}$ defined in (B21) as a function of $k$ for interface scales $k_{\mathrm{inter}} = 5, 15\,\mathrm{GeV}$. The chosen $k_{\mathrm{inter}}$ does not affect severely the IR dynamics.

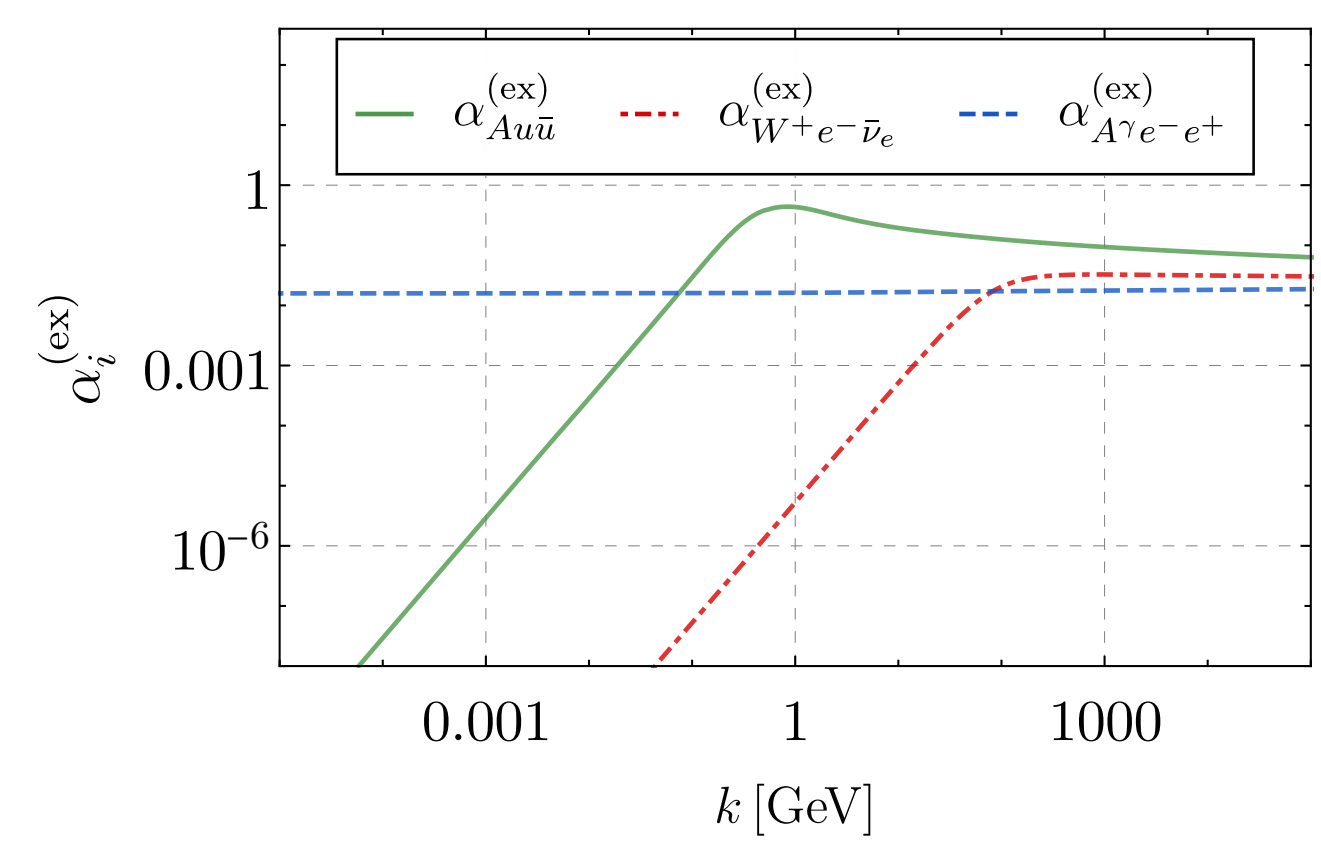

**Figure 13**. The exchange couplings of the quark-gluon coupling $\alpha^{(\mathrm{ex})}_{Au\bar{u}}$ (green, solid), the electron-$W^{\pm}$ coupling $\alpha^{(\mathrm{ex})}_{W^{\pm}e^{-}\bar{\nu}_e}$ (red, dash-dotted), and the photon-electron coupling $\alpha^{(\mathrm{ex})}_{A^{\gamma}e^{-}e^{+}}$ (blue, dashed) as defined in (B21) and (B31). The decrease of the strong and weak coupling towards small momenta is triggered by the QCD mass gap and the mass of the $W$ boson, respectively. The electromagnetic coupling does not decrease in the IR due to the absence of a photon mass.

avatars of the strong fine structure constants agree at two loops (in mass-independent RG schemes), $(\alpha_s)_i = \alpha_s$, but run differently with the cutoff scale and momenta in the IR due to physical threshold effects introduced by the gluon and quark mass gaps $\vec{m}^2$, (B20).

The above couplings are *exchange* couplings that describe the scattering of quarks ($\alpha_{s,Aq\bar{q}}$), gluons ($\alpha_{s,A^3}$), and ghosts ($\alpha_{s,Ac\bar{c}}$). These couplings occur directly in diagrams with the respective gluon lines, and carry directly the decoupling of gluons in the diagrams below the mass gap of QCD. In contradistinction, the electroweak $g_i$'s, depicted in Figure **3**, do not include the exchange propagators and are also called effective charges, see e.g. [105, 106]. For the definition and computation of a respective charge in QCD see [107, 108].

For the explicit computation, we utilise the 2+1 flavour IR-QCD results of [66]. For that purpose, we separate the 2+1 flavour part from the contributions $\eta_A^{(c,b,t)}$ of the heavier quarks

$$\eta^{(\mathrm{ASSM})}_{A,k} = \eta^{(2+1)}_{A,k} + \eta^{(c,b,t)}_{A,k}\,, \tag{B22}$$

where the 2+1 flavour contribution in (B22) depends on the full coupling and mass gaps of the ASSM,

$$\eta^{(2+1)}_{A,k} = \eta^{(2+1)}_{A,k}\left(\vec{\alpha}^{(\mathrm{ASSM})}_{s,k}, \left(\vec{m}^{(\mathrm{ASSM})}_{k}\right)^2\right). \tag{B23}$$

The strong coupling is a multiplicative factor in all (QCD) diagrams and hence we have

$$\eta_{A,k}(\alpha_{s,k}, \vec{m}^2_k) \approx \eta_{A,k}(\vec{m}^2_k)\alpha_{s,k}\,, \tag{B24}$$

where $\alpha_{s,k}$ is the strong coupling avatar of the lightest field, the ghost,

$$\alpha_{s,k} = \alpha_{A\bar{c}c,k}\,. \tag{B25}$$

The identification of all avatars in (B21) with the ghost-gluon coupling fails in the IR below the respective mass thresholds, see [59, 66]. There, however, the contributions to $\eta_{A,k}$ drop out. Furthermore, the quark masses in the 2+1 flavour computation are tuned to the physical ones. In summary, this supports the approximation

$$\eta^{(\mathrm{ASSM})}_{A,k} \approx \eta^{(2+1)}_{A,k}\frac{\alpha^{(\mathrm{ASSM})}_{s,k}}{\alpha^{(2+1)}_{s,k}} + \eta^{(c,b,t)}_{A,k}\,, \tag{B26}$$

also displayed in (19). Here, $\eta^{(2+1)}_{A,k}$ is now the genuine 2+1 flavour result and the difference between the ASSM strong fine structure constant and that in 2+1 flavour QCD is carried by the ratio of the couplings (B25). The contribution from the heavier quarks, $\eta^{(c,b,t)}_{A,k}$, is simply that of the ASSM.

Finally, we use that in leading order the ghost propagator dressing and ghost-gluon vertex dressing are insensitive to the number of quarks at all scales, and hence they drop out of the ratio. This leaves us with our final equation for the anomalous dimension,

$$\eta^{(\mathrm{ASSM})}_{A,k} \approx -\frac{\partial_t Z^{(2+1)}_{A,k}}{Z^{(\mathrm{ASSM})}_{A,k}} + \eta^{(c,b,t)}_{A,k}\,, \tag{B27}$$

where $\partial_t Z^{(2+1)}_{A,k}$ is the flow of the wave-function renormalisation in 2+1 flavour QCD, not in the ASSM. This can be re-arranged as a flow equation for the gluon wave-function renormalisation,

$$\partial_t Z^{(\mathrm{ASSM})}_{A,k} \approx \partial_t Z^{(2+1)}_{A,k} - \eta^{(c,b,t)}_{A,k} Z^{(\mathrm{ASSM})}_{A,k}\,. \tag{B28}$$

The right-hand side depends on the external input $\partial_t Z^{(2+1)}_{A,k}$ from 2+1 flavour QCD taken from [66], while all the other terms are that in the ASSM.

The flow (B27) is now solved for IR-QCD scales below the interface scale $k_{\text{inter}}$ in the range (17). The flow (B27) requires as an input the gluon wave-function renormalisation $Z_{A,k}^{(\text{ASSM})}$ at the interface scale. This input is fixed by the consistency condition that the anomalous dimension computed above and below $k_{\text{inter}}$ have to agree at the interface scale,

$$\left( \eta_{A,k_{\text{inter}}^+}^{(\text{ASSM})} - \eta_{A,k_{\text{inter}}^-}^{(\text{ASSM})} \right) = 0 \,, \quad k^\pm = \lim_{\epsilon \to 0} k \pm \epsilon \,. \quad \text{(B29)}$$

Inserting (B27) into (B29) leads us to

$$Z_{A,k_{\text{inter}}^-}^{(\text{ASSM})} = - \frac{\partial_t Z_{A,k_{\text{inter}}^-}^{(2+1)}}{\eta_{A,k_{\text{inter}}^+}^{(\text{ASSM})} - \eta_{A,k_{\text{inter}}^+}^{(c,b,t)}} \,, \quad \text{(B30)}$$

where the denominator is nothing but the $2+1$ flavour part of the gluon anomalous dimension in the ASSM. This provides us with the initial condition for solving (B27), also discussed in Section II C 1. For interface scales $5\,\text{GeV} \lesssim k_{\text{inter}} \lesssim 15\,\text{GeV}$, see (17), we find a small dependence of the IR coupling on the interface scale. This is depicted in Figure 12.

It is convenient to define *exchange* couplings not only for gluonic couplings but also for the other gauge fields,

$$\alpha_i^{(\text{ex})} = \frac{g_i^2}{4\pi} \left( \frac{1}{1 + m_i^2/k^2} \right) \,, \quad \text{(B31)}$$

which account for the masses of the mediating field in the scattering process with the intermediate gauge field coupled via $g_i$: the prefactor $g_i^2/(4\pi)$ are the fine structure constants derived from the $g_i$'s shown in Figure 3. The factor $1/(1 + m_i^2/k^2)$ constitutes the dimensionless exchange propagator of the gauge field.

In Figure 13, we depict the strong, weak, and hypercharge exchange couplings from fermion-gauge vertices. For scales below the threshold of the propagating field, the exchange coupling decreases reflecting the suppression of the interaction in the diagrams.

## Appendix C: Eigenperturbations and the global UV Higgs potential

The scaling equations (66) are of the Liouville-Sturm type for general $\Theta_i$, and the solutions $\varphi_i$ are Kummer's function of the first kind. Here, we recall the solution (67) including the Higgs anomalous dimension, which is vanishing due to the de-Donder gauge (A17) with $\alpha = 0$ and $\beta = 1$. The solution reads

$$\varphi_i(\bar\rho) = c_i \, M\left( -\frac{4 + \Theta_i}{2 + \eta_\Phi}, 2, 32\pi^2 \left(1 + \frac{\eta_\Phi}{2}\right) \bar\rho \right), \quad \text{(C1)}$$

with a free normalisation $c_i$. For $\Theta = -4 + (2 + \eta_\Phi)\, n$ with $n \in \mathbb{N}$, the Kummer function $M$ reduces to a polynomial with the $\bar\rho \to \infty$ limit

$$\varphi_i \to \frac{1}{\Gamma\left(4 + \frac{\Theta_i}{2}\right)} (32\pi^2 \bar\rho)^{\frac{4 + \Theta_i}{2 + \eta_\Phi}} \,. \quad \text{(C2)}$$

In turn, for all other $\Theta$'s the solutions grow exponentially for $\bar\rho \to \infty$,

$$\varphi_i \to \frac{1}{\Gamma\left(-2 - \frac{\Theta_i}{2}\right)} \frac{1}{(32\pi^2 \bar\rho)^{4 - \frac{\Theta_i}{2}}} e^{32\pi^2 \left(1 + \frac{\eta_\Phi}{2}\right)\bar\rho} \,, \quad \text{(C3)}$$

for

$$\Theta_i \neq -4 + (2 + \eta_\Phi)n \,, \qquad n \in \mathbb{N} \,. \quad \text{(C4)}$$

Hence, the eigenfunctions and the potential are best expanded in Hermite polynomials $H_n(\bar\phi)$,

$$H_n(\bar\phi) = (-1)^n e^{\bar\phi^2} \partial_{\bar\phi}^n e^{-\bar\phi^2} \,, \qquad \bar\phi^2 = 32\pi^2 a \bar\rho \,, \quad \text{(C5)}$$

with $a < 1$. This basis is square integrable and orthogonal with

$$\int_{\mathbb{R}} \mathrm{d}\bar\phi \, H_m(\bar\phi) H_n(\bar\phi) \, e^{-\bar\phi^2} = 2^n n! \sqrt{\pi} \, \delta_{nm} \,, \quad \text{(C6)}$$

and hence we can expand the $\varphi_i$ in the Hermite polynomials. We arrive at

$$\varphi_i(\bar\rho) = \sum_{n=0}^{\infty} c_n^{(i)} H_n(\bar\phi) \,, \quad \text{(C7)}$$

with

$$c_n^{(i)} = \frac{1}{2^n n! \sqrt{\pi}} \int_{\mathbb{R}} \mathrm{d}\bar\phi \, \varphi_i(\bar\rho) \, H_n(\bar\phi) \,. \quad \text{(C8)}$$

This concludes our analysis of the Higgs fixed-point potential and the eigenperturbations.

## Appendix D: UV Higgs potentials at different gravity parameters

In Section V, we have discussed the rich landscape of Higgs fixed-point potentials. Here, we study the potentials at the gravity parameters (A) with

$$g_h^* = 0.219 \,, \qquad \mu_h^* = -0.730 \,, \quad \text{(D1)}$$

and (B) with

$$g_h^* = 0.190 \,, \qquad \mu_h^* = -0.695 \,, \quad \text{(D2)}$$

marked in Figure 9 which characterise each of the regimes found.

The potential at point (A) belongs to the red regime and is shown in Figure 14. The scalar mass parameter and all scalar couplings $\lambda_{\Phi,2n}$ are negative. In particular, for the lowest-order expansion, we find

$$\mu_\Phi^* = -0.230 \,, \qquad \lambda_{\Phi,4}^* = -2.22 \,. \quad \text{(D3)}$$

The critical exponents show a reasonable convergence pattern and in particular, the six lowest values are well

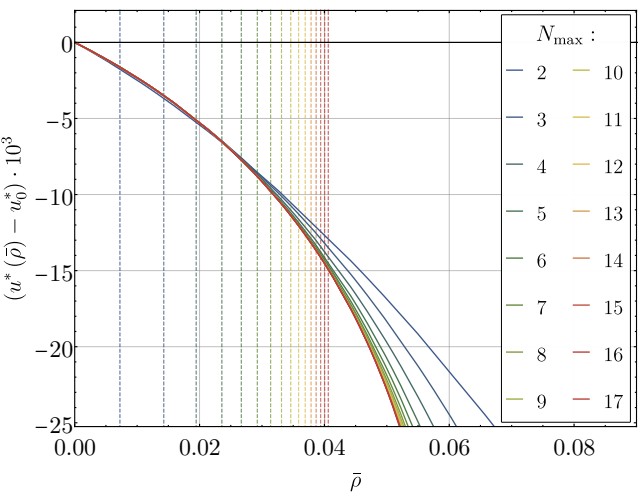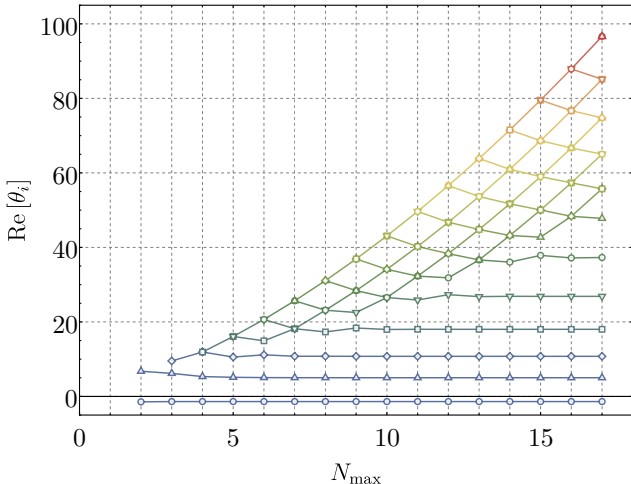

**Figure 14**. UV potential (left) and critical exponents (right) at different Taylor expanded orders for the gravitational fixed point (A) in Figure 9 with $g_h = 0.219$ and $\mu_h = -0.730$. The vertical dashed lines in the left panel show the reliability regimes at each order of the Taylor expansion with a relative error of 1% as defined in (55). The fixed point has one relevant direction with lowest critical exponent $\min\left[\theta_i^{(A)}\right] = -1.39$.

converged. Remarkably, the higher-order critical exponents always form complex conjugate pairs. All critical exponents are strongly shifted towards irrelevance compared to their canonical value, for example, $\theta_{17} = \mathcal{O}(100)$ while the canonical value is 30. Moreover, a single relevant direction is found. Recall from the discussion in Section V B that the transition from green to red regimes is given by the disappearance of a relevant direction. To further investigate this regime and in particular the boundary to the green region, it would be interesting to use a non-trivial background for the scalar field.

Point (B) falls in a regime where two fixed-point potentials exist. Additionally to the non-trivial flat potential discussed in Figure 5, a stable potential is found, shown in Figure 15. The fixed-point values for the lowest order expansion read

$$\mu_\Phi^* = 0.0487\,, \qquad \lambda_{\Phi,4}^* = -0.0758\,. \qquad (D4)$$

Note their similar absolute magnitude. This property is preserved for higher orders and shows the peculiarity of this solution. This potential is stable within the reliability regime of the Taylor expansion but no statements can be made beyond this regime.

The critical exponents display a very quick convergence even at low orders of the Taylor expansion, see the right panel of Figure 15. Compared to the critical exponents of point (A), the critical exponents remain close to their canonical values. Interestingly, this potential has no relevant directions as the minimal critical exponent reads

$$\min\left[\theta_i^{(B)}\right] = 0.043\,. \qquad (D5)$$

Therefore this fixed point shows maximal predictivity as all SM parameters related to the Higgs potential are

predicted by the fixed point, namely, the ratio between the top and Higgs mass as well as the ratio between the Planck and EW scale. However, the trajectory emanating from this fixed-point potential does not lead to a physical IR limit in the present truncation. Particularly, this trajectory shows a negative quartic coupling for all scales below the Planck scale and does not generate a $k_{SSB}$ scale.

## Appendix E: Systematics

In this section, we discuss the systematics of the present approximations as well as consistency checks and partial systematic error estimates. Specifically, we comment on the general structure of the flows in the combined vertex and derivative expansion, and the systematic extension of the current approximation in Appendix E 1. In Appendix E 2, we discuss specifics of the gravity-matter intersection couplings and the embedding of this part in other works as well as the algorithmic effort, mainly due to this sector and pure gravity. In Appendix E 3 we discuss the systematic error introduced by adjusting the present MOM² strong coupling to $\overline{\text{MS}}$ values. In Appendix E 4, we discuss the effects of a non-trivial background on the Higgs- and Goldstone-type of Yukawa couplings.

### 1. Anomalous dimensions

In this expansion scheme the flows projected on the dimensionless couplings $\vec{\lambda} = (\mu, g_3, \lambda_3, ..., )$ of the gravity-

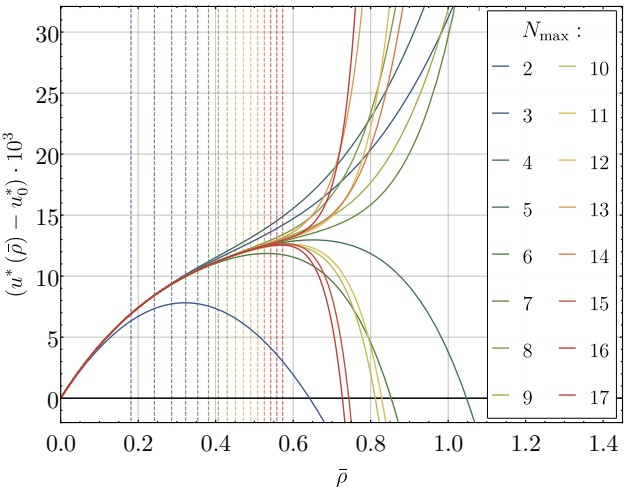
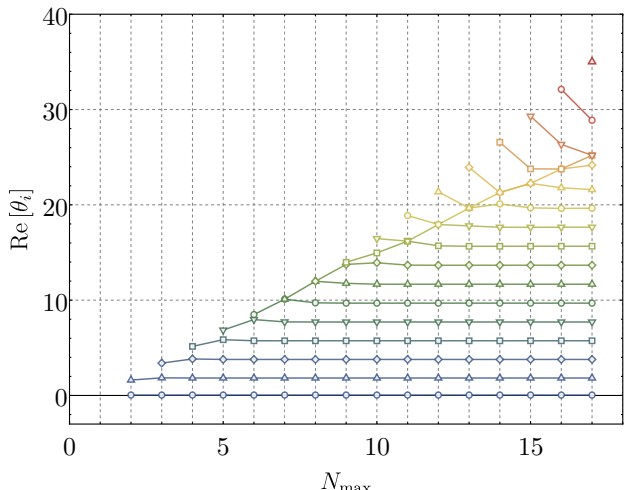

**Figure 15.** UV potential (left) and critical exponents (right) at different Taylor expanded orders for the gravitational fixed-point (B) in Figure 9 with $g_h = 0.190$ and $\mu_h = -0.695$. The vertical dashed lines in the left panel show the reliability regimes at each order of the Taylor expansion with a relative error of 1% as defined in (55). The minimum critical exponent is $\min\left[\theta_i^{(\mathrm{B})}\right] = 0.043$ and therefore, the fixed point (B) has no relevant direction and shows a maximal predictivity. As discussed in Section V A, the IR trajectory of this FP does not lead to a physical IR.

SM system (including the mass parameters) read

$$\left(\partial_t + d_{\lambda_i} + \frac{1}{2}\sum_{n=1}^{n_i} \eta_{\phi_{i_n}}(0)\right)\lambda_i = F_{\lambda_i}^{(0)} + \sum_n \eta_{\phi_n}(k)F_{\lambda_i,n}^{(1)},$$

(E1)

where $d_{\lambda_i}$ are the canonical dimensions of the couplings $\lambda_i$ and we have split the right-hand side into the contributions at vanishing anomalous dimensions $F_{\lambda_i}^{(0)}$ and proportional to the anomalous dimensions $F_{\lambda_i}^{(1)}$. The anomalous dimension on the left-hand side is evaluated at $p = 0$ and arises from the scale derivative of the wave-function renormalisation. On the right-hand side, the anomalous dimensions are evaluated at the cutoff momenta $k$ and come from the regulator insertion in the diagrams. The latter is an approximation as the anomalous dimensions are integrated over the loop momentum $q$. Since the integrands of the flow peak at about $q^2 \approx k^2$, it is a well-working approximation to approximate $\eta_{\phi_i}(q^2) \to \eta_{\phi_i}(k^2)$.

The equation for the momentum-dependent anomalous dimensions has a similar structure,

$$\eta_{\phi_i}(p) = F_{\eta_{\phi_i}}^{(0)}(p) + \sum_n \eta_{\phi_n}(k)F_{\eta_{\phi_i},n}^{(1)}(p).$$

(E2)

Due to the loop momentum approximation $\eta_{\phi_i}(q^2) \approx \eta_{\phi_i}(k^2)$ we only obtain a closed set of equations for $p = k$. Its solution can be obtained by inverting the resulting matrix from coupled system of anomalous dimensions. Given the large field content of the ASSM, the complete system of coupled equations cannot be solved exactly and we resort to an iterative process. Specifically, we replace $\eta_{\phi_n}(k) \to \eta_{\phi_n}(0)$ on the right-hand of (E1) and, at lowest

order, use (E2) evaluated at $p = 0$ with $\eta_{\phi_n}(k) = 0$. At the next order, we do not set $\eta_{\phi_n}(k) = 0$ in (E2) but instead replace $\eta_{\phi_n}(k) \to \eta_{\phi_n}(0)$ and use again (E2) evaluated at $p = 0$ with $\eta_{\phi_n}(k) = 0$ for it. This defines the iterative process that we use. We used the first-order approximation of this iteration and found a good agreement with the zeroth-order approximation. This highlights the subleading role of the higher-order anomalous dimensions in the sub-Planckian regime of the ASSM, and in consequence we use the zeroth-order approximation for the results presented here.

### 2. Gravity-matter couplings

In our implementation of the interplay of gravity and matter, we distinguish the effects of matter on gravity from the gravity on matter. For the former we rely on particular studies of scalar-fermion- [16, 74] and gauge-gravity [23] systems to compute the effect of the SM matter content on the Newton coupling and the cosmological constant. The results there presented are easily extendable to arbitrary matter contents and provide a consistent RG-scheme with inclusion of mass thresholds.

For the effects of gravity on matter, we employed the same truncation as the cited literature and proceeded to compute the full entangled SM-gravity system. The number of $n$-point functions relevant to the theory significantly increases and needs for automatization. For example, the SM alone presents $\sim 130$ $n$-point functions while coupled to gravity $\sim 270$. For the derivation of the $n$-point functions we relied on the Mathematica package VERTEXPAND [132].

There are some subtle points in the inclusion of gravity along the EW sector and non-trivial backgrounds of the Higgs field that require further explanation. The gauge-fixing and ghost actions do not produce graviton interactions due to the background metric used in these terms. It is important to note that as this does not affect standard gauge-fixings (as the used in the strong sector), it does in the special $R_\xi$-gauge. Since the full metric is implicit in the contraction of the covariant derivatives applied on the scalar kinetic term, mixings between Goldstone, gauge and graviton fields do not cancel with the corresponding term generated from an $R_\xi$-gauge-fixing action containing the full metric. This naturally creates additional diagrams contributing to the self-energy of gauge and Goldstone bosons.

In general, these diagrams do not pose a severe problem as they are proportional to the background of the Higgs field. Moreover, in the EW broken phase, the gravitational coupling is negligible, and at Planckian scales, the flow of the curvature mass indicates that $\mu_\Phi > 0$, e.g. a vanishing trivial minimum.

### 3. IR-QCD

In Section IV A, we have discussed the determination of the strong coupling in the present approach: its value in the $\overline{\text{MS}}$-scheme at $k = M_Z$ is given by

$$\bar{\alpha}_s := \alpha_{s,k=M_Z} = \frac{\left(g_{3,k=M_Z}^{\overline{\text{MS}}}\right)^2}{4\pi} \approx 0.118\,. \quad \text{(E3)}$$

It has been shown in [68], that the MOM$^2$ RG-scheme used in the fRG, leads to an enhancement of the running gauge couplings $\alpha_s(p)$ compared to the $\overline{\text{MS}}$ value. In pure Yang-Mills, the respective enhancement factor for $\alpha_s$ is approximately $4/3$, [65], dropping to an enhancement factor of approximately 1.2 in the 2+1 flavour case (measured in the regime between 10-40 GeV). Moreover, in these works it is also shown that $\alpha_{s,k}(0) \lesssim \alpha_{s,k=0}(p=k)$, leading to a further (subleading) enhancement.

Being short of a full analysis, which requires the full implementation of IR-QCD, we use a linear extrapolation in flavour numbers, which leads to an enhancement factor of 1.07 in the ASSM compared to the $\overline{\text{MS}}$ value at the $M_Z$ scale

$$\alpha_{s,k=M_Z} = 1.07\,\bar{\alpha}_s\,. \quad \text{(E4)}$$

Furthermore, we use a variation of the strong MOM$^2$ coupling $\alpha_s$ at $k = M_Z$ as an error estimate for our computation. We vary the strong coupling in the range

$$\alpha_{s,k=M_Z} \in \left[\bar{\alpha}_s,\, 1.10\,\bar{\alpha}_s\right]. \quad \text{(E5)}$$

The main impact of this variation is the shift of the zero crossing of the quartic Higgs coupling, commonly referred to as metastability scale $k_{\text{meta}}$. In Figure 16, we show $k_{\text{meta}}$ as a function of the strong coupling (E5). For our

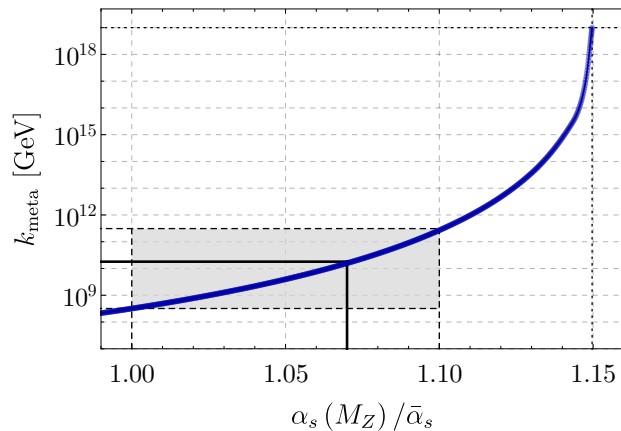

**Figure 16**. Metastability scale as a function of $\alpha_{s,k=M_Z}/\bar{\alpha}_s$. Here, $\bar{\alpha}_s \approx 0.118$ is the $\overline{\text{MS}}$-value with at the EW scale $p = M_Z$ (and $k = 0$), see (E3). For each value of $\alpha_{s,k=M_Z}$ the remaining parameters of the ASSM are set to reproduce the experimental values. Throughout this work, we use $\alpha_{s,k=M_Z} = 1.07\,\bar{\alpha}_s$ marked by a plain line, see (E4). The shaded interval, defined in (E5), is used as a systematic error estimate.

central value (E4), we find a metastability scale slightly above $10^{10}$ GeV in good agreement with perturbative computation in the $\overline{\text{MS}}$-scheme [109]. Interestingly, for $1.15\,\alpha_s$ this scale hits the Planck scale,

$$k_{\text{meta}}(1.15\,\bar{\alpha}_s) = \frac{1}{\sqrt{G_{\text{N}}}}\,, \quad \text{(E6)}$$

where $G_{\text{N}}$ is the physical value of the Newton constant at $p = 0$. The metastability scale is computed in a $\phi^4$ approximation of the Higgs potential, which is a good approximation below the Planck scale. Around the Planck scale, this approximation cannot be trusted but we nonetheless might expect the disappearance of a metastability scale for a large enough strong coupling. This concludes our systematic error analysis of the strong coupling.

### 4. Yukawa couplings

The Yukawa couplings in the fermion-scalar interactions are subject to a scalar-field dependence $\vec{y}(\rho)$. Such is key to reproducing multi-scalar production from fermion scattering, see for example [44, 133]. Although this dependency is not explicitly considered in the Yukawa couplings defined in (A10) the flow equation is known to generate such non-trivial contributions. As the flow is evaluated on the equations of motion, the scalar field dependency is only generated in the modes with a non-trivial background. To quantify such difference, we take advantage of the previous point and define two Yukawa couplings,

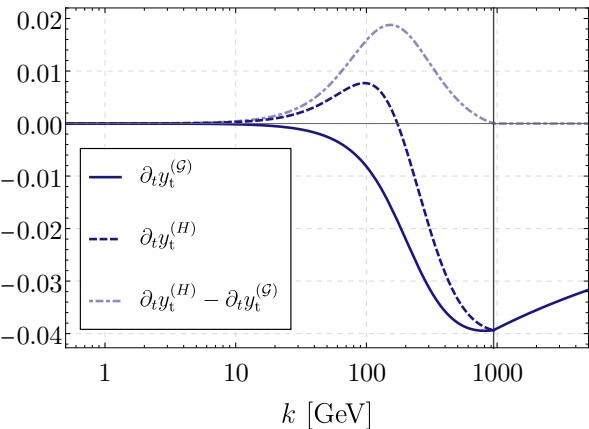

**Figure 17**. Flow of the top-quark Godstone (plain) and Higgs (dashed) Yukawa couplings defined in (E7) and their difference (dot-dashed) (E8). The difference is given by the scalar-field dependence of the Yukawa coupling. In the symmetric phase both couplings agree due to equal vanishing background.

the coupling to Higgs and Goldstones

$$
\partial_t y_t^{(H)}(\bar\rho) = \mathrm{Tr}\Big[ P_{Ht\bar t}(p)\mathrm{Flow}^{(3)}_{Ht\bar t}\Big]
$$
$$
+ \frac{1}{2}(\eta_H + \eta_{t_L} + \eta_{t_R}) y_t^{(H)}(\bar\rho)\,,
$$
$$
\partial_t y_t^{(\mathcal{G})}(\bar\rho) = \mathrm{Tr}\Big[ P_{\mathcal{G}^0 t\bar t}(p)\mathrm{Flow}^{(3)}_{\mathcal{G}t\bar t}\Big]
$$
$$
+ \frac{1}{2}(\eta_{\mathcal{G}^0} + \eta_{t_L} + \eta_{t_R}) y_t^{(\mathcal{G})}(\bar\rho)\,.
$$
$$(E7)$$

While the former contains the complete dependency in the broken phase, by subtracting the latter we find

$$
\partial_t y_t^{(H)}(\bar\rho) - \partial_t y_t^{(\mathcal{G})}(\bar\rho) = \bar\rho\ \partial_t|_{\bar\rho}\Big( \partial_{\bar\rho} y_t^{(\mathcal{G})}(\bar\rho)\Big)\,. \qquad (E8)
$$

In fact, one single Yukawa coupling enters the theory but distinguishing as in (E7) and computing two different flows allows us to obtain the differences caused by a non-trivial background. All flows in (E7) and (E8) are shown in Figure 17. The flow of the Higgs Yukawa coupling shows a maximum at $k \sim 100\,\mathrm{GeV}$, which is not present in the Goldstone coupling. In the symmetric phase $k > k_{\mathrm{SSB}}$, both couplings agree due to equal vanishing background. This shows that there is a small qualitative difference between the two couplings. In the SM flows in Figures 3 and 4, we did not include both couplings separately but assumed them to be equal. The Goldstone Yukawa couplings were chosen as their appearance in fermion-scalar vertices is predominant.

## Appendix F: Pole masses

For $k \to 0$, the relevant parameters of the ASSM can be determined by respective particle physics measurements and the classical (low energy) Newton coupling

and the cosmological constant. The matter and gauge couplings can be determined by selected cross sections with a scattering momentum of the order $\lesssim 10^2\,\mathrm{GeV}$, typically chosen close to the EW scale. Most of these couplings and masses are not a very sensitive input, with the exception of the pole mass of the top quark: it has a relatively large experimental error, and its value (as the largest mass parameter) has a considerable influence on stability estimates of the Higgs potential.

We argue now, that its determination can be obtained with a very small systematic error from the current approximation: to begin with, the missing momentum dependence of the matter couplings and propagators discussed in Section II B has a subleading impact on the flow of couplings, wave-function renormalisation and masses at vanishing momentum. This has been checked thoroughly in IR-QCD ($p \lesssim 10\,\mathrm{GeV}$), where the genuine momentum dependence is far stronger than here due to the significantly larger coupling and the resonant hadronic correlations, see in particular [66], utilising also the momentum dependent QCD correlation functions from [59].

However, while the feedback of the momentum dependences is subleading, they are required for a determination of the coupling strengths via scattering processes and the determination of the pole masses. Specifically, for the present task of computing the top-pole mass $M_{t,\mathrm{pole}}$, the momentum-dependent mass function of the top quark is required. We parametrise the two-point function as

$$
\Gamma^{(2)}_{t\bar t}(p) = Z_{t,k}(p)\left[\mathrm{i}\slashed{p} + M_{t,k}(p)\right]. \qquad (F1)
$$

with the cutoff- and momentum-dependent wave function $Z_{t,k}(p)$ and mass function $M_{t,k}(p)$. For momentum-independent dressing functions this reduces to the approximation used in the present work,

$$
Z_{t,k} = Z_{t,k}(p=0)\,, \qquad m_{t,k} = M_{t,k}(p=0)\,. \qquad (F2)
$$

At $k = 0$, (F1) leads us to the propagator

$$
G_t(p) = \frac{1}{Z_t(p)} \frac{1}{p^2 + M_t(p)^2}\left[-\mathrm{i}\slashed{p} + M_{t,k}(p)\right], \qquad (F3)
$$

with $Z_t(p) = Z_{t,k=0}(p)$ and $M_t(p) = M_{t,k=0}(p)$. The pole mass $M_t^{(\mathrm{pole})}$ is extracted at $k = 0$ from

$$
\left[p^2 + M_t(p)^2\right]_{p^2 = -\left[M_t^{(\mathrm{pole})}\right]^2} = 0\,, \qquad (F4)
$$

or equivalently from

$$
\left[Z_t(p)p^2 + Z_t(p)M_t(p)^2\right]_{p^2 = -\left[M_t^{(\mathrm{pole})}\right]^2} = 0\,, \qquad (F5)
$$

which follows from (F4) by multiplying with $Z_t(p)$.

In the following, we determine the pole mass by using (F5): The combination $Z_t(p)M_t(p)^2$ is obtained analytically from the integrated flow of the top two-point function

$$
\partial_t[Z_t(p)M_t(p)] = \frac{1}{4}\,\mathrm{tr}_D\ \partial_t\Gamma^{(2)}_{t\bar t}(p)\,, \qquad (F6)
$$

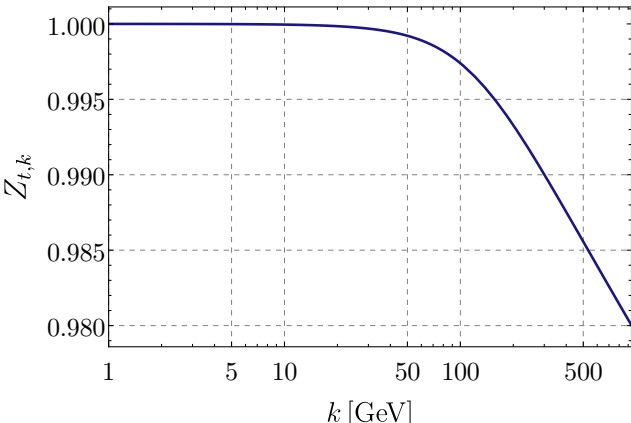

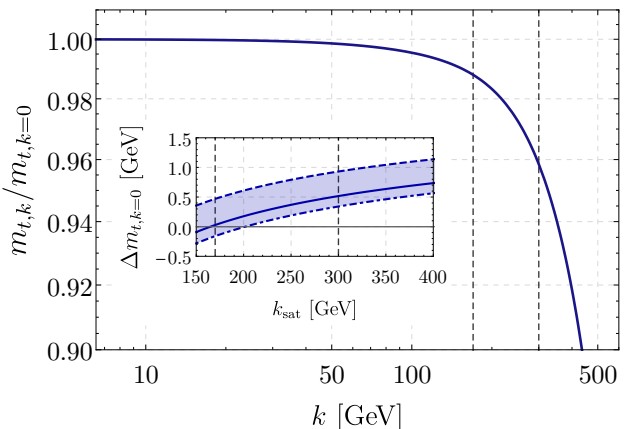

**Figure 18**. Wave-function renormalisation $Z_{t,k}$ of the top quark as a function of the cutoff scale $k$. The $k$-dependence in the broken phase is negligible, and is well approximated by $Z_{t,k} = 1$, leading to the estimate $Z_t(p) = 1$ in (F8).

**Figure 19**. Top mass parameter $m_{t,k}$ measured in units of $m_t = m_{t,k=0}$ as a function of the cutoff scale. The mass parameter saturates at cutoff scales $k \approx 200\,\mathrm{GeV}$, and the vertical lines indicate the borders of the cutoff range (F11) for $k_{\mathrm{sat}}$. In the inlay, we depict the variation of $m_t$ required for the experimental pole mass (45e) as a function of the saturation scale $k_{\mathrm{sat}}$ used for $g_{3,k=k_{\mathrm{sat}}}$ in (F21). The upper dashed line is obtained for $\alpha_s = \bar{\alpha}_s$, the lower dot-dashed line is obtained for $\alpha_s = 1.1\,\bar{\alpha}_s$ (see (E5)) and the central plain line for the ASSM flavour number extrapolation $\alpha_s = 1.07\,\bar{\alpha}_s$. This covers the uncertainty in mapping the $\overline{\mathrm{MS}}$-value of the coupling to the $\mathrm{MOM}^2$ value used in the fRG computation.

with the Dirac trace $\mathrm{tr}_D$. The analytic momentum dependence allows us to resolve the mass function for timelike momenta $p^2 < 0$.

The second ingredient in (F5) is the wave function renormalisation $Z_t(p)$. Here we use that the momentum dependence of correlation functions $\mathcal{O}_{k=0}(p)$ with a mild cutoff dependence $\mathcal{O}_k(p=0)$ at vanishing momentum is typically well approximated by

$$\mathcal{O}_{k=0}(p) \approx \mathcal{O}_{k=\alpha p}(p=0)\,. \tag{F7}$$

This has also been discussed in Section II B 1, and is well-tested in QCD, where quantitative correlation with the full momentum dependence as well as the cutoff dependence have been computed and compared, see in particular [57, 59, 65, 66]. In the present case of the wave function renormalisation of the top quark, its cutoff dependence is negligible in the symmetry broken phase for $k \le k_{\mathrm{SSB}} = 930\,\mathrm{GeV}$, see (47). Indeed, the $k$-dependence of $Z_{t,k}$ is very small, as shown in Figure 18, where $Z_{t,k}$ is shown for $k \lesssim k_{\mathrm{SSB}}$.

In summary, the above analysis suggests that a quantitative estimate is already obtained with $Z_t(p) = 1$ also for timelike momenta within the regime $|p^2| \le (210^2\,\mathrm{GeV})^2$. From the minimal variation of the cutoff dependence in Figure 18 in this regime, we estimate the respective systematic error with $0.5\%$, which translates into an error of roughly $0.8\,\mathrm{GeV}$ for the top pole mass determination. A direct computation will be presented elsewhere, together with further pole mass determinations.

Within this approximation we are led to

$$\left[ p^2 + Z_t(p)M_t(p)^2 \right]_{p^2 = -\left[M_t^{(\mathrm{pole})}\right]^2} \approx 0\,, \tag{F8}$$

and it is left to compute the combination $Z_t(p)M_t(p)^2$ for timelike momenta $p^2 \le 0$, in particular including the pole position. We utilise that the sub-Planckian structure of the flow facilitates the Wick rotation. First, the mass

function does not receive contributions for cutoff scales $k \ge k_{\mathrm{SSB}} \approx 10^3\,\mathrm{GeV}$, see (47). For larger scales, the theory is chiral and hence $M_{t,k \ge k_{\mathrm{SSB}}}(p) = 0$. Accordingly,

$$Z_t(p)M_t(p) = \int_{k_{\mathrm{SSB}}}^0 \frac{\mathrm{d}k}{k}\, \partial_t[Z_t(p)M_{t,k}(p)]\,. \tag{F9}$$

For scales $k \le k_{\mathrm{SSB}}$, the SM couplings quickly settle at their IR value and can be taken to be $k$-independent with the exception of the strong coupling, see Figure 3.

The strong coupling changes significantly between $k_{\mathrm{SSB}}$ and $k = 0$, see Figure 3. We now use, that the flow of the top mass quickly saturates, see Figure 19, and does receive negligible contributions below the top scale. More specifically, the QCD contributions in the flow are proportional to

$$\frac{\alpha_{s,k}}{\left(1 + \frac{M_{t,k}^2}{k^2}\right)^3}\,, \tag{F10}$$

and higher powers in the denominator. These terms drop rapidly for $k < M_{t,k}$ towards smaller cutoff scales despite the rising coupling, which is responsible for the saturation of the QCD part of the flow of $m_{t,k}$. Accordingly, we can estimate its contribution by using an average coupling at the saturation scale $k_{\mathrm{sat}}$. We estimate this saturation scale $k_{\mathrm{sat}}$ very conservatively as

$$k_{\mathrm{sat}} \in [170\,,\,300]\,\mathrm{GeV}\,, \tag{F11}$$

and use a respective constant strong coupling $g_3(k_{\text{sat}})$ in the following computation of the pole mass. Varying this coupling for cutoffs in the range (F11) provides us with a very conservative systematic error estimate. Moreover, the above argument implies that the minimal value

$$k_{\text{sat}} = 170 \, \text{GeV} \,, \tag{F12}$$

is a good choice for the computation of the mass parameter $m_t$ from the pole mass condition (F8). Finally, we have to also take into account the uncertainty in the size of the coupling due to mapping the $\overline{\text{MS}}$-coupling to that in the MOM$^2$ scheme in the fRG. Hence, we additionally employ the conservative estimate (E5) and vary the coupling between its $\overline{\text{MS}}$-value $\bar{\alpha}_s$ and $1.1 \, \bar{\alpha}_s$.

In summary this leaves us with the explicit cutoff dependence of the regulator in the loops contributing to $\partial_t \left[ Z_t(p) M_{t,k}(p) \right]$. Then, the flow is (explicitly) a total derivative w.r.t. to the cutoff scale,

$$\partial_t \left[ Z_t(p) M_{t,k}(p) \right] = \frac{\mathrm{d}}{\mathrm{d}t} \left[ Z_t M_t \right]_k^{\text{1loop}}(p) \,, \tag{F13}$$

where $\left[ Z_t M_t \right]_k^{\text{1loop}}(p)$ are the standard one-loop diagrams with all couplings and parameters taken at $k = 0$ and the only cutoff dependence in the loops comes from the regulators, either $R_{k=0} = 0$ or $R_{k=k_{\text{SSB}}}$ in the second term. Consequently, in this regime the flow can be integrated analytically, and leads to a difference of standard one-loop diagrams $\left[ Z_t M_t \right]_k^{\text{1loop}}(p)$,

$$\Delta \left[ Z_t M_t \right]^{\text{1loop}}(p) = \left[ Z_t M_t \right]_{k=0}^{\text{1loop}}(p) - \left[ Z_t M_t \right]_{k=k_{\text{SSB}}}^{\text{1loop}}(p) \,. \tag{F14}$$

Now we use that for $p \ll k_{\text{SSB}}$ the second term is well approximated by its value at $p = 0$. This leads us to a simple one-loop expression with the full (fRG-resummed) couplings and vertices at $k \to 0$,

$$\left[ Z_t M_t \right]^{\text{1loop}}(p \lesssim k_{\text{SSB}}) = Z_t m_t + \Delta \left[ Z_t M_t \right](p) \,, \tag{F15}$$

with the wave function $Z_t = Z_{t,k=0}(p = 0)$ and top mass parameter $m_t = M_{t,k=0}(p = 0)$ in our approximation, and

$$\Delta \left[ Z_t M_t \right](p) = \left[ Z_t M_t \right]^{\text{1loop}}(p) - \left[ Z_t M_t \right]_t^{\text{1loop}}(0) \,. \tag{F16}$$

The first term $Z_t m_t$ in (F15) is a direct result from the ASSM flow. The second term, $\Delta \left[ Z_t M_t \right](p)$ has the form of the one-loop gap equation for the $t$-quark, subtracted at $p = 0$; however with the full coupling and mass parameters of the SM at $k = 0$. It is a finite difference of loops and can be evaluated within dimensional renormalisation for computational convenience. Its momentum dependence can be read off from the scalar parts of the one-loop self-energy diagrams in (F6). Generically, the momentum

structure of the diagrams reads

$$\mathcal{I}(p, m_i) = \mu^{2\varepsilon} \int \frac{\mathrm{d}^d q}{(2\pi)^d} \int_0^1 \mathrm{d}x$$

$$\times \left[ \frac{1}{[q^2 + \Delta(p)]^2} - \frac{1}{[q^2 + \Delta(0)]^2} \right]$$

$$= -\frac{1}{(4\pi)^2} \int_0^1 \mathrm{d}x \, \log \frac{\Delta(p)}{\Delta(0)} \,, \tag{F17}$$

where $d = 4 - 2\varepsilon$, $\mu$ is the RG scale in dimensional regularisation, and $\Delta$ is

$$\Delta(p) = m_i^2(1 - x) + x \, m_{\text{t}}^2 + x(1 - x)p^2 \,. \tag{F18}$$

The right-hand side of (F17) does not depend on $\mu$ as the integral does not require renormalisation in the first place and the use of dimensional regularisation was only for computational means.

In (F17), $\Delta(p)$ with $m_i = 0$ occurs in the top self-energy diagrams with massless modes such as gluons and photons (at the relevant scales), and the general case with $m_i \neq 0$ is used in top self-energy diagrams with massive modes such as the intermediate vector bosons $Z^0, W^\pm$.

For general masses $m_i$, the self-energy diagrams are computed as

$$\mathcal{I}(p^2, m_i)$$

$$= 1 + \frac{1}{2} \left( \frac{m_{\text{t}}^2 + m_i^2}{m_{\text{t}}^2 - m_i^2} + \frac{m_{\text{t}}^2 - m_i^2}{p^2} \right) \log \frac{m_{\text{t}}^2}{m_i^2}$$

$$- \frac{\sqrt{m_i^4 + (m_{\text{t}}^2 + p^2)^2 + 2m_i^2(p^2 - m_{\text{t}}^2)}}{p^2}$$

$$\times \operatorname{arcoth} \left[ \frac{m_{\text{t}}^2 + m_i^2 + p^2}{\sqrt{m_i^4 + (m_{\text{t}}^2 + p^2)^2 + 2m_i^2(p^2 - m_{\text{t}}^2)}} \right] \,. \tag{F19}$$

For massless modes, $m_i = 0$, (F19) reduces to

$$\mathcal{I}(p^2) = 1 - \left( 1 + \frac{m_{\text{t}}^2}{p^2} \right) \log \left( 1 + \frac{p^2}{m_{\text{t}}^2} \right) \,, \tag{F20}$$

where we have introduced $\mathcal{I}(p^2) = \mathcal{I}(p^2, 0)$. With (F19) and (F20), we arrive at the final expression for the momentum-dependent mass function,

$$\Delta[Z_t(p) M_t(p)]$$

$$= \frac{m_{\text{t}}}{16 \, \pi^2} \left[ 4 \, g_3^2 \, \mathcal{I}(p^2) + 3 \left( \frac{2 \, g_2 \sin \theta_W}{3} \right)^2 \mathcal{I}(p^2) \right.$$

$$- \frac{h_t^2}{2} \mathcal{I}(p^2, m_H) + \frac{h_t^2}{2} \mathcal{I}(p^2, m_Z) + h_b^2 \, \mathcal{I}(p^2, m_W)$$

$$\left. - \frac{g_Y^2 \, (1 + 2 \cos 2\theta_W)}{9} \mathcal{I}(p^2, m_Z) \right] \,, \tag{F21}$$

computed in the gauge (A12). The first two terms on the right-hand side stem from gluon and photon mediated diagrams, and the terms in the second line stem from Higgs, $\mathcal{G}^0$, $\mathcal{G}^\pm$, and $Z^0$. Vertices, wave functions and the mass function are given by the full couplings and wave functions at $k = 0$ and $p = 0$ except for the strong coupling $g_3(k_{\text{sat}})$ which is evaluated at $k_{\text{sat}} = 170\,\text{GeV}$, see (F12).

Equation (F15) with (F21) allows us to resolve (F8) for the pole position. This fixes the Euclidean mass parameter to

$$m_t = 165.4^{+0.9}_{-0.2}\,\text{GeV}\,, \tag{F22}$$

for the PDG pole mass (cross section measurements) $M_t = 172.5\,\text{GeV}$, see (45e). The error in (F22) is solely due to the error estimate of the value of the strong coupling used in (F21). It is discussed below (F9), and is summarised in the inlay of Figure 19: The upper variation comes from the variation of the saturation scale up to $k_{\text{sat}} = 300\,\text{GeV}$

with $\alpha_s = \bar\alpha_s(300\,\text{GeV})$, while the lower error is related to the estimate of the maximal MOM$^2$-value of the strong fine structure constant in (E5) with $\alpha_s = 1.1\,\bar\alpha_s(170\,\text{GeV})$.

We consider the (systematic) error estimate in (F22) a very conservative one. Note also that the theoretical systematic error is of the size of the experimental one.

A first prediction concerns the decay width $\Gamma_t$ of the Higgs. We obtain

$$\Gamma^{(\text{theo})}_{t,\text{pole}} = 1.72^{+0.09}_{-0.41}\,\text{GeV}\,. \tag{F23}$$

which agrees quantitatively with that in the PDG one, $\Gamma^{(\text{exp})}_{t,\text{pole}} = 1.42^{+0.19}_{-0.15}\,\text{GeV}$ in (45f). Note, that while the error in (F22) constitutes a systematic error estimate, the error in (F23) only describes the relative weighting of QCD and non-QCD contributions. In a pure QCD system, our analysis would lead to a vanishing error on the decay width in (F23).

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
