# Peer review of "The Asymptotically Safe Standard Model: From quantum gravity to dynamical chiral symmetry breaking"

_SciPost Physics_

## Round 1 · Referee Report · Anonymous · 2023-1-5

Report

In this paper, the authors present the first results of asymptotically safe gravity-matter systems, including the full standard model in the matter sector. To my knowledge, this is the first self-consistent FRG calculation of beta functions for the standard model couplings in the presence of gravity. It corresponds to an important milestone in asymptotically safe quantum gravity literature. The main result was the construction of RG-trajectories connecting an asymptotically safe trans-Planckian regime with a phenomenologically viable IR regime featuring electroweak SSB and chiral SB in the strong sector of the standard model.
In my opinion, the results presented in this paper are sufficiently relevant and deserve to be published in this journal. However, before I give my full recommendation, I would like to ask the authors to address a few points.

1 - The UV completion of the Yukawa sector is quite subtle. In this paper, they achieve the correct sign for UV completion in the Yukawa sector by neglecting one of the diagrams contributing to the flow of the Yukawa couplings. In my opinion, this is not a satisfactory solution for the problem of UV completion in the Yukawa sector. Based on the comments at the beginning of section III.D, the author seems to share a similar opinion. Therefore, I think it would be valuable if they could add a short "roadmap" discussing possible future directions to address the problem of UV completion of Yukawa couplings in a more satisfactory way.

2 - If the gravitational contribution to the flow of the abelian-gauge and Yukawa couplings comes with a negative sign, we might find a second scenario for UV completion featuring non-vanishing abelian-gauge and Yukawa couplings in the fixed point regime. RG trajectories emanating from such a fixed point structure tend to be more "predictive" than trajectories emanating from vanishing matter couplings in the UV. For example, this is the basis for the "prediction" of the top-bottom mass difference from asymptotic safety (arXiv: 1803.04027). I would like to know if the authors explored this possibility in their analysis. In particular, would such a scenario be compatible with a phenomenologically viable IR regime? It would be interesting to see a comment about this question in the paper.

3 - I am very puzzled by their estimate of systematic uncertainties related to truncation choice. In particular, the authors claim: "we believe that this source of error is subleading, significantly smaller than 1 GeV". From what I understood, this value comes from including higher-order contributions to the anomalous dimensions. But it does not take into account all the remaining higher-order operators that we could add to a truncation. In my opinion, the authors should add a more detailed explanation on how to get to such an estimative, or they should not say that the truncation error is significantly smaller than 1 GeV.

  • validity: high
  • significance: high
  • originality: high
  • clarity: high
  • formatting: excellent
  • grammar: excellent

Author:  Álvaro Pastor Gutiérrez  on 2023-03-17  [id 3491]

(in reply to Report 1 on 2023-01-05)
Category:
remark
answer to question
correction

We thank the referee for their careful work, the detailed report, and the suggestions for improvement. Here are our answers to the remarks of the referee.

1.- We agree with the referee that Yukawa sector is quite subtle. In the current approximation, the sign of the gravity contribution to the Yukawa flow is regulator dependent. The sign of the Yukawa flow that leads to a UV fixed point is simulated by a vanishing tadpole contribution. We have discussed in Sec. 3A, how improving the momentum dependences of the flows will stabilize the Yukawa sector. Evidently, this is one of the most pressing issues in the asymptotically safe Standard Model and is subject to current improvements.

The improvements required in the flow of the Yukawa coupling are two-fold: The first concerns the momentum dependences of the $n$-point correlation functions, in particular, that of the Yukawa coupling itself. The second improvement concerns the interaction terms contributing to the Yukawa flow, for instance, higher-order curvature invariants and form factors such as $R \phi \bar \psi \psi$ and $R f(\Delta) R$. We have included this roadmap discussion in the manuscript.

2.- Fixed points with a non-vanishing Yukawa or U(1)-gauge coupling are indeed very interesting due to their predictive nature. In our computation, these fixed points exist but do not lead to a viable phenomenology in the IR: they either lead to a too-large top mass or a too-large fine-structure constant. Our focus was to exactly match the SM in the IR and therefore we do not extensively discuss these solutions in the paper. We have now added a corresponding paragraph.

3.- We emphasise this truncation error is the {\it remaining} error, after accounting for the uncertainty on the fixed-point values, the top mass, and the strong coupling. This remaining error is dominated by the sub-Planckian regime and is therefore similar to that of standard perturbation theory. There, the dominant errors stem from the top mass and the strong coupling, and effects from higher-loop orders are subleading. In straight analogy, we find these errors are dominating and of the order one GeV. Consequently, we find that the remaining truncation error is smaller than 1 GeV.

We understand that the referee is surprised by this small truncation error. The main reason is that the non-perturbative UV regime is decoupling very quickly with a power law and therefore higher-order operators do not strongly affect this particular result of our work.

We have made the unfortunate choice in the manuscript to discuss the remaining error first. We have now changed the ordering of the error discussion.

---

## Round 1 · Referee Report · Anonymous · 2023-2-21

Report

The work “The asymptotically safe standard model: from quantum gravity to dynamical chiral symmetry breaking” uses functional renormalization group methods to study the phase space of quantum gravity supplemented by matter degrees of freedom. As its novel feature, all degrees of freedom present in the standard model of particle physics and many of their interactions are included in the study. Notably, the work reports on important, new results along this research line. Specifically,

1) it identifies a new gravity-matter fixed point. The key feature of this fixed point is the structure of the Higgs potential which comes with two relevant directions. This distinguishes the new fixed point from the Gaussian matter fixed point, since it permits the creation of a non-trivial Higgs potential when going towards energies below the Planck scale.

2) the authors make a significant effort in order to construct the UV-critical hypersurfaces of the gravity-matter fixed point. Specifically, a lot of thought has been gone into placing the standard model of particle physics within this setting.

3) as a main result, the authors argue that the standard model of particle physics is located in the UV-critical hypersurface of their new gravity matter fixed point.

An important insight obtained along this analysis is that including a significant number of power-counting irrelevant couplings in the Higgs potential is essential for making the new fixed point visible.

In my opinion, the work constitutes a milestone in the gravitational asymptotic safety program. It certainly fulfills the publication criteria of SciPost Physics and I am happy to recommend the article for publication.

There are some weaker points about this work which I feel should be mentioned in this report. Given the breakthrough-nature of the work, it is acceptable to postpone the ultimate clarification of these questions to future work:

1) The derivation of the results builds on a series of approximations. These are necessary in order to make the computation technically feasible. Each of these approximations is explained in reasonable detail and the authors provide arguments suggesting their validity in each case. From the main text, it is then difficult to deduce what is actually done and what the impact of the entire set of approximations actually amounts to. In my opinion having an overview on all approximations and identifications made along the computation (either in diagrammatic or tabulated form) would make the work much more accessible.

2) An important point is the discussion of potential sources of theoretical errors in Section VI, as it highlights the effects which could ultimately affect the conclusions drawn in the work. I have the impression that the quoted errors may ultimately be too stringent with respect to the omission of contributions not considered in the present setting.

  • validity: high
  • significance: high
  • originality: high
  • clarity: ok
  • formatting: perfect
  • grammar: perfect

Author:  Álvaro Pastor Gutiérrez  on 2023-03-17  [id 3492]

(in reply to Report 2 on 2023-02-21)
Category:
remark
question

We thank the referee for their careful work, the detailed report, the positive feedback, and the suggestions for improvement. Here are our answers to the remarks of the referee.

1 .- We thank the referee for this useful suggestion. We have now added an Appendix in which we summarise the approximations.

2 .- We agree with the referee that the discussion of the error sources is important for the conclusions of our work. This is precisely why we aimed at a quantitative error estimate. While these errors may seem small considering the complexity of the system, we believe that they provide a fair assessment of uncertainty. We have improved the description of the error sources to further sustain our estimate.

---

## Editorial Decision

resubmitted